# Understanding and Improving Continuous Adversarial Training for LLMs via In-context Learning Theory

**Shaopeng Fu & Di Wang**
Provable Responsible AI and Data Analytics (PRADA) Lab
King Abdullah University of Science and Technology, Saudi Arabia
{shaopeng.fu, di.wang}@kaust.edu.sa

## Abstract

Adversarial training (AT) is an effective defense for large language models (LLMs) against jailbreak attacks, but performing AT on LLMs is costly. To improve the efficiency of AT for LLMs, recent studies propose continuous AT (CAT) that searches for adversarial inputs within the continuous embedding space of LLMs during AT. While CAT has achieved empirical success, its underlying mechanism, *i.e.*, why adversarial perturbations in the embedding space can help LLMs defend against jailbreak prompts synthesized in the input token space, remains unknown. This paper presents the first theoretical analysis of CAT on LLMs based on in-context learning (ICL) theory. For linear transformers trained with adversarial examples from the embedding space on in-context linear regression tasks, we prove a robust generalization bound that has a negative correlation with the perturbation radius in the embedding space. This clearly explains why CAT can defend against jailbreak prompts from the LLM's token space. Further, the robust bound shows that the robustness of an adversarially trained LLM is closely related to the singular values of its embedding matrix. Based on this, we propose to improve LLM CAT by introducing an additional regularization term, which depends on singular values of the LLM's embedding matrix, into the objective function of CAT. Experiments on real-world LLMs demonstrate that our method can help LLMs achieve a better jailbreak robustness-utility tradeoff. The code is available at https://github.com/fshp971/continuous-adv-icl.

## 1 Introduction

While large language models (LLMs) are increasingly adopted in various real-world applications, their safety is found to be compromised by jailbreak attacks (Wei et al., 2023). By feeding jailbreak prompts, which are specially constructed harmful instructions, one can "jailbreak" safety-aligned targeted LLMs to induce harmful behaviors in them. To ensure the robustness of LLMs against jailbreak attacks, one of the most effective defenses is adversarial training (AT) (Mazeika et al., 2024; Fu et al., 2025), which trains LLMs on synthesized jailbreak prompts to help them better recognize and refuse these harmful inputs. However, the synthesis of jailbreak prompts during AT usually requires solving discrete optimization problems and is thus computation-consuming (Zou et al., 2023; Chao et al., 2023), which restricts the use of AT for LLMs in practice.

To improve the efficiency of AT for LLMs, recent studies introduce *continuous AT (CAT)* (Xhonneux et al., 2024; Sheshadri et al., 2024; Arditi et al., 2024; Dékány et al., 2025), which performs AT on LLMs with adversarial inputs synthesized in the LLMs' *continuous* token embedding space. Compared with the vanilla AT for LLMs, CAT can use projected gradient descent (PGD; Madry et al. 2018) to search for adversarial examples in the embedding space, which is significantly faster than that in vanilla AT where one needs to perform a heuristic search to find jailbreak prompts in the input token space. However, despite the empirical success of CAT, the reason behind it is still unknown. In fact, the training data in CAT and vanilla LLM AT are very different from each other: data in CAT are sequences of embedding vectors, while data in vanilla AT are sequences of token

indices. This raises the research question: *Why can adversarial perturbations in the embedding space help LLMs learn to defend against jailbreak prompts from the original input token space?*

In light of recent advances in understanding LLM jailbreak robustness via in-context learning (ICL) theory (Fu et al., 2025; Kumano et al., 2025; Anwar et al., 2025), this paper presents the first theoretical analysis of LLM CAT, also based on ICL theory. We rigorously study how AT helps improve the robustness of linear transformers trained on linear regression tasks against in-context suffix adversarial attacks. To simulate the embedding space adversarial perturbation process in CAT, we introduce an additional trainable embedding matrix into linear transformers and perform adversarial perturbations on ICL input embeddings obtained from this embedding matrix, rather than on the vanilla ICL inputs. Under our new *ICL embedding AT* theoretical framework, we prove a robust generalization upper bound for linear transformers trained via ICL embedding AT. This robust bound has a negative correlation with the embedding space adversarial perturbation radius during ICL embedding AT, which clearly explains why adversarial perturbations in the embedding space can help LLMs defend against jailbreak prompts from the original input space.

Besides, our robust generalization upper bound is closely related to the singular values of the embedding matrix in linear transformers. Specifically, it suggests that an adversarially trained linear transformer with an embedding matrix that has **"not too large nor too small singular values"** would enjoy a small robust upper bound and thus strong adversarial robustness. Based on this finding, we further propose *Embedding Regularized Continuous AT (ER-CAT)*, a new LLM AT approach improved from CAT by introducing the variance of the embedding matrix singular values as an additional regularization term into the objective function of the original CAT. The motivation is to simultaneously reduce large singular values and increase small singular values, thereby helping to improve the robustness of LLMs according to our proven theory. To verify ER-CAT, we conducted experiments on six real-world LLMs and six common jailbreak attacks. Results show that compared with original CAT, ER-CAT helps LLMs achieve a better jailbreak robustness-utility tradeoff.

## 2 RELATED WORKS

**Jailbreak attacks.** Jailbreak prompts (Wei et al., 2023) are adversarial examples (Szegedy et al., 2014; Goodfellow et al., 2015) that can induce targeted unsafe behaviors from LLMs. Existing jailbreak attacks include *token-level* and *prompt-level* attacks. Token-level attacks synthesize jailbreak prompts via modifying/inserting tokens in prompts with heuristic methods (Zou et al., 2023; Sadasivan et al., 2024; Hayase et al., 2024; Zhu et al., 2024; Paulus et al., 2025; Jin et al., 2024; Liao & Sun, 2024; Andriushchenko et al., 2025), while prompt-level attacks use prompts crafted by humans (Wei et al., 2023; Li et al., 2023; Shen et al., 2024; Xu et al., 2024) or LLM-based agents (Chao et al., 2023; Liu et al., 2024b;a; Sabbaghi et al., 2025; Zhang et al., 2025) to jailbreak LLMs. Though the synthesis of jailbreak prompts in token-level attacks and prompt-level attacks is different from each other, CAT has been shown to be able to defend against both types of attacks.

**LLMs adversarial training (AT).** To tackle jailbreak attacks, an effective method is to align LLMs via AT (Madry et al., 2018) to help them better recognize and refuse harmful inputs. A standard LLM AT aims to solve a minimax problem that minimizes the training loss on most adversarial jailbreak prompts (Mazeika et al., 2024). However, searching for jailbreak prompts in the discrete token space of LLMs is resource-intensive, limiting the broader application of LLM AT. More recent studies propose *continuous* AT (CAT) for LLMs, in which adversarial examples are embedding vectors obtained from adversarially perturbing token embeddings of the original prompts (Xhonneux et al., 2024; Casper et al., 2024; Sheshadri et al., 2024; Yu et al., 2025). Such a perturbation process can be efficiently performed with gradient-based optimizations, which thus significantly reduces the computational burden of LLM AT. Dékány et al. (2025) adopt both jailbreak prompts and perturbed prompt embeddings as adversarial examples to further improve the performance of LLM AT.

**In-context learning (ICL) theory.** ICL theory aims to understand how transformer-based LLMs can make decisions for different task-specific sequential context inputs (i.e., "prompts") without adjusting model parameters. Existing ICL works have proven that one can construct explicit transformers layer by layer to mimic the process of learning a variety of function classes (Garg et al., 2022; Von Oswald et al., 2023; Ahn et al., 2023; Chen et al., 2024; Wang et al., 2024; Mahankali et al., 2024; Li et al., 2025). Efforts have also been made to analyze how one-layer transformers can learn ICL prediction abilities from massive task-specific contextual data (Lu et al., 2024; Magen

et al., 2024; Frei & Vardi, 2025; Shi et al., 2024; Zhang et al., 2024; Yang et al., 2024b; Huang et al., 2023; Wu et al., 2024; Lin et al., 2024; Lu et al., 2025). More recent studies have leveraged ICL theory to analyze the adversarial robustness of LLMs. Anwar et al. (2025) find that by only perturbing a single in-context sample in the entire real space, one can manipulate the ICL prediction of transformers to arbitrary results. Fu et al. (2025) study AT of ICL models under more restricted and realistic ICL adversarial attacks, where each ICL sample can only be perturbed within a restricted space. They prove that AT on contextual data with a very small number of perturbed in-context samples can already help trained transformers achieve strong robustness. Kumano et al. (2025) show that adversarially pretrained transformers focus more on robust features (Ilyas et al., 2019) and thus can generalize to downstream tasks robustly without additional AT. Our analysis mainly stems from Fu et al. (2025), with the goal of explaining the robust generalization ability of continuous AT.

## 3 PRELIMINARIES

**Large language models (LLMs)**. An LLM is a function that maps sequential inputs to sequential outputs based on a parameterized distribution $p_\theta$. Let $x \in \mathcal{V}^{|x|}$ be an input prompt of length $|x|$, where $\mathcal{V} := \{1, \cdots, |\mathcal{V}|\}$ is the token space. The probability that the LLM generates a response $y \in \mathcal{V}^{|y|}$ of length $|y|$ is: $p_\theta(y|x) = \prod_{i=1}^{|y|} p_\theta(y_i|x \oplus y_{1:(i-1)})$, where "$\oplus$" denotes concatenation.

**Jailbreak attacks.** Current jailbreak attacks can be divided into *token-level* and *prompt-level* attacks. Given two token sequences $x^{(h)}$ and $y^{(h)}$, where $x^{(h)}$ is a harmful instruction and $y^{(h)}$ is a targeted harmful response, a standard token-level attack concatenates a synthesized adversarial suffix $x^{(s)}$ to the prompt $x^{(h)}$ to form a jailbreak prompt that increases the probability of the LLM in generating $y^{(h)}$. The synthesis of the suffix $x^{(s)}$ can be formalized as solving the below problem,

$$\min_{x^{(s)} \in \mathcal{V}^{|x^{(s)}|}} - \log p_\theta(y^{(h)}|x^{(h)} \oplus x^{(s)}). \tag{1}$$

Meanwhile, a prompt-level attack uses an attack oracle $\mathcal{A}$, which can be human experts or AI agents, to directly rewrite the original harmful prompt $x^{(h)}$ to a jailbreak one $\hat{x}^{(h)}$, as shown below,

$$\min_{\hat{x}^{(h)} \sim \mathcal{A}(x^{(h)})} - \log p_\theta(y^{(h)}|\hat{x}^{(h)}). \tag{2}$$

It should be noted that performing both token-level and prompt-level attacks is resource-consuming, as solving Eq. (1) requires sophisticated discrete optimizations, while solving Eq. (2) requires human annotation or additional computing resources for AI agents to perform inference.

**Continuous AT for LLMs**. Let $D^{(h)}$ be a *safety dataset*, where each sample $(x, y, \tilde{y})$ consists of a harmful prompt $x$, a targeted harmful response $y$, and a *safe* reference response $\tilde{y}$. Additionally, let $D^{(u)}$ be a *utility dataset*, where each sample $(x, y)$ consists of a pair of a normal instruction and its answer. Then, Mazeika et al. (2024) formalize the first AT algorithm for LLMs as solving the following optimization problem:

$$\min_\theta \left\{ - \underset{(x,y,\tilde{y}) \in D^{(h)}}{\mathbb{E}} \underbrace{\left[ \log p_\theta(\tilde{y}|\hat{x}^*) + \log(1 - p_\theta(y|\hat{x}^*)) \right]}_{\text{Adversarial Loss}} - \underset{(x,y) \in D^{(u)}}{\mathbb{E}} \underbrace{\log p_\theta(y|x)}_{\text{Utility Loss}} \right\}, \tag{3}$$

where $\hat{x}^* = \arg\max_{\hat{x} \in \mathcal{B}(x,y)} \log p_\theta(y|\hat{x})$ is the jailbreak prompt for the harmful input-output pair $(x, y)$ from the safety set and $\mathcal{B}(x, y)$ denotes the search space of jailbreak prompts. In Eq. (3), the adversarial loss helps LLMs learn to respond harmlessly even when the most adversarial jailbreak prompts are present, while the utility loss helps retain the utility of pre-trained LLMs. As explained before, finding strong jailbreak prompts $\hat{x} \in \mathcal{B}(x, y)$ from the search space determined by the used attacks is costly, which limits the efficiency of LLM AT.

More recent studies suggest using continuous AT (CAT) to reduce the computational overhead of vanilla LLM AT. Let $\mathcal{E}(\cdot)$ be the embedding function of the LLM $f_\theta$, with an embedding matrix $W^E \in \mathbb{R}^{d \times |\mathcal{V}|}$ as its parameter, where $W^E_{:,v} \in \mathbb{R}^d$ is the embedding vector for the token $v \in \mathcal{V}$. For any token sequence $x \in \mathcal{V}^{|x|}$, $\mathcal{E}(\cdot)$ maps it to its embedding sequence as $\mathcal{E}(x) := (W^E_{:,x_1} \cdots W^E_{:,x_{|x|}}) \in \mathbb{R}^{d \times |x|}$. CAT is then formalized as solving the below problem (Xhonneux et al., 2024),

$$\min_\theta \left\{ -\alpha \cdot \underset{(x,y,\tilde{y}) \in D^{(h)}}{\mathbb{E}} \log \frac{p_\theta(\tilde{y}|\mathcal{E}(x) + \delta^*)}{p_\theta(y|\mathcal{E}(x) + \delta^*)} - \underset{(x,y) \in D^{(u)}}{\mathbb{E}} \log p_\theta(y|x) \right\}, \tag{4}$$

where $\delta^* = [\arg\max_{\|\delta_1\|_2, \cdots, \|\delta_{|x|}\|_2 \leq \epsilon} \log p_\theta(y|\mathcal{E}(x) + \delta)] \in \mathbb{R}^{d \times |x|}$ is the *most* adversarial perturbation $\delta^*$ for the harmful input-output pair $(x, y)$ from the safety set $D^{(h)}$, $\epsilon > 0$ is the embedding space perturbation radius, $\alpha > 0$ is a weight parameter, and $(\mathcal{E}(x) + \delta^*) := ((\mathcal{E}(x_1) + \delta_1^*) \cdots (\mathcal{E}(x_{|x|} + \delta_{|x|}^*))) \in \mathbb{R}^{d \times |x|}$ is the perturbed harmful prompt embeddings. The idea behind LLM CAT in Eq. (4) is to adopt adversarially perturbed embedding sequences as adversarial examples rather than jailbreak prompts in the vanilla LLM AT in Eq. (3). Additionally, the embedding space adversarial perturbation $\delta^*$ can usually be searched via the fast projected-gradient descent (PGD) method (Madry et al., 2018), which makes CAT efficient in practice.

## 4 THEORETICAL ANALYSIS FOR CONTINUOUS AT

While LLM CAT has achieved empirical success (Xhonneux et al., 2024; Casper et al., 2024; Sheshadri et al., 2024; Dékány et al., 2025), the mechanism behind it, *i.e.*, **why adversarial perturbations in the embedding space help LLMs defend against jailbreak prompts from the token space**, remains unclear. This section tackles this research question by conducting a theoretical analysis for CAT based on in-context learning theory (Zhang et al., 2024; Fu et al., 2025; Kumano et al., 2025).

**In-context learning (ICL) theory.** In the ICL theory, a *prompt* input of length $N$, specified by a *task* indexed by $\tau$, is formalized as a sequence $(x_{\tau,1}, y_{\tau,2}, \cdots, x_{\tau,N}, y_{\tau,N}, x_{\tau,q})$, where the first $N$ labeled samples $\{(x_{\tau,i}, y_{\tau,i})\}_{i=1}^N$ are task-specific in-context training samples, and the last item, $x_{\tau,q}$, is the query sample. Then, the goal of an ICL model is to make a prediction for the query sample $x_{\tau,q}$ solely based on the $N$ in-context training data.

**ICL linear regression.** Our analysis focuses on training ICL models on different linear regression tasks. Suppose $\tau$ is a task index and $w_\tau \in \mathbb{R}^{d_0}$ is the corresponding task weight drawn from $w_\tau \sim \mathcal{N}(0, I_{d_0})$. We assume that each ICL training point $x_{\tau,i}$ $(1 \leq i \leq N)$ and the query point $x_{\tau,q}$ are drawn from $x_{\tau,i}, x_{\tau,q} \sim \mathcal{N}(0, \Lambda)$ where $\Lambda \in \mathbb{R}^{d_0 \times d_0}$ is the covariance matrix, and their labels are $y_{\tau,i} = w_\tau^\top x_{\tau,i}$ and $y_{\tau,q} = w_\tau^\top x_{\tau,q}$. Then, the ICL input $Z_\tau$ specified by the task $\tau$ is given by

$$Z_\tau := \begin{pmatrix} x_{\tau,1} & \cdots & x_{\tau,N} & x_{\tau,q} \\ y_{\tau,1} & \cdots & y_{\tau,N} & 0 \end{pmatrix} \in \mathbb{R}^{(d_0+1) \times (N+1)}. \tag{5}$$

**Other notations.** We denote $[n] := \{1, \cdots, n\}$ for any $n \in \mathbb{N}^+$. For any $A \in \mathbb{R}^{n \times m}$, we denote $\|A\|_{2,\infty} := \max_{1 \leq i \leq m} \|A_{i,:}\|_2$, $\|A\|_2$ be the operator norm, and $\|A\|_F$ be the Frobenius norm. Besides, $\lambda_i(A)$, $\lambda_{\max}(A)$, and $\lambda_{\min}(A)$ denote its $i$-th largest, largest, and smallest eigenvalues, while $\sigma_i(A)$, $\sigma_{\max}(A)$, and $\sigma_{\min}(A)$ denote its $i$-th largest, largest, and smallest singular values. Finally, we denote $\text{Tr}(A) := \sum_{i=1}^n A_{i,i}$ for any $A \in \mathbb{R}^{n \times n}$. We use standard Big O notation $\mathcal{O}(\cdot)$.

In the remainder, we will first establish an *ICL embedding AT* problem for linear transformers and explain why it can approximate real-world LLM CAT under the ICL theoretical framework. A robust generalization bound will then be proved for linear transformers trained from ICL embedding AT. Based on this bound, we will explain why CAT can work and how to further improve CAT.

### 4.1 ICL ADVERSARIAL TRAINING IN EMBEDDING SPACE

**Linear self-attention with embedding module (LSA-E).** Linear self-attention (LSA) models are linear transformers that have been widely used for theoretical ICL analysis (Zhang et al., 2024; Shi et al., 2024; Frei & Vardi, 2025). However, the LSA model studied in the previous work does not have an input embedding module and thus cannot be naturally adopted for the analysis of LLM CAT, which requires performing adversarial perturbations in the input embedding space. To tackle this challenge, we design a novel **LSA-with-Embedding (LSA-E)** model to approximate real-world CAT in our theoretical analysis. Specifically, let $\mathcal{E}(\cdot)$ be an embedding function that maps any ICL input $Z_\tau \in \mathbb{R}^{(d_0+1) \times (N+1)}$ to its ICL embedding matrix $\mathcal{E}(Z_\tau) \in \mathbb{R}^{(d+1) \times (N+1)}$ as follows:

$$\mathcal{E}(Z_\tau) := \begin{pmatrix} W^E x_{\tau,1} & \cdots & W^E x_{\tau,N} & W^E x_{\tau,q} \\ y_{\tau,1} & \cdots & y_{\tau,N} & 0 \end{pmatrix} \in \mathbb{R}^{(d+1) \times (N+1)}, \tag{6}$$

where $W^E \in \mathbb{R}^{d \times d_0}$ is the trainable parameter of the embedding function and $d$ is the dimension of the embedding space. Intuitively, the function $\mathcal{E}(\cdot)$ in Eq. (6) aims to map each in-context point from

the original input space $\mathbb{R}^{d_0}$ to the embedding space $\mathbb{R}^d$ via a linear mapping. With the embedding function $\mathcal{E}(\cdot)$ in Eq. (6), the LSA-E model $f_{\text{LSAE},\theta}$ is then formalized as below,

$$f_{\text{LSAE},\theta}(Z_\tau) := \left[ \mathcal{E}(Z_\tau) + W^V \mathcal{E}(Z_\tau) \frac{\mathcal{E}(Z_\tau)^\top W^{KQ} \mathcal{E}(Z_\tau)}{N} \right] \in \mathbb{R}^{(d+1)\times(N+1)},$$

where $W^{KQ} \in \mathbb{R}^{(d+1)\times(d+1)}$ is a matrix fused from the key and query projection matrices, $W^V \in \mathbb{R}^{(d+1)\times(d+1)}$ is the value projection matrix, and $\theta := (W^E, W^{KQ}, W^V)$ contains all trainable parameters of the LSA-E model. The prediction $\hat{y}_{q,\theta}(Z_\tau)$ for the query point $x_{\tau,q}$ is given by the right-bottom entry of the LSA-E model output, *i.e.*, $\hat{y}_{q,\theta}(Z_\tau) := f_{\text{LSAE},\theta}(Z_\tau)_{(d+1),(N+1)}$. If we follow Zhang et al. (2024); Frei & Vardi (2025); Fu & Wang (2024) to write matrices $W^{KQ}$ and $W^V$ as $W^\square = \begin{pmatrix} W_{11}^\square & w_{12}^\square \\ (w_{21}^\square)^\top & w_{22}^\square \end{pmatrix}$, $\square \in \{KQ, V\}$, where $W_{11}^\square \in \mathbb{R}^{d\times d}$, $w_{22}^\square \in \mathbb{R}$, and $w_{12}^\square, w_{21}^\square \in \mathbb{R}^{d\times 1}$, then the LSA-E model prediction $\hat{y}_{q,\theta}$ can be further simplified as follows,

$$\hat{y}_{q,\theta}(Z_\tau) := \begin{pmatrix} (w_{21}^V)^\top & w_{22}^V \end{pmatrix} \frac{\mathcal{E}(Z_\tau)\mathcal{E}(Z_\tau)^\top}{N} \begin{pmatrix} W_{11}^{KQ} \\ (w_{21}^{KQ})^\top \end{pmatrix} W^E x_{\tau,q}. \tag{7}$$

**ICL embedding AT for LSA-E models.** To approximate the real-world setting of LLM CAT, the theoretical ICL embedding AT also adopts ICL adversarial examples found in the embedding space to train LSA-E models. An ICL adversarial example in the embedding space is defined as below,

$$\mathcal{E}^{\text{adv}}(Z_\tau, \Delta_\tau^E) := \begin{pmatrix} W^E x_{\tau,1} + \delta_{\tau,1}^E & \cdots & W^E x_{\tau,N} + \delta_{\tau,N}^E & W^E x_{\tau,q} \\ y_{\tau,1} & \cdots & y_{\tau,N} & 0 \end{pmatrix} \in \mathbb{R}^{(d+1)\times(N+1)}, \tag{8}$$

where $Z_\tau$ is the ICL input for the task $\tau$ (see Eq. (5)), $W^E$ is the parameter of the embedding function $\mathcal{E}(\cdot)$ (see Eq. (6)), and $\Delta_\tau^E := \begin{pmatrix} \delta_{\tau,1}^E & \cdots & \delta_{\tau,N}^E \end{pmatrix} \in \mathbb{R}^{d\times N}$ denotes all adversarial perturbations added to the embeddings of the $N$ in-context training points. The prediction of the LSA-E model $f_{\text{LSAE},\theta}$ for the adversarial example $\mathcal{E}^{\text{adv}}(Z_\tau, \Delta_\tau^E)$ is then given by the following $\hat{y}_{q,\theta}^{\text{adv}}(Z_\tau, \Delta_\tau^E)$,

$$\hat{y}_{q,\theta}^{\text{adv}}(Z_\tau, \Delta_\tau^E) := \begin{pmatrix} (W_{21}^V)^\top & w_{22}^V \end{pmatrix} \frac{\mathcal{E}^{\text{adv}}(Z_\tau, \Delta_\tau^E)\mathcal{E}^{\text{adv}}(Z_\tau, \Delta_\tau^E)^\top}{N} \begin{pmatrix} W_{11}^{KQ} \\ (W_{21}^{KQ})^\top \end{pmatrix} W^E x_{\tau,q}. \tag{9}$$

With all these notations, the ICL embedding AT for an LSA-E model is eventually formalized as the following minimax optimization problem,

$$\min_\theta \mathcal{L}_{\text{LSAE}}^{\text{adv}}(\theta) := \min_\theta \left\{ \mathbb{E}_\tau \max_{\|\Delta_\tau^{E\top}\|_{2,\infty} \leq \epsilon} \frac{1}{2} |\hat{y}_{q,\theta}^{\text{adv}}(Z_\tau, \Delta_\tau^E) - y_{\tau,q}|^2 \right\}, \tag{10}$$

Where $\epsilon > 0$ is the embedding space adversarial perturbation radius and the expectation $\mathbb{E}_\tau$ is calculated over the randomness of $w_\tau, x_{\tau,1}, \ldots, x_{\tau,N}, x_{\tau,q}$. The restriction in the inner maximization in Eq. (10) ensures that each adversarial perturbation $\delta_{\tau,i}^E$ is confined within the ball-sphere $\|\delta_{\tau,i}^E\|_2 \leq \epsilon$.

**Robust generalization risk for LSA-E models.** We use the ICL suffix adversarial attack (Fu et al., 2025) to assess the robustness of ICL models. Nevertheless, we note that our experiments in Section 5 also consider attacks beyond suffix jailbreaking. Specifically, given an ICL input $Z_\tau$ with context length $N$, the ICL suffix adversarial attack adversarially perturbs the last $M$ ($M \leq N$) in-context training points of $Z_\tau$ as follows,

$$Z_{\tau,M}^{\text{adv}} := \begin{pmatrix} x_{\tau,1} & \cdots & x_{\tau,N-M} & x_{\tau,N-M+1} + \delta_{\tau,1}^O & \cdots & x_{\tau,N} + \delta_{\tau,M}^O & x_{\tau,q} \\ y_{\tau,1} & \cdots & y_{\tau,N-M} & y_{\tau,N-M+1} & \cdots & y_{\tau,N} & 0 \end{pmatrix}, \tag{11}$$

where $\delta_{\tau,i}^O \in \mathbb{R}^{d_0}$ is the adversarial perturbation added to the $i$-th ICL suffix point $x_{\tau,N-M+i}$. Although the ICL suffix adversarial example $Z_{\tau,M}^{\text{adv}}$ in Eq. (11) looks similar to the adversarial example defined in Eq. (8), the mechanisms behind them are very different: in Eq. (11), adversarial perturbations are directly added to in-context points, whereas in Eq. (8), perturbations are added to the embeddings of these in-context points. The robust generalization risk $\mathcal{R}_{\rho,M}(\theta)$ with perturbation $\rho$ and adversarial suffix length $M$ for an LSA-E model $f_{\text{LSAE},\theta}$ is then defined as below,

$$\mathcal{R}_{\rho,M}^{\text{adv}}(\theta) = \mathbb{E}_\tau \max_{\|\Delta_\tau^{O\top}\|_{2,\infty} \leq \rho} \frac{1}{2} |\hat{y}_{q,\theta}(Z_{\tau,M}^{\text{adv}}) - y_{\tau,q}|^2, \tag{12}$$

where $\hat{y}_{q,\theta}$ is the LSA-E prediction function in Eq. (7), $Z_{\tau,M}^{\mathrm{adv}}$ is the ICL suffix adversarial example in Eq. (11), $\Delta_\tau^O := \begin{pmatrix} \delta_1^O & \cdots & \delta_M^O \end{pmatrix} \in \mathbb{R}^{d_0 \times M}$ contains all perturbations added to the suffix of $Z_{\tau,M}^{\mathrm{adv}}$, and $\rho > 0$ is the adversarial perturbation radius for the suffix attack, which restricts each perturbation $\delta_i^O$ to the ball-sphere $\|\delta_i^O\|_2 \leq \rho$. A lower robust risk $\mathcal{R}_{\rho,M}^{\mathrm{adv}}(\theta)$ indicates stronger adversarial robustness of the model $f_{\mathrm{LSAE},\theta}$, and vice versa.

## 4.2 Bridging ICL Embedding AT and LLM Continuous AT

Before introducing our main theoretical results, here we explain why the established ICL embedding AT in Eq. (10) can be a good artifact for approximating real-world LLM CAT in Eq. (3) under the ICL theory by analyzing the similarities between them.

**Firstly, LSA-E models are very similar to real-world LLMs.** Ahn et al. (2024) empirically show that the linear self-attention module in LSA-E models share similar properties with those non-linear ones in LLMs and thus are useful for theoretically understanding LLMs. We further argue that **the embedding processes of LSA-E models and real-world LLMs are also very similar**. If we replace each token in an LLM prompt with its one-hot encoding vector defined over the token vocabulary space, the embedding process for each token in an LLM can be seen as a matrix multiplication between the LLM's embedding matrix and the corresponding token's one-hot encoding. This matrix multiplication-based LLM prompt embedding process is almost identical to the ICL input embedding process shown in Eq. (6), where input features are also linearly transformed by the LSA-E model's embedding matrix. Therefore, we believe the embedding spaces of both LSA-E models and LLMs are similar, as they are both obtained via linear transformation.

**Secondly, the training goals of ICL embedding AT and LLM CAT are very similar to each other.** The two AT problems both aim to enhance models' robustness by training them on sequential data where their embeddings are adversarially perturbed. The only difference is that in ICL embedding AT, the goal of adversarial perturbations is to reduce the utility of linear regression prediction made by LSA-E models, while in LLM CAT such a goal is to induce harmful content from LLMs.

**Finally, the adversarial robustness of LSA-E models is also very similar to the jailbreak robustness of LLMs.** Fu et al. (2025) has already illustrated why ICL suffix adversarial attacks are similar to real-world jailbreak attacks through theoretical and empirical justifications. Since we also leverage this ICL suffix adversarial attack to assess the robustness of LSA-E models, we believe that analyzing the robust generalization ability of LSA-E models will effectively help us understand how LLMs trained from CAT gain robustness against jailbreak attacks.

## 4.3 Robust Generalization Bound of ICL Embedding AT

We now start to establish a robust generalization bound for the LSA-E model trained via ICL embedding AT as formalized in Eq. (10). The derivation consists of three steps: (1) derive an upper bound $\tilde{\mathcal{L}}_{\mathrm{LSAE}}^{\mathrm{adv}}(\theta)$ for the original loss function $\mathcal{L}_{\mathrm{LSAE}}^{\mathrm{adv}}(\theta)$ in ICL embedding AT (see Eq.(10)) and formalize a surrogate AT problem that would minimize the upper bound $\tilde{\mathcal{L}}_{\mathrm{LSAE}}^{\mathrm{adv}}(\theta)$; (2) calculate the closed-form solution for the previously obtained surrogate ICL embedding AT problem; and (3) prove a robust generalization bound for the LSA-E model trained with the surrogate problem.

**Surrogate ICL embedding AT.** The surrogate problem for Eq. (10) is formalized as

$$\min_\theta \tilde{\mathcal{L}}_{\mathrm{LSAE}}^{\mathrm{adv}}(\theta) = \min_\theta \Big\{ \ell_1(\theta) + \ell_2(\theta) + \ell_3(\theta) + \ell_4(\theta) \Big\}, \tag{13}$$

where $\tilde{\mathcal{L}}_{\mathrm{LSAE}}^{\mathrm{adv}}(\theta) := \sum_{i=1}^4 \ell_i(\theta)$ is the surrogate objective function, and

$$\ell_1(\theta) := 2 \cdot \mathbb{E}_\tau \Big| \begin{pmatrix} (w_{21}^V)^\top & w_{22}^V \end{pmatrix} \frac{\mathcal{E}(Z_\tau)\mathcal{E}(Z_\tau)^\top}{N} \begin{pmatrix} W_{11}^{KQ} \\ (w_{21}^{KQ})^\top \end{pmatrix} W^E x_{\tau,q} - y_{\tau,q} \Big|^2,$$

$$\ell_2(\theta) := \frac{2\epsilon^2}{N} \cdot \|w_{21}^V\|_2^2 \cdot \mathbb{E}_\tau \Big[ \Big\| \begin{pmatrix} W^E x_{\tau,1} & \cdots & W^E x_{\tau,N} \\ y_{\tau,1} & \cdots & y_{\tau,N} \end{pmatrix}^\top \begin{pmatrix} W_{11}^{KQ} \\ (w_{21}^{KQ})^\top \end{pmatrix} W^E x_{\tau,q} \Big\|_2^2 \Big],$$

$$\ell_3(\theta) := \frac{2\epsilon^2}{N} \cdot \mathbb{E}_\tau \Big[ \Big\| \begin{pmatrix} (w_{21}^V)^\top & w_{22}^V \end{pmatrix} \begin{pmatrix} W^E x_{\tau,1} & \cdots & W^E x_{\tau,N} \\ y_{\tau,1} & \cdots & y_{\tau,N} \end{pmatrix} \Big\|_2^2 \Big] \cdot \mathbb{E}_\tau \Big[ \|W_{11}^{KQ} W^E x_{\tau,q}\|_2^2 \Big],$$

$$\ell_4(\theta) := 2\epsilon^4 \cdot \|w_{21}^V\|_2^2 \cdot \mathbb{E}_\tau\Big[\|W_{11}^{KQ}W^E x_{\tau,q}\|_2^2\Big].$$

The new objective function $\tilde{\mathcal{L}}_{\text{LSAE}}^{\text{adv}}(\theta)$ is a closed-form upper bound for the original ICL embedding AT loss function $\mathcal{L}_{\text{LSAE}}^{\text{adv}}(\theta)$, as shown in the following Lemma 1 (see Appendix B.2 for the proof).

**Lemma 1.** *For the objective function $\mathcal{L}_{\text{LSAE}}^{\text{adv}}(\theta)$ in ICL embedding AT (Eq. (10)) and the objective function $\tilde{\mathcal{L}}_{\text{LSAE}}^{\text{adv}}(\theta)$ in surrogate ICL embedding AT (Eq. (13)), we uniformly have $\mathcal{L}_{\text{LSAE}}^{\text{adv}}(\theta) \leq \tilde{\mathcal{L}}_{\text{LSAE}}^{\text{adv}}(\theta)$ for any $\theta := (W^E, W^{KQ}, W^V)$.*

The reason for studying the surrogate ICL embedding AT problem in Eq. (13) instead of the original problem is because the objective function $\mathcal{L}_{\text{LSAE}}^{\text{adv}}(\theta)$ in the original AT problem is difficult to tackle in a closed-form manner. With the new surrogate $\tilde{\mathcal{L}}_{\text{LSAE}}^{\text{adv}}(\theta)$, which is in closed-form, one can easily analyze the training dynamics of the LSA-E model trained from the surrogate problem. Further, since the surrogate objective function is an upper bound of the original one, minimizing it can also help to reduce the original AT loss and thus improve the robustness of the trained LSA-E model.

**Closed-form solution of the surrogate problem.** To solve the new surrogate AT problem in Eq. (13), we first make the below Assumption 1 on the initialization of the model parameter $\theta$.

**Assumption 1.** *Let $\zeta > 0$ be a parameter and $\Theta \in \mathbb{R}^{d \times d}$ be any matrix satisfying $\|\Theta\Theta^\top\|_F = 1$ and $\Theta\Lambda \neq 0_{d \times d}$. We assume that $W^V(0) = \begin{pmatrix} 0_{d \times d} & 0_{d \times 1} \\ 0_{1 \times d} & \zeta \end{pmatrix}$ and $W^{KQ}(0) = \begin{pmatrix} \zeta\Theta\Theta^\top & 0_{d \times 1} \\ 0_{1 \times d} & 0 \end{pmatrix}$.*

**Remark 1.** *Assumption 1 is widely adopted in the ICL theoretical analysis (Zhang et al., 2024; Frei & Vardi, 2025; Wu et al., 2024; Fu et al., 2025). The idea behind Assumption 1 is to (1) zero out terms $w_{22}^{KQ}$, $w_{12}^{KQ}$, $W_{11}^V$, and $w_{12}^V$ that do not contribute to the ICL prediction function $\hat{y}_{q,\theta}$ in Eq. (7) and (2) zero out terms $w_{21}^{KQ}$ and $w_{21}^V$ to ensure symmetric initialization.*

Under Assumption 1, the optimal solution for the LSA-E model trained from surrogate embedding ICL AT is calculated as the following Theorem 1 (see Appendix B.3 for the proof).

**Theorem 1** (Optimal solution of surrogate ICL embedding AT). *Suppose Assumption 1 holds and $f_{\text{LSAE},\theta}$ is trained from the surrogate embedding AT problem defined in Eq. (13) with continuous gradient flow. Then, the optimal model parameter $\theta_* := (W_*^E, W_*^{KQ}, W_*^V)$ should satisfies $w_{*,12}^{KQ} = w_{*,21}^{KQ} = w_{*,12}^V = w_{*,21}^V = 0_{d \times 1}$, $w_{*,22}^{KQ} = 0$, $W_{*,11}^V = 0_{d \times d}$, and*

$$w_{*,22}^V(W_*^E)^\top W_{*,11}^{KQ}W_*^E = (W_*^E)^\top \Big(W_*^E \Gamma_N \Lambda (W_*^E)^\top + \text{Tr}(\Lambda)\epsilon^2 I_d\Big)^{-1} W_*^E \Lambda,$$

*where $\Gamma_N := \big(\frac{N+1}{N}\Lambda + \frac{1}{N}\text{Tr}(\Lambda)I_{d_0}\big) \in \mathbb{R}^{d_0 \times d_0}$.*

Applying Theorem 1 to the LSA-E model prediction function in Eq. (7), the optimal prediction function $\hat{y}_{q,\theta*}(\cdot)$ given by the model $f_{\text{LSAE},\theta*}$ trained from surrogate ICL embedding AT is as follows,

$$\hat{y}_{q,\theta*}(Z_\tau) := \frac{1}{N}Y_\tau(X_\tau)^\top(W_*^E)^\top\Big(W_*^E \Gamma_N \Lambda(W_*^E)^\top + \text{Tr}(\Lambda)\epsilon^2 I_d\Big)^{-1} W_*^E \Lambda x_{\tau,q}. \quad (14)$$

**Robust generalization ability.** Finally, we leverage the robust generalization risk $\mathcal{R}_{\rho,M}^{\text{adv}}$ to assess the robustness of the optimal model $f_{\text{LSAE}}(\theta_*)$ trained from surrogate ICL embedding AT. Formally, we prove a robust generalization upper bound for the robust risk $\mathcal{R}_{\rho,M}^{\text{adv}}(\theta_*)$, as shown in the following Theorem 2 (see Appendix B.4 for the proof).

**Theorem 2** (Robust generalization upper bound). *Suppose Assumption 1 holds, $d \leq d_0$, and $\theta_*$ is the solution of the surrogate ICL embedding AT in Eq. (13) obtained from Theorem 1. We have*

$$\mathcal{R}_{\rho,M}^{\text{adv}}(\theta_*) \leq \mathcal{O}\Big(\frac{(1 + M\rho^2/N^2) \cdot \sum_{i=1}^d \sigma_i(W_*^E)^4}{\sigma_{\min}(W_*^E)^4 + \epsilon^4}\Big) + \mathcal{O}(1).$$

**Remark 2.** *Theorem 2 additionally requires that the embedding space dimension $d$ of the LSA-E model be no larger than the input in-context sample dimension $d_0$. Such a requirement means that the LSA-E models would implicitly compress the input data.*

### 4.4 IMPLICATIONS

So far, we have calculated the optimal ICL prediction function $\hat{y}_{q,\theta_*}$ obtained from surrogate embedding ICL AT in Eq. (14) and further prove a robust generalization bound for it in Theorem 2. We now start to investigate how these results can help to understand and improve CAT for LLMs.

**Embedding-space adversarial perturbations provably enhance input-space adversarial robustness.** The robust generalization bound in Theorem 2 clearly shows that the robust risk $\mathcal{R}^{\text{adv}}_{\rho,M}(\theta_*)$ of the trained LSA-E model, which is calculated based on adversarial ICL examples from the input space, has a negative correlation with the embedding-space adversarial perturbation radius $\epsilon$. A large perturbation radius $\epsilon$ in the embedding-space can help reduce the robust upper bound and thus improve the robustness of the trained LSA-E model against adversarial ICL examples. This explains the main mechanism behind ICL embedding AT and also that in CAT for real-world LLMs.

**The role of the embedding matrix in ICL robust generalization.** An interesting observation from Eq. (14) is that the optimal ICL prediction function $\hat{y}_{q,\theta*}$ depends only on the embedding matrix $W^E_*$ but not on remaining LSA-E model parameters $W^{KQ}_*$ and $W^V_*$. Therefore, a "good" embedding matrix $W^E_*$ is expected to provide strong robustness for the ICL model. Besides, from Theorem 2, we have two insights: (1) **if those large singular values of $W^E_*$ are not "too large"**, then the numerator of the first term in the robust upper bound can be reduced, which helps to reduce the overall bound; and (2) **if those small singular values of $W^E_*$ are not "too small"**, it helps to increase the denominator of the first term in the robust upper bound, which also helps to reduce the overall bound. Thus, we may expect an LSA-E model or even a real-world LLM to have an embedding matrix that has **"not too large nor too small singular values"** for strong model robustness.

**Improve CAT with an optimized embedding matrix.** Based on the previous analysis, we now propose ***Embedding-Regularized continuous AT (ER-CAT)***, a new AT method for LLMs designed by introducing an additional regularization term, defined as the variance of all singular values of the LLM embedding matrix, into the objective function of CAT in Eq. (4). Concretely, training an LLM $f_\theta$ via ER-CAT is formalized as solving the following optimization problem:

$$\min_\theta \mathcal{L}_{\text{ER-CAT}}(\theta, \alpha, \beta) := \min_\theta \Big\{ \underbrace{\mathcal{L}_{\text{CAT}}(\theta, \alpha)}_{\text{CAT loss in Eq. (4)}} + \beta \cdot \underbrace{\frac{\sum_{i=1}^d [\sigma_i(W^E) - \overline{\sigma}(W^E)]^2}{d}}_{\text{Embedding-Regularization Term}} \Big\}, \quad (15)$$

where $\beta > 0$ is the coefficient for the regularization term and $\overline{\sigma}(W^E) := \frac{1}{d}\sum_{i=1}^d \sigma_i(W^E)$ is the mean of all singular values of $W^E$. The reason for using the variance of singular values as a regularization term is that minimizing it can help to reduce too large singular values and increase too small singular values of the embedding matrix simultaneously, which, as explained before, helps to reduce the overall robust upper bound in Theorem 2. In addition, while in theory the singular values of $W^E$ in Eq. (15) are not differentiable, in practice their gradient calculation can be automatically handled by native PyTorch functions. This enables us to implement our ER-CAT method easily.

## 5 EMPIRICAL ANALYSIS OF ER-CONTINUOUS AT

In this section, we follow Eq. (15) to perform the theory-inspired ER-CAT on real-world LLMs, which can further help to empirically justify our proved robust generalization bound in Theorem 2.

### 5.1 EXPERIMENTAL SETUP

**Models.** We adopt six common pre-trained LLMs, which are: Vicuna-7B-v1.5 (Zheng et al., 2023), Mistral-7B-Instruct-v0.3 (Jiang et al., 2023), Llama-2-7B-Chat (Touvron et al., 2023), Llama-3-8B-Instruct (Grattafiori et al., 2024), Qwen2.5-7B-Instruct (Yang et al., 2024a), and Gemma-2B-it (Team et al., 2024). All models were downloaded from the Hugging Face model repository.

**Datasets.** During LLM AT, we follow Xhonneux et al. (2024) to use the training set of Harmbench (Mazeika et al., 2024) as the safety data and UltraChat 200K (Ding et al., 2023) as the utility data. During robustness evaluation, we follow Fu et al. (2025) to use a safety datasets that consists of the first 50 samples from the test set of Harmbench (Mazeika et al., 2024) and the first 50 samples

Table 1: ASR on different models and attacks. A low ASR indicates a strong model robustness.

| Model | Type | Avg@5 ASR (%) ↓ | | | | | |
|-------|------|-----|-------|-----|-------------|--------------|------|
| | | GCG | BEAST | GCQ | Zhu's AutoDAN | DeepInception | PAIR |
| Vicuna-7B | Original | 84.6 | 81.8 | 75.0 | 14.8 | 39.8 | 64.4 |
| | CAT | **12.6** | **14.4** | **4.6** | 0.4 | **5.8** | **7.8** |
| | ER-CAT (Ours) | 16.4 | 16.2 | 6.4 | 2.2 | 8.2 | 16.0 |
| Mistral-7B | Original | 74.6 | 65.8 | 69.8 | 43.0 | 49.8 | 56.0 |
| | CAT | **7.4** | **5.0** | **3.2** | 0.8 | 1.0 | **12.2** |
| | ER-CAT (Ours) | 7.6 | 6.6 | **3.2** | **0.4** | **0.8** | 16.2 |
| Llama-2-7B | Original | 41.0 | 18.2 | 5.6 | 7.2 | 30.8 | 24.2 |
| | CAT | 23.6 | 17.2 | 8.0 | 4.0 | 8.0 | 13.4 |
| | ER-CAT (Ours) | **15.6** | **10.4** | **1.2** | **0.4** | **2.0** | **4.6** |
| Llama-3.1-8B | Original | 11.2 | 20.8 | 6.0 | 5.4 | 37.6 | 41.8 |
| | CAT | 3.4 | **4.4** | **0.0** | **0.0** | **0.0** | 5.0 |
| | ER-CAT (Ours) | **2.4** | 9.0 | **0.0** | **0.0** | **0.0** | **3.8** |
| Qwen2.5-7B | Original | 71.2 | 71.6 | 59.8 | 15.6 | 58.5 | 46.2 |
| | CAT | 20.6 | 21.2 | 17.8 | **0.0** | **0.2** | 15.2 |
| | ER-CAT (Ours) | **16.8** | **15.4** | **6.6** | **0.0** | 1.4 | **13.6** |
| Gemma-2B | Original | 41.8 | 37.2 | 10.6 | 4.0 | 15.4 | 21.0 |
| | CAT | 18.0 | 11.2 | **2.6** | **0.2** | 0.2 | 4.4 |
| | ER-CAT (Ours) | **16.0** | **10.4** | 6.2 | 1.6 | **0.0** | **3.6** |

Table 2: LC-WinRate on different models. A high LC-WinRate indicates a strong model utility.

| Type | (Utility) LC-WinRate (%) ↑ | | | | | |
|------|-----------|------------|------------|--------------|------------|----------|
| | Vicuna-7B | Mistral-7B | Llama-2-7B | Llama-3.1-8B | Qwen2.5-7B | Gemma-2B |
| Original | 76.86 | 90.96 | 86.70 | 85.99 | 91.14 | 63.96 |
| CAT | 36.66 | 15.76 | 67.51 | 45.71 | 77.07 | 41.75 |
| ER-CAT (Ours) | 65.13 | 29.09 | 65.76 | 29.74 | 74.06 | 40.37 |

from AdvBench (Zou et al., 2023). For the utility analysis, we follow Dubois et al. (2024) to use the AlpacaEval dataset for calculating the LC-WinRate utility metric.

**Adversarial training.** We use AdamW to train each model via CAT in Eq. (4) or our ER-CAT in Eq. (15), where the embedding space perturbation radius $\epsilon$ is fixed to $0.05$. To improve the efficiency of tuning LLMs, LoRA (Hu et al., 2022) is applied to the embedding layer and all query and key projection matrices in attention layers. For the hyperparameter $\alpha$ of CAT, we follow Xhonneux et al. (2024) to set it as $0.5$. For the hyperparameters $\alpha$ and $\beta$ of our ER-CAT, we set them to $0.1$ and $0.2$, respectively. We also follow Xhonneux et al. (2024) to apply the loss cut-off technique to the objectives of both CAT and ER-CAT to avoid over-optimizing, but with less strict thresholds to help the trained LLMs better preserve utility. Please refer to Appendix C.1 for omitted details.

**Jailbreak attacks.** We use six different jailbreak attacks to assess the jailbreak robustness of LLMs. Among them, four attacks are token-level suffix attacks, which are: GCG (Zou et al., 2023), BEAST (Sadasivan et al., 2024), GCQ (Hayase et al., 2024), and Zhu's AutoDAN (Zhu et al., 2024). The remaining two attacks are prompt-level attacks, which are: DeepInception (Li et al., 2023) and PAIR (Chao et al., 2023). Please refer to Appendix C.2 for implementation details.

**Evaluations.** We evaluate the jailbreak robustness and the utility of trained LLMs. For the robustness evaluation, we report the **Avg@5 Attack Success Rate (Avg@5 ASR)** of jailbreak attacks. Specifically, each jailbreak prompt needs to repeatedly attack the targeted model for $5$ times. An LLM-based judger from Mazeika et al. (2024) is used to determine whether an attack is succeed or not. The final Avg@5 ASR is averaged on all repeated attack results. For the utility evaluation, we report the AlpacaEval's **Length-controlled WinRate (LC-WinRate)** (Dubois et al., 2024) of targeted models against a reference Davinci003 model, evaluated under the Llama-3-70B-Instruct model. An LC-WinRate of $50\%$ means that the output qualities of the two models are equal, and higher LC-WinRate means that the targeted model is better than the reference model.

## 5.2 RESULTS ANALYSIS

**Robustness&utility.** Avg@5 ASR and LC-WinRate on different models are reported in Table 1 and Table 2, respectively. We have two main observations. **Firstly, when maintaining the same**

Table 3: Time cost on different AT methods on different models.

| Method | Time Cost (s) | | | | | |
|---|---|---|---|---|---|---|
| | Vicuna-7B | Mistral-7B | Llama-2-7B | Llama-3.1-8B | Qwen2.5-7B | Gemma-2B |
| CAT | 987.81 | 933.00 | 934.16 | 904.59 | 801.42 | 335.99 |
| ER-CAT (Ours) | 1074.87 | 1052.12 | 1100.33 | 1094.90 | 1004.30 | 456.81 |

Table 4: LC-WinRate (%) and ASRs (%) of LLMs trained from ER-CAT under different embedding regularization coefficient $\beta$.

| Model | Utility or ASR | Embedding-Regularization coefficient $\beta$ in Eq. (15) | | | | | |
|---|---|---|---|---|---|---|---|
| | | 0.2 | 0.4 | 0.5 | 0.6 | 0.8 | 1.0 |
| Vicuna-7B | LC-WinRate | 59.04 | 57.19 | 65.13 | 70.30 | 68.26 | 59.73 |
| | GCG | 13.8 | 12.0 | 16.4 | 30.0 | 17.0 | 16.2 |
| | BEAST | 13.6 | 12.4 | 16.2 | 26.0 | 15.8 | 14.4 |
| Qwen2.5-7B | LC-WinRate | 74.60 | 74.52 | 74.06 | 65.93 | 68.67 | 72.15 |
| | GCG | 16.8 | 17.8 | 16.8 | 14.2 | 14.6 | 18.8 |
| | BEAST | 18.4 | 16.0 | 10.4 | 13.0 | 12.0 | 18.0 |

**level of jailbreak robustness, ER-CAT achieves better utility.** From the ASR results on Vicuna and Mistral, we find that our ER-CAT was beaten by CAT by no more than $4\%$ in most of the attack scenarios. However, Vicuna and Mistral trained from ER-CAT achieved a nearly two-times better LC-WinRate than those trained from CAT. **Secondly, when maintaining the same level of utility, ER-CAT achieves stronger jailbreak robustness.** For Llama-2 and Qwen2.5, ER-CAT reduces LC-WinRate by no more than $3\%$ when compared with that of CAT. However, ER-CAT helps Llama-2 reduce ASR on GCG and BEAST attacks by around $7\%$, and Qwen2.5 reduce ASR on GCQ by $11\%$. All these suggest that our ER-CAT can achieve a better robustness-utility tradeoff.

**Time cost.** As explained in Section 4.4, calculating the embedding-regularization term (see Eq. (15)) for ER-CAT can be efficiently implemented via native PyTorch functions. Here we empirically justify that ER-CAT does not add significant time overhead when compared with the original CAT. Specifically, we collect and present the time cost of performing CAT and our ER-CAT on different base models in Table 3, from which we find that ER-CAT only increases the time cost by 100 to 200 seconds. This suggests that the relative time overhead of ER-CAT is low.

**Ablation studies on the coefficient $\beta$ in ER-CAT.** In our main experiments, the embedding regularization coefficient $\beta$ in ER-CAT (see Eq. (15)) is fixed to 0.5. We now analyze how this coefficient $\beta$ affects the performance of models trained from ER-CAT. Specifically, we vary the coefficient $\beta$ within the range $[0, 1]$, train models via ER-CAT, and calculate their utility (*i.e.*, LC-WinRate) and robustness (*i.e.*, GCG and BEAST). Preliminary results are reported in Table 4, from which we surprisingly find that varying the coefficient $\beta$ does not change the utility or the robustness of the trained model too much. We deduce this is because we use the AdamW optimizer to perform the training, and the gradient normalization procedure in AdamW implicitly performs reweighting on the ER-CAT objective to mitigate the effect of tuning coefficients for different terms.

## 6 CONCLUSIONS

This paper aims to theoretically explain the mechanism behind CAT for LLMs, *i.e.*, why embedding space adversarial perturbations help LLMs learn to defend against jailbreak prompts from the token space. We first establish a new ICL embedding AT problem under ICL theory and show that this problem is a good theoretical artifact for approximating real-world LLM CAT. A robust generalization upper bound for this new ICL embedding AT is then proved, which shows a negative correlation with the embedding space perturbation radius. This clearly explains why CAT achieves empirical success. Our bound also suggests that the jailbreak robustness of an LLM is closely related to singular values of its embedding matrix. Thereby, we design a new ER-CAT approach for LLMs, with the goal of optimizing the LLM embedding matrix to be more robust. Experiments on real-world LLMs and jailbreak attacks suggest that our ER-CAT enjoys a better robustness-utility tradeoff.

ACKNOWLEDGMENTS

Di Wang and Shaopeng Fu are supported in part by the funding BAS/1/1689-01-01,RGC/3/7125-01-01, FCC/1/5940-20-05, FCC/1/5940-06-02, and King Abdullah University of Science and Technology (KAUST) – Center of Excellence for Generative AI, under award number 5940 and a gift from Google.

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

# A  LLMs USAGE IN THIS PAPER

LLMs were used only occasionally to help polish the writing (propose new words, grammar and spelling correction). All technical ideas, experimental designs, analyses, conclusions, writing were developed and carried out entirely by the authors. Authors have full responsibility for the final text.

# B  PROOFS

This section collects all proofs omitted from the main text. Without loss of generality, we assume that the order of differentiation and integration (or say expectation) is interchangeable.

## B.1  TECHNICAL LEMMAS

This section collects technical lemmas that will be repeatedly used in our proofs.

**Lemma B.1** (c.f. Lemma D.2 in Zhang et al. (2024)). *If $x \in \mathbb{R}^{d \times 1}$ is Gaussian random vector of $d$ dimension, mean zero and covariance matrix $\Lambda$, and $A \in \mathbb{R}^{d \times d}$ is a fixed matrix. Then*

$$\mathbb{E}[xx^\top A xx^\top] = \Lambda(A + A^\top)\Lambda + \mathrm{Tr}(A\Lambda)\Lambda.$$

**Lemma B.2** (c.f. Lemma A.2 in Fu et al. (2025)). *If $x \in \mathbb{R}^{d \times 1}$ is Gaussian random vector of $d$ dimension, mean zero and covariance matrix $\Lambda$, and $A \in \mathbb{R}^{d \times d}$ is a fixed matrix. Then*

$$\mathbb{E}[x^\top A x] = \mathrm{Tr}(A\Lambda).$$

**Lemma B.3** (c.f. Lemma A.3 in Fu et al. (2025)). *For any matrices $A \in \mathbb{R}^{n \times m}$ and $B \in \mathbb{R}^{m \times n}$,*

$$\mathrm{Tr}(AB) = \mathrm{Tr}(BA).$$

**Lemma B.4** (c.f. Lemma 1 in Wang et al. (1986)). *Let $A, B \in \mathbb{R}^{n \times n}$ be two symmetric matrices and $A$ is further positive semidefinite. Then*

$$\lambda_{\min}(B) \cdot \mathrm{Tr}(A) \leq \mathrm{Tr}(AB) \leq \lambda_{\max}(B) \cdot \mathrm{Tr}(A),$$

*where $\lambda_{\min}(B)$ and $\lambda_{\max}(B)$ are the minimal and maximal eigenvalues of $B$ respectively.*

**Lemma B.5** (Rayleigh Quotient Theorem; Also in part of Theorem 4.2.2 in Horn & Johnson (2012)). *Let $A \in \mathbb{R}^{n \times n}$ be a symmetric matrix. We have*

$$\lambda_{\max}(A) = \max_{x \in \mathbb{R}^n, x \neq 0_n} \frac{x^\top A x}{x^\top x} = \max_{x \in \mathbb{R}^n, \|x\|_2 = 1} x^\top A x,$$

$$\lambda_{\min}(A) = \min_{x \in \mathbb{R}^n, x \neq 0_n} \frac{x^\top A x}{x^\top x} = \min_{x \in \mathbb{R}^n, \|x\|_2 = 1} x^\top A x.$$

## B.2  PROOF OF LEMMA 1

*Proof of Lemma 1.* Denote that $X_\tau := (x_{\tau,1} \quad \cdots \quad x_{\tau,N}) \in \mathbb{R}^{d_0 \times N}$, $Y_\tau := (y_{\tau,1} \quad \cdots \quad y_{\tau,N}) \in \mathbb{R}^{1 \times N}$ and $\Delta_\tau^E := (\delta_{\tau,1}^E \quad \cdots \quad \delta_{\tau,N}^E) \in \mathbb{R}^{d \times N}$. Then, by applying the inequality that $|a + b|^2 \leq 2 \cdot (a^2 + b^2)$, the ICL embedding AT loss $\mathcal{L}_{\mathrm{LSAE}}^{\mathrm{adv}}(\theta)$ defined in Eq. (10), can be bounded as follows,

$$\mathcal{L}_{\mathrm{LSAE}}^{\mathrm{adv}}(\theta) := \mathbb{E}_\tau \max_{\|\Delta_\tau^{E^\top}\|_{2,\infty} \leq \epsilon} 2 \cdot |\hat{y}_{q,\theta}^{\mathrm{adv}}(Z_\tau, \Delta_\tau^E) - y_{\tau,q}|^2$$

$$\leq \mathbb{E}_\tau \max_{\|\Delta_\tau^{E^\top}\|_{2,\infty} \leq \epsilon} 2 \cdot \left|((w_{21}^V)^\top \quad w_{22}^V) \cdot \frac{\begin{pmatrix} W^E X_\tau & W^E x_{\tau,q} \\ Y_\tau & 0 \end{pmatrix} \cdot \begin{pmatrix} W^E X_\tau & W^E x_{\tau,q} \\ Y_\tau & 0 \end{pmatrix}^\top}{N} \cdot \begin{pmatrix} W_{11}^{KQ} \\ (w_{21}^{KQ})^\top \end{pmatrix} \cdot W^E x_{\tau,q} - y_{\tau,q}\right|^2$$

$$+ \mathbb{E}_\tau \max_{\|\Delta_\tau^{E^\top}\|_{2,\infty} \leq \epsilon} \frac{2}{N^2} \cdot \left|((w_{21}^V)^\top \quad w_{22}^V) \cdot \begin{pmatrix} \Delta_\tau^E & 0_{d \times 1} \\ 0_{1 \times N} & 0 \end{pmatrix} \cdot \begin{pmatrix} W^E X_\tau & W^E x_{\tau,q} \\ Y_\tau & 0 \end{pmatrix}^\top \cdot \begin{pmatrix} W_{11}^{KQ} \\ (w_{21}^{KQ})^\top \end{pmatrix} \cdot W^E x_{\tau,q}\right|^2$$

$$+ \mathbb{E}_\tau \max_{\|\Delta_\tau^{E^\top}\|_{2,\infty} \leq \epsilon} \frac{2}{N^2} \cdot \left|((w_{21}^V)^\top \quad w_{22}^V) \cdot \begin{pmatrix} W^E X_\tau & W^E x_{\tau,q} \\ Y_\tau & 0 \end{pmatrix} \cdot \begin{pmatrix} \Delta_\tau^E & 0_{d \times 1} \\ 0_{1 \times N} & 0 \end{pmatrix}^\top \cdot \begin{pmatrix} W_{11}^{KQ} \\ (w_{21}^{KQ})^\top \end{pmatrix} \cdot W^E x_{\tau,q}\right|^2$$

$$+ \mathbb{E}_{\tau} \max_{\|\Delta_\tau^{E\top}\|_{2,\infty} \le \epsilon} \frac{2}{N^2} \left| \begin{pmatrix} (w_{21}^V)^\top & w_{22}^V \end{pmatrix} \cdot \begin{pmatrix} \Delta_\tau^E & 0_{d\times 1} \\ 0_{1\times N} & 0 \end{pmatrix} \cdot \begin{pmatrix} \Delta_\tau^E & 0_{d\times 1} \\ 0_{1\times N} & 0 \end{pmatrix}^\top \cdot \begin{pmatrix} W_{11}^{KQ} \\ (w_{21}^{KQ})^\top \end{pmatrix} \cdot W^E x_{\tau,q} \right|^2$$

$$\le 2 \cdot \mathbb{E}_{\tau} \left| \begin{pmatrix} (w_{21}^V)^\top & w_{22}^V \end{pmatrix} \cdot \frac{\mathcal{E}(Z_\tau)\mathcal{E}(Z_\tau)^\top}{N} \cdot \begin{pmatrix} W_{11}^{KQ} \\ (w_{21}^{KQ})^\top \end{pmatrix} \cdot W^E x_{\tau,q} - y_{\tau,q} \right|^2$$

$$+ \underbrace{\mathbb{E}_{\tau} \max_{\|\Delta_\tau^{E\top}\|_{2,\infty} \le \epsilon} \frac{2}{N^2} \cdot \left| (w_{21}^V)^\top \cdot \Delta_\tau^E \cdot \begin{pmatrix} W^E X_\tau \\ Y_\tau \end{pmatrix}^\top \cdot \begin{pmatrix} W_{11}^{KQ} \\ (w_{21}^{KQ})^\top \end{pmatrix} \cdot W^E x_{\tau,q} \right|^2}_{:=A_1(\theta)}$$

$$+ \underbrace{\mathbb{E}_{\tau} \max_{\|\Delta_\tau^{E\top}\|_{2,\infty} \le \epsilon} \frac{2}{N^2} \cdot \left| \begin{pmatrix} (w_{21}^V)^\top & w_{22}^V \end{pmatrix} \cdot \begin{pmatrix} W^E X_\tau \\ Y_\tau \end{pmatrix} \cdot (\Delta_\tau^E)^\top \cdot W_{11}^{KQ} \cdot W^E x_{\tau,q} \right|^2}_{:=A_2(\theta)}$$

$$+ \underbrace{\mathbb{E}_{\tau} \max_{\|\Delta_\tau^{E\top}\|_{2,\infty} \le \epsilon} \frac{2}{N^2} \cdot \left| (w_{21}^V)^\top \cdot \Delta_\tau^E (\Delta_\tau^E)^\top \cdot W_{11}^{KQ} \cdot W^E x_{\tau,q} \right|^2}_{:=A_3(\theta)}. \tag{B.1}$$

For $A_1(\theta)$ in Eq. (B.1), we have

$$A_1(\theta) := \mathbb{E}_{\tau} \max_{\|\Delta_\tau^{E\top}\|_{2,\infty} \le \epsilon} \frac{2}{N^2} \cdot \left| (w_{21}^V)^\top \cdot \Delta_\tau^E \cdot \begin{pmatrix} W^E X_\tau \\ Y_\tau \end{pmatrix}^\top \cdot \begin{pmatrix} W_{11}^{KQ} \\ (w_{21}^{KQ})^\top \end{pmatrix} \cdot W^E x_{\tau,q} \right|^2$$

$$\le \mathbb{E}_{\tau} \max_{\|\Delta_\tau^{E\top}\|_{2,\infty} \le \epsilon} \frac{2}{N^2} \cdot \|(w_{21}^V)^\top \Delta_\tau^E\|_2^2 \cdot \left\| \begin{pmatrix} W^E X_\tau \\ Y_\tau \end{pmatrix}^\top \begin{pmatrix} W_{11}^{KQ} \\ (w_{21}^{KQ})^\top \end{pmatrix} W^E x_{\tau,q} \right\|_2^2$$

$$\le \mathbb{E}_{\tau} \frac{2}{N^2} \cdot N \|w_{21}^V\|_2^2 \epsilon^2 \cdot \left\| \begin{pmatrix} W^E X_\tau \\ Y_\tau \end{pmatrix}^\top \begin{pmatrix} W_{11}^{KQ} \\ (w_{21}^{KQ})^\top \end{pmatrix} W^E x_{\tau,q} \right\|_2^2$$

$$= \frac{2\epsilon^2}{N} \cdot \|w_{21}^V\|_2^2 \cdot \mathbb{E}_{\tau} \left[ \left\| \begin{pmatrix} W^E X_\tau \\ Y_\tau \end{pmatrix}^\top \begin{pmatrix} W_{11}^{KQ} \\ (w_{21}^{KQ})^\top \end{pmatrix} W^E x_{\tau,q} \right\|_2^2 \right]. \tag{B.2}$$

For $A_2(\theta)$ in Eq. (B.1), we have

$$A_2(\theta) := \mathbb{E}_{\tau} \max_{\|\Delta_\tau^{E\top}\|_{2,\infty} \le \epsilon} \frac{2}{N^2} \cdot \left| \begin{pmatrix} (w_{21}^V)^\top & w_{22}^V \end{pmatrix} \cdot \begin{pmatrix} W^E X_\tau \\ Y_\tau \end{pmatrix} \cdot (\Delta_\tau^E)^\top \cdot W_{11}^{KQ} \cdot W^E x_{\tau,q} \right|^2$$

$$\le \mathbb{E}_{\tau} \max_{\|\Delta_\tau^{E\top}\|_{2,\infty} \le \epsilon} \frac{2}{N^2} \cdot \left\| \begin{pmatrix} (w_{21}^V)^\top & w_{22}^V \end{pmatrix} \begin{pmatrix} W^E X_\tau \\ Y_\tau \end{pmatrix} \right\|_2^2 \cdot \|(\Delta_\tau^E)^\top W_{11}^{KQ} W^E x_{\tau,q}\|_2^2$$

$$\le \mathbb{E}_{\tau} \frac{2}{N^2} \cdot \left\| \begin{pmatrix} (w_{21}^V)^\top & w_{22}^V \end{pmatrix} \begin{pmatrix} W^E X_\tau \\ Y_\tau \end{pmatrix} \right\|_2^2 \cdot N\epsilon^2 \|W_{11}^{KQ} W^E x_{\tau,q}\|_2^2$$

$$= \frac{2\epsilon^2}{N} \cdot \mathbb{E}_{\tau} \left[ \left\| \begin{pmatrix} (w_{21}^V)^\top & w_{22}^V \end{pmatrix} \begin{pmatrix} W^E X_\tau \\ Y_\tau \end{pmatrix} \right\|_2^2 \right] \cdot \mathbb{E}_{\tau} \left[ \|W_{11}^{KQ} W^E x_{\tau,q}\|_2^2 \right]. \tag{B.3}$$

For $A_3(\theta)$ in Eq. (B.1), we have

$$A_3(\theta) := \mathbb{E}_{\tau} \max_{\|\Delta_\tau^{E\top}\|_{2,\infty} \le \epsilon} \frac{2}{N^2} \cdot \left| (w_{21}^V)^\top \cdot \Delta_\tau^E (\Delta_\tau^E)^\top \cdot W_{11}^{KQ} \cdot W^E x_{\tau,q} \right|^2$$

$$\le \mathbb{E}_{\tau} \max_{\|\Delta_\tau^{E\top}\|_{2,\infty} \le \epsilon} \frac{2}{N^2} \cdot \|(w_{21}^V)^\top \Delta_\tau^E\|_2^2 \cdot \|(\Delta_\tau^E)^\top W_{11}^{KQ} W^E x_{\tau,q}\|_2^2$$

$$\le \frac{2}{N^2} \cdot \mathbb{E}_{\tau} \left[ \max_{\|\Delta_\tau^{E\top}\|_{2,\infty} \le \epsilon} \|(w_{21}^V)^\top \Delta_\tau^E\|_2^2 \cdot \max_{\|\Delta_\tau^{E\top}\|_{2,\infty} \le \epsilon} \|(\Delta_\tau^E)^\top W_{11}^{KQ} W^E x_{\tau,q}\|_2^2 \right]$$

$$\le \frac{2}{N^2} \cdot \mathbb{E}_{\tau} \left[ N \|(w_{21}^V)\|_2^2 \epsilon^2 \cdot N\epsilon^2 \|W_{11}^{KQ} W^E x_{\tau,q}\|_2^2 \right]$$

$$= 2\epsilon^4 \cdot \mathbb{E}_{\tau} \|(w_{21}^V)\|_2^2 \cdot \mathbb{E}_{\tau} \left[ \|W_{11}^{KQ} W^E x_{\tau,q}\|_2^2 \right] \tag{B.4}$$

Inserting Eqs.(B.2), (B.3) and (B.4) into Eq.(B.1, we eventually have that

$$
\begin{aligned}
&\mathcal{L}_{\mathrm{LSAE}}^{\mathrm{adv}}(\theta) \\
&\leq 2 \cdot \mathbb{E}_{\tau}\left|\left((w_{21}^V)^\top \quad w_{22}^V\right) \cdot \frac{\mathcal{E}(Z_\tau)\mathcal{E}(Z_\tau)^\top}{N} \cdot \begin{pmatrix} W_{11}^{KQ} \\ (w_{21}^{KQ})^\top \end{pmatrix} \cdot W^E x_{\tau,q} - y_{\tau,q}\right|^2 \\
&\quad + \frac{2\epsilon^2}{N} \cdot \|w_{21}^V\|_2^2 \cdot \mathbb{E}_{\tau}\left[\left\| \begin{pmatrix} W^E X_\tau \\ Y_\tau \end{pmatrix}^\top \begin{pmatrix} W_{11}^{KQ} \\ (w_{21}^{KQ})^\top \end{pmatrix} W^E x_{\tau,q}\right\|_2^2\right] \\
&\quad + \frac{2\epsilon^2}{N} \cdot \mathbb{E}_{\tau}\left[\left\| \left((w_{21}^V)^\top \quad w_{22}^V\right) \begin{pmatrix} W^E X_\tau \\ Y_\tau \end{pmatrix}\right\|_2^2\right] \cdot \mathbb{E}_{\tau}\left[\|W_{11}^{KQ} W^E x_{\tau,q}\|_2^2\right] \\
&\quad + 2\epsilon^4 \cdot \mathbb{E}_{\tau}\|(w_{21}^V)\|_2^2 \cdot \mathbb{E}_{\tau}\left[\|W_{11}^{KQ} W^E x_{\tau,q}\|_2^2\right] = \tilde{\mathcal{L}}_{\mathrm{LSAE}}^{\mathrm{adv}}(\theta),
\end{aligned}
$$

which completes the proof. $\qquad\square$

### B.3 PROOF OF THEOREM 1

The proof idea of Theorem 1 is similar to that in Zhang et al. (2024) and Fu et al. (2025). Specifically, we first show that when training the LSA-E model $f_{\mathrm{LSAE}}^{\mathrm{adv}}(\theta)$ via continuous gradient flow, $w_{21}^{KQ}$ and $w_{21}^V$ stay zero during the surrogate ICL embedding AT (Lemma B.6), which can help us further simplify the surrogate objective function $\tilde{\mathcal{L}}_{\mathrm{LSAE}}^{\mathrm{adv}}(\theta)$ (Lemma B.7). The global minimizer for the surrogate ICL embedding AT problem is then derived based on the simplified surrogate objective function (Lemma B.8).

We start by stating and proving Lemma B.6.

**Lemma B.6.** *Suppose Assumption 1 holds and the LSAE model $f_{\mathrm{LSAE}}^{\mathrm{adv}}(\theta)$ is trained via minimizing the surrogate objective function $\tilde{\mathcal{L}}_{\mathrm{LSAE}}^{\mathrm{adv}}(\theta)$ in Eq. (13) with continuous gradient flow. Then, for any continuous training time $t \geq 0$, we uniformly have that $w_{21}^{KQ}(t) = w_{21}^V(t) = 0_{d\times 1}$.*

*Proof.* Under Assumption 1, we already have that $w_{21}^{KQ}(0) = w_{21}^V(0) = 0_{d\times 1}$ at the initial training time $t = 0$. As a result, to prove Lemma B.6, we only need to further show that gradients for parameters $w_{21}^{KQ}(t)$ and $w_{21}^V(t)$, which are given by continuous gradient flows, stay zero during the overall surrogate ICL embedding AT. Formally, we need to prove that for any $t \geq 0$,

$$
\begin{aligned}
\partial_t w_{21}^{KQ}(t) &:= -\partial_{w_{21}^{KQ}} \tilde{\mathcal{L}}_{\mathrm{LSAE}}^{\mathrm{adv}}(\theta) = 0_{1\times d}, \\
\partial_t w_{21}^V(t) &:= -\partial_{w_{21}^V} \tilde{\mathcal{L}}_{\mathrm{LSAE}}^{\mathrm{adv}}(\theta) = 0_{1\times d}.
\end{aligned}
$$

To this end, we adopt the decomposition of $\tilde{\mathcal{L}}_{\mathrm{LSAE}}^{\mathrm{adv}}(\theta)$ in Eq. (13) here as follows,

$$
\tilde{\mathcal{L}}_{\mathrm{LSAE}}^{\mathrm{adv}}(\theta) := \ell_1(\theta) + \ell_2(\theta) + \ell_3(\theta) + \ell_4(\theta),
$$

where

$$
\ell_1(\theta) := 2 \cdot \mathbb{E}_{\tau}\left|\left((w_{21}^V)^\top \quad w_{22}^V\right) \cdot \frac{\mathcal{E}(Z_\tau)\mathcal{E}(Z_\tau)^\top}{N} \cdot \begin{pmatrix} W_{11}^{KQ} \\ (w_{21}^{KQ})^\top \end{pmatrix} \cdot W^E x_{\tau,q} - y_{\tau,q}\right|^2,
$$

$$
\ell_2(\theta) := \frac{2\epsilon^2}{N} \cdot \|w_{21}^V\|_2^2 \cdot \mathbb{E}_{\tau}\left[\left\| \begin{pmatrix} W^E X_\tau \\ Y_\tau \end{pmatrix}^\top \begin{pmatrix} W_{11}^{KQ} \\ (w_{21}^{KQ})^\top \end{pmatrix} W^E x_{\tau,q}\right\|_2^2\right],
$$

$$
\ell_3(\theta) := \frac{2\epsilon^2}{N} \cdot \mathbb{E}_{\tau}\left[\left\| \left((w_{21}^V)^\top \quad w_{22}^V\right) \begin{pmatrix} W^E X_\tau \\ Y_\tau \end{pmatrix}\right\|_2^2\right] \cdot \mathbb{E}_{\tau}\left[\|W_{11}^{KQ} W^E x_{\tau,q}\|_2^2\right],
$$

$$
\ell_4(\theta) := 2\epsilon^4 \cdot \mathbb{E}_{\tau}\|(w_{21}^V)\|_2^2 \cdot \mathbb{E}_{\tau}\left[\|W_{11}^{KQ} W^E x_{\tau,q}\|_2^2\right],
$$

and $X_\tau := (x_{\tau,1} \ \cdots \ x_{\tau,N}) \in \mathbb{R}^{d\times N}$, $Y_\tau := (y_{\tau,1} \ \cdots \ y_{\tau,N}) \in \mathbb{R}^{1\times N}$. Then, we will show that when $w_{21}^{KQ} = w_{21}^V = 0_{d\times 1}$, one always has $\partial_{w_{21}^{KQ}}\ell_i(\theta) = \partial_{w_{21}^V}\ell_i(\theta) = 0_{1\times d}$ for every $i \in [4]$, which thus automatically demonstrates that $\partial_{w_{21}^{KQ}}\tilde{\mathcal{L}}_{\mathrm{LSAE}}^{\mathrm{adv}}(\theta) = \partial_{w_{21}^V}\tilde{\mathcal{L}}_{\mathrm{LSAE}}^{\mathrm{adv}}(\theta) = 0_{1\times d}$ for any continuous training time $t \geq 0$ under Assumption 1.

**Step 1: Show that $w_{21}^{KQ} = w_{21}^V = 0_{d\times 1}$ indicates $\partial_{w_{21}^{KQ}}\ell_1(\theta) = \partial_{w_{21}^V}\ell_1(\theta) = 0_{1\times d}$.** For the $i$-th entry $w_{21,i}^{KQ}$ of $w_{21}^{KQ}$, we have that

$$
\begin{aligned}
&\partial_{w_{21,i}^{KQ}}\ell_1(\theta)\Big|_{w_{21}^{KQ}=w_{21}^V=0_{d\times 1}} \\
&= 4\cdot\mathop{\mathbb{E}}_\tau\Big[\Big(\big((w_{21}^V)^\top \quad w_{22}^V\big)\cdot\frac{\mathcal{E}(Z_\tau)\mathcal{E}(Z_\tau)^\top}{N}\cdot\begin{pmatrix}W_{11}^{KQ}\\(w_{21}^{KQ})^\top\end{pmatrix}\cdot W^E x_{\tau,q}-y_{\tau,q}\Big) \\
&\qquad\quad \cdot\big((w_{21}^V)^\top \quad w_{22}^V\big)\cdot\frac{\mathcal{E}(Z_\tau)\mathcal{E}(Z_\tau)^\top}{N}\cdot\begin{pmatrix}0_{d\times d}\\e_i^\top\end{pmatrix}\cdot W^E x_{\tau,q}\Big]_{w_{21}^{KQ}=w_{21}^V=0_{d\times 1}} \\
&= 4\cdot\mathop{\mathbb{E}}_\tau\Big[\Big(\big(0_{1\times d} \quad w_{22}^V\big)\cdot\frac{\mathcal{E}(Z_\tau)\mathcal{E}(Z_\tau)^\top}{N}\cdot\begin{pmatrix}W_{11}^{KQ}\\0_{1\times d}\end{pmatrix}\cdot W^E x_{\tau,q}-y_{\tau,q}\Big) \\
&\qquad\quad \cdot\big(0_{1\times d} \quad w_{22}^V\big)\cdot\frac{\mathcal{E}(Z_\tau)\mathcal{E}(Z_\tau)^\top}{N}\cdot\begin{pmatrix}0_{d\times d}\\e_i^\top\end{pmatrix}\cdot W^E x_{\tau,q}\Big] \\
&= 4\cdot\mathop{\mathbb{E}}_\tau\Big[\Big(\frac{1}{N}\cdot w_{22}^V\cdot(Y_\tau \quad 0)\cdot\big(W^E X_\tau \quad W^E x_{\tau,q}\big)^\top\cdot W_{11}^{KQ}\cdot W^E x_{\tau,q}-y_{\tau,q}\Big) \\
&\qquad\quad \cdot\Big(\frac{1}{N}\cdot w_{22}^V\cdot(Y_\tau \quad 0)\cdot(Y_\tau \quad 0)^\top\cdot e_i^\top\cdot W^E x_{\tau,q}\Big)\Big] \\
&= 4\cdot\mathop{\mathbb{E}}_\tau\Big[\Big(\frac{1}{N}\cdot w_{22}^V\cdot w_\tau^\top\cdot X_\tau\cdot(W^E X_\tau)^\top\cdot W_{11}^{KQ}\cdot W^E x_{\tau,q}-w_\tau^\top\cdot x_{\tau,q}\Big) \\
&\qquad\quad \cdot\Big(\frac{1}{N}\cdot w_{22}^V\cdot w_\tau^\top\cdot X_\tau X_\tau^\top\cdot w_\tau\cdot e_i^\top\cdot W^E x_{\tau,q}\Big)\Big],
\end{aligned}
\tag{B.5}
$$

where $e_i\in\mathbb{R}^{d\times 1}$ denotes an elementary vector that its $i$-th entry is 1 and all remaining entries are 0. We then have two observations for Eq. (B.5): (1) for the first multiplication term in Eq. (B.5), each of its summarization term contains exactly one element from the task parameter $w_\tau$; and (2) for the second multiplication term in Eq. (B.5), each of its summarization term contains exactly two elements from $w_\tau$. Based on these observations and the independency of $w_\tau$ with respect to $X_\tau$ and $x_{\tau,q}$, Eq. (B.5) can further be re-organized as follows,

$$
\partial_{w_{21,i}^{KQ}}\ell_1(\theta)\Big|_{w_{21}^{KQ}=w_{21}^V=0_{d\times 1}} = \sum_{k=1}^d\sum_{j=1}^d\sum_{l=1}^d\mathop{\mathbb{E}}_\tau\Big[B_{j,k,l}(X_\tau,x_{\tau,q},\theta)\Big]\cdot\mathop{\mathbb{E}}_\tau\Big[w_{\tau,k}\cdot w_{\tau,j}\cdot w_{\tau,l}\Big],
\tag{B.6}
$$

where $\mathbb{E}_\tau[B_{j,k,l}(X_\tau,x_{\tau,q},\theta)]$ is the coefficient for the term $\mathbb{E}_\tau[w_{\tau,k}\cdot w_{\tau,j}\cdot w_{\tau,l}]$ and depends only on $X_\tau$, $x_{\tau,q}$ and $\theta$. Recall that $w_\tau\sim\mathcal{N}(0,I_{d_0})$, which means $\mathbb{E}_\tau[w_{\tau,k}\cdot w_{\tau,j}\cdot w_{\tau,l}]=0$ holds for any $k,j,l\in[d]$. Combine this result with Eq. (B.6), we thus have

$$
\partial_{w_{21,i}^{KQ}}\ell_1(\theta)\Big|_{w_{21}^{KQ}=w_{21}^V=0_{d\times 1}} = 0,\quad\forall i\in[d],
$$

which indicates $\partial_{w_{21}^{KQ}}\ell_1(\theta)\Big|_{w_{21}^{KQ}=w_{21}^V=0_{d\times 1}} = 0_{1\times d}$.

Besides, for the $i$-th element $w_{21,i}^V$ of $w_{21}^V$, we similarly have that

$$
\begin{aligned}
&\partial_{w_{21,i}^V}\ell_1(\theta)\Big|_{w_{21}^{KQ}=w_{21}^V=0_{d\times 1}} \\
&= 4\cdot\mathop{\mathbb{E}}_\tau\Big[\Big(\big((w_{21}^V)^\top \quad w_{22}^V\big)\cdot\frac{\mathcal{E}(Z_\tau)\mathcal{E}(Z_\tau)^\top}{N}\cdot\begin{pmatrix}W_{11}^{KQ}\\(w_{21}^{KQ})^\top\end{pmatrix}\cdot W^E x_{\tau,q}-y_{\tau,q}\Big) \\
&\qquad\quad \cdot\Big(\big(e_i^\top \quad 0\big)\cdot\frac{\mathcal{E}(Z_\tau)\mathcal{E}(Z_\tau)^\top}{N}\cdot\begin{pmatrix}W_{11}^{KQ}\\(w_{21}^{KQ})^\top\end{pmatrix}\cdot W^E x_{\tau,q}\Big)\Big]_{w_{21}^{KQ}=w_{21}^V=0_{d\times 1}} \\
&= 4\cdot\mathop{\mathbb{E}}_\tau\Big[\Big(\frac{1}{N}\cdot w_{22}^V\cdot w_\tau^\top\cdot X_\tau\cdot(W^E X_\tau)^\top\cdot W_{11}^{KQ}\cdot W^E x_{\tau,q}-w_\tau^\top\cdot x_{\tau,q}\Big) \\
&\qquad\quad \cdot\Big(\frac{1}{N}\cdot e_i^\top\cdot\big(W^E X_\tau \quad W^E x_{\tau,q}\big)\cdot\big(W^E X_\tau \quad W^E x_{\tau,q}\big)^\top\cdot W_{11}^{KQ}\cdot W^E x_{\tau,q}\Big)\Big].
\end{aligned}
\tag{B.7}
$$

From Eq. (B.7) we notice that: (1) each summarization term in the first multiplication term of Eq. (B.7) contains exactly one element from $w_\tau$, and (2) the second multiplication term of Eq. (B.7) does not contain any element from $w_\tau$. Thus, Eq. (B.7) can be re-organized as below,

$$\partial_{w_{21,i}^V}\ell_1(\theta)\Big|_{w_{21}^{KQ}=w_{21}^V=0_{d\times1}} = \sum_{k=1}^{d}\mathbb{E}_\tau\Big[B_j'(X_\tau,x_{\tau,q},\theta)\Big]\cdot\mathbb{E}_\tau[w_{\tau,k}], \tag{B.8}$$

where $\mathbb{E}_\tau[B_j'(X_\tau,x_{\tau,q},\theta)]$ is the coefficient for the term $\mathbb{E}_\tau[w_{\tau,k}]$ and depends only on $X_\tau$, $x_{\tau,q}$ and $\theta$. As a result, by again applying $w_\tau\sim\mathcal{N}(0,I_{d_0})$, we have $\mathbb{E}_\tau[w_{\tau,k}]=0$ for any $k\in[d]$, which means

$$\partial_{w_{21,i}^V}\ell_1(\theta)\Big|_{w_{21}^{KQ}=w_{21}^V=0_{d\times1}} = 0 \quad \forall i\in[d],$$

and thus indicates $\partial_{w_{21}^V}\ell_1(\theta)\Big|_{w_{21}^{KQ}=w_{21}^V=0_{d\times1}} = 0_{1\times d}$

**Step 2: Show that** $w_{21}^{KQ} = w_{21}^V = 0_{d\times1}$ **indicates** $\partial_{w_{21}^{KQ}}\ell_2(\theta) = \partial_{w_{21}^V}\ell_2(\theta) = 0_{1\times d}$. For $\partial_{w_{21}^{KQ}}\ell_2(\theta)$, we have

$$\partial_{w_{21}^{KQ}}\ell_2(\theta)\Big|_{w_{21}^{KQ}=w_{21}^V=0_{d\times1}} = \left[\frac{2\epsilon^2}{N}\cdot\|w_{21}^V\|_2^2\cdot\partial_{w_{21}^{KQ}}\left(\mathbb{E}_\tau\|\begin{pmatrix}W^EX_\tau\\Y_\tau\end{pmatrix}^\top\begin{pmatrix}W_{11}^{KQ}\\(w_{21}^{KQ})^\top\end{pmatrix}W^Ex_{\tau,q}\|_2^2\right)\right]_{w_{21}^{KQ}=w_{21}^V=0_{d\times1}}$$

$$= \frac{2\epsilon^2}{N}\cdot\|0_{d\times1}\|_2^2\cdot\partial_{w_{21}^{KQ}}\left(\mathbb{E}_\tau\|\begin{pmatrix}W^EX_\tau\\Y_\tau\end{pmatrix}^\top\begin{pmatrix}W_{11}^{KQ}\\(w_{21}^{KQ})^\top\end{pmatrix}W^Ex_{\tau,q}\|_2^2\right)_{w_{21}^{KQ}=0_{d\times1}}$$

$$= 0_{1\times d}.$$

For $\partial_{w_{21}^V}\ell_2(\theta)$, we have

$$\partial_{w_{21}^V}\ell_2(\theta)\Big|_{w_{21}^{KQ}=w_{21}^V=0_{d\times1}} = \left[\frac{2\epsilon^2}{N}\cdot2\cdot(w_{21}^V)^\top\cdot\mathbb{E}_\tau\|\begin{pmatrix}W^EX_\tau\\Y_\tau\end{pmatrix}^\top\begin{pmatrix}W_{11}^{KQ}\\(w_{21}^{KQ})^\top\end{pmatrix}W^Ex_{\tau,q}\|_2^2\right]_{w_{21}^{KQ}=w_{21}^V=0_{d\times1}}$$

$$= \left[\frac{2\epsilon^2}{N}\cdot2\cdot(0_{d\times1})^\top\cdot\mathbb{E}_\tau\|\begin{pmatrix}W^EX_\tau\\Y_\tau\end{pmatrix}^\top\begin{pmatrix}W_{11}^{KQ}\\(w_{21}^{KQ})^\top\end{pmatrix}W^Ex_{\tau,q}\|_2^2\right]_{w_{21}^{KQ}=0_{d\times1}}$$

$$= 0_{1\times d}.$$

**Step 3: Show that** $w_{21}^{KQ} = w_{21}^V = 0_{d\times1}$ **indicates** $\partial_{w_{21}^{KQ}}\ell_3(\theta) = \partial_{w_{21}^V}\ell_3(\theta) = 0_{1\times d}$. For $w_{21}^{KQ}$, since it does not exist in $\ell_3(\theta)$, thus we always have $\partial_{w_{21}^{KQ}}\ell_3(\theta)=0$.

Besides, for the $i$-th element $w_{21,i}^V$ of $w_{21}^V$, we have that

$$\partial_{w_{21,i}^V}\ell_3(\theta)\Big|_{w_{21}^{KQ}=w_{21}^V=0_{d\times1}}$$

$$= \frac{2\epsilon^2}{N}\cdot\mathbb{E}_\tau\left[2\cdot\left((w_{21}^V)^\top\quad w_{22}^V\right)\cdot\begin{pmatrix}W^EX_\tau\\Y_\tau\end{pmatrix}\cdot\begin{pmatrix}W^EX_\tau\\Y_\tau\end{pmatrix}^\top\cdot\left((e_i)^\top\quad 0\right)^\top\right]_{w_{21}^{KQ}=w_{21}^V=0_{d\times1}}\cdot\mathbb{E}_\tau\left[\|W_{11}^{KQ}W^Ex_{\tau,q}\|_2^2\right]$$

$$= \frac{2\epsilon^2}{N}\cdot\mathbb{E}_\tau\left[2\cdot\left((0_{d\times1})^\top\quad w_{22}^V\right)\cdot\begin{pmatrix}W^EX_\tau\\Y_\tau\end{pmatrix}\cdot\begin{pmatrix}W^EX_\tau\\Y_\tau\end{pmatrix}^\top\cdot\left((e_i)^\top\quad 0\right)^\top\right]\cdot\mathbb{E}_\tau\left[\|W_{11}^{KQ}W^Ex_{\tau,q}\|_2^2\right]$$

$$= \frac{2\epsilon^2}{N}\cdot\mathbb{E}_\tau\left[2\cdot w_{22}^V\cdot Y_\tau\cdot(W^EX_\tau)^\top\cdot e_i\right]\cdot\mathbb{E}_\tau\left[\|W_{11}^{KQ}W^Ex_{\tau,q}\|_2^2\right]$$

$$= \frac{4\epsilon^2}{N}\cdot\mathbb{E}_\tau\left[w_{22}^V\cdot w_\tau^\top\cdot W^EX_\tau\cdot(W^EX_\tau)^\top\cdot e_i\right]\cdot\mathbb{E}_\tau\left[\|W_{11}^{KQ}W^Ex_{\tau,q}\|_2^2\right]$$

$$= \frac{4\epsilon^2}{N}\cdot w_{22}^V\cdot\mathbb{E}_\tau[w_\tau^\top]\cdot\mathbb{E}_\tau\left[W^EX_\tau\cdot(W^EX_\tau)^\top\cdot e_i\right]\cdot\mathbb{E}_\tau\left[\|W_{11}^{KQ}W^Ex_{\tau,q}\|_2^2\right]$$

$$= \frac{4\epsilon^2}{N}\cdot w_{22}^V\cdot0_{1\times d}\cdot\mathbb{E}_\tau\left[W^EX_\tau\cdot(W^EX_\tau)^\top\cdot e_i\right]\cdot\mathbb{E}_\tau\left[\|W_{11}^{KQ}W^Ex_{\tau,q}\|_2^2\right] = 0,$$

where $e_i \in \mathbb{R}^{d \times 1}$ is an elementary vector that its $i$-th entry is 1 and all other remaining entries are 0. As a result, we thus have

$$\partial_{w_{21}^V} \ell_3(\theta)\Big|_{w_{21}^{KQ}=w_{21}^V=0_{d\times 1}} = 0_{1\times d}.$$

**Step 4: Show that** $w_{21}^{KQ} = w_{21}^V = 0_{d\times 1}$ **indicates** $\partial_{w_{21}^{KQ}} \ell_4(\theta) = \partial_{w_{21}^V} \ell_4(\theta) = 0_{1\times d}$. For $w_{21}^{KQ}$, since it does not exists in $\ell_4(\theta)$, thus we always have $\partial_{w_{21}^{KQ}} \ell_3(\theta) = 0$.

Besides, for $\partial_{w_{21}^V} \ell_4(\theta)$, we have

$$\partial_{w_{21}^V} \ell_4(\theta)\Big|_{w_{21}^{KQ}=w_{21}^V=0_{d\times 1}} = 2\epsilon^4 \cdot [2 \cdot (w_{21}^V)^\top]_{w_{21}^V=0_{d\times 1}} \cdot \mathbb{E}_\tau \Big[\|W_{11}^{KQ} W^E x_{\tau,q}\|_2^2\Big]$$

$$= 2\epsilon^4 \cdot [2 \cdot (0_{d\times 1})^\top] \cdot \mathbb{E}_\tau \Big[\|W_{11}^{KQ} W^E x_{\tau,q}\|_2^2\Big]$$

$$= 0_{1\times d}.$$

The proof is completed. $\qquad\square$

Based on Lemma B.6, we then simplify the objective function $\tilde{\mathcal{L}}_{\text{LSAE}}^{\text{adv}}(\theta)$ in the surrogate ICL embedding AT in Eq. (13), as shown in the following Lemma B.7.

**Lemma B.7.** *Under Assumption 1, the surrogate ICL embedding AT objective function $\tilde{\mathcal{L}}_{\text{LSAE}}^{\text{adv}}(\theta)$ defined in Eq. (13) can be simplified as*

$$\tilde{\mathcal{L}}_{\text{LSAE}}^{\text{adv}}(\theta) = 2(w_{22}^V)^2 \cdot \text{Tr}\Big[\big(W^E \Gamma_N \Lambda (W^E)^\top + \text{Tr}(\Lambda)\epsilon^2 I_d\big) \cdot (W_{11}^{KQ} W^E \Lambda^{\frac{1}{2}}) \cdot (W_{11}^{KQ} W^E \Lambda^{\frac{1}{2}})^\top\Big]$$

$$- 4w_{22}^V \cdot \text{Tr}\Big[(W_{11}^{KQ} W^E \Lambda^{\frac{1}{2}}) \cdot \Lambda^{\frac{3}{2}} (W^E)^\top\Big] + 2\text{Tr}(\Lambda),$$

*where $\Gamma_N := (\frac{N+1}{N}\Lambda + \frac{1}{N}\text{Tr}(\Lambda)I_{d_0}) \in \mathbb{R}^{d_0 \times d_0}$.*

*Proof.* When Assumption 1 holds, by applying Lemma B.6, $w_{21}^{KQ}$ and $w_{21}^V$ become zero vectors during the surrogate AT. Then, $\ell_2(\theta)$ and $\ell_4(\theta)$ in the surrogate AT loss $\tilde{\mathcal{L}}_{\text{LSAE}}^{\text{adv}}(\theta)$ will stay zero.

Besides, for $\ell_1(\theta)$ in $\tilde{\mathcal{L}}_{\text{LSAE}}^{\text{adv}}(\theta)$ in Eq. (13), it becomes

$$\ell_1(\theta) = 2 \cdot \mathbb{E}_\tau \Big| \big( \begin{matrix} 0_{1\times d} & w_{22}^V \end{matrix} \big) \cdot \frac{\mathcal{E}(Z_\tau)\mathcal{E}(Z_\tau)^\top}{N} \cdot \begin{pmatrix} W_{11}^{KQ} \\ 0_{1\times d} \end{pmatrix} \cdot W^E x_{\tau,q} - y_{\tau,q}\Big|^2$$

$$= 2 \cdot \mathbb{E}_\tau |\frac{1}{N} \cdot w_{22}^V \cdot \big( \begin{matrix} Y_\tau & 0 \end{matrix} \big) \cdot \big( \begin{matrix} W^E X_\tau & W^E x_{\tau,q} \end{matrix} \big)^\top \cdot W_{11}^{KQ} \cdot W^E x_{\tau,q} - y_{\tau,q}|^2$$

$$= \frac{2(w_{22}^V)^2}{N^2} \mathbb{E}_\tau \cdot \Big[x_{\tau,q}^\top ((W^E)^\top W_{11}^{KQ} W^E)^\top X_\tau Y_\tau^\top \cdot Y_\tau X_\tau^\top ((W^E)^\top W_{11}^{KQ} W^E) x_{\tau,q}\Big]$$

$$- \frac{4w_{22}^V}{N} \cdot \mathbb{E}_\tau \Big[Y_\tau X_\tau^\top ((W^E)^\top W_{11}^{KQ} W^E) x_{\tau,q} \cdot y_{\tau,q}\Big] + 2 \cdot \mathbb{E}_\tau [y_{\tau,q}^2]$$

$$= \frac{2(w_{22}^V)^2}{N^2} \mathbb{E}_\tau \Big[x_{\tau,q}^\top ((W^E)^\top W_{11}^{KQ} W^E)^\top X_\tau X_\tau^\top \cdot \mathbb{E}_\tau[w_\tau w_\tau^\top] \cdot X_\tau X_\tau^\top ((W^E)^\top W_{11}^{KQ} W^E) x_{\tau,q}\Big]$$

$$- \frac{4w_{22}^V}{N} \cdot \mathbb{E}_\tau \Big[w_\tau^\top \cdot X_\tau X_\tau^\top ((W^E)^\top W_{11}^{KQ} W^E) x_{\tau,q} x_{\tau,q}^\top \cdot w_\tau\Big] + 2 \cdot \mathbb{E}_\tau[w_\tau^\top \cdot x_{\tau,q} x_{\tau,q}^\top \cdot w_\tau]$$

$$= 2(w_{22}^V)^2 \cdot \text{Tr}\Big[((W^E)^\top W_{11}^{KQ} W^E)^\top \cdot \underbrace{\mathbb{E}_\tau[\frac{1}{N^2} X_\tau X_\tau^\top X_\tau X_\tau^\top] \cdot ((W^E)^\top W_{11}^{KQ} W^E) \cdot \Lambda}_{\text{Lemma B.2}}\Big]$$

$$- 4w_{22}^V \cdot \underbrace{\text{Tr}\Big[\frac{1}{N} \mathbb{E}_\tau[X_\tau X_\tau^\top] \cdot ((W^E)^\top W_{11}^{KQ} W^E) \cdot \mathbb{E}_\tau[x_{\tau,q} x_{\tau,q}^\top]\Big]}_{\text{Lemma B.2}} + 2\text{Tr}(\Lambda)$$

$$= 2(w_{22}^V)^2 \text{Tr}\Big[((W^E)^\top W_{11}^{KQ} W^E)^\top \cdot \frac{1}{N^2}\Big(\sum_{i=1}^N \mathbb{E}_\tau[x_{\tau,i} x_{\tau,i}^\top x_{\tau,i} x_{\tau,i}^\top] + \sum_{i\neq j} \mathbb{E}_\tau[x_{\tau,i} x_{\tau,i}^\top x_{\tau,j} x_{\tau,j}^\top]\Big) \cdot ((W^E)^\top W_{11}^{KQ} W^E) \cdot \Lambda\Big]$$

$$- 4w_{22}^V \cdot \mathrm{Tr}\Big[\Lambda \cdot ((W^E)^\top W_{11}^{KQ} W^E) \cdot \Lambda\Big] + 2\mathrm{Tr}(\Lambda)$$

$$= 2(w_{22}^V)^2 \mathrm{Tr}\Big[((W^E)^\top W_{11}^{KQ} W^E)^\top \cdot \frac{1}{N^2}\Big(N\underbrace{(2\Lambda^2 + \mathrm{Tr}(\Lambda)\Lambda)}_{\text{Lemma B.1}} + (N^2 - N)\Lambda^2\Big) \cdot ((W^E)^\top W_{11}^{KQ} W^E) \cdot \Lambda\Big]$$

$$- 4w_{22}^V \cdot \mathrm{Tr}\Big[\Lambda \cdot ((W^E)^\top W_{11}^{KQ} W^E) \cdot \Lambda\Big] + 2\mathrm{Tr}(\Lambda)$$

$$= 2(w_{22}^V)^2 \mathrm{Tr}\Big[((W^E)^\top W_{11}^{KQ} W^E)^\top \cdot \Gamma_N \Lambda \cdot ((W^E)^\top W_{11}^{KQ} W^E) \cdot \Lambda\Big]$$

$$- 4w_{22}^V \cdot \mathrm{Tr}\Big[\Lambda \cdot ((W^E)^\top W_{11}^{KQ} W^E) \cdot \Lambda\Big] + 2\mathrm{Tr}(\Lambda), \tag{B.9}$$

where $\Gamma_N := (\frac{N+1}{N}\Lambda + \frac{1}{N}\mathrm{Tr}(\Lambda) I_{d_0}) \in \mathbb{R}^{d_0 \times d_0}$.

For $\ell_3(\theta)$ in $\tilde{\mathcal{L}}_{\mathrm{LSAE}}^{\mathrm{adv}}(\theta)$ in Eq. (13), we have

$$\ell_3(\theta) = \frac{2\epsilon^2}{N} \cdot \mathbb{E}_\tau\Big[\| \begin{pmatrix} 0_{1\times d} & w_{22}^V \end{pmatrix} \begin{pmatrix} W^E X_\tau \\ Y_\tau \end{pmatrix} \|_2^2 \Big] \cdot \mathbb{E}_\tau\Big[\|W_{11}^{KQ} W^E x_{\tau,q}\|_2^2\Big]$$

$$= \frac{2\epsilon^2 (w_{22}^V)^2}{N} \cdot \mathbb{E}_\tau \|Y_\tau\|_2^2 \cdot \mathbb{E}_\tau\Big[x_{\tau,q}^\top (W_{11}^{KQ} W^E)^\top W_{11}^{KQ} W^E x_{\tau,q}\Big]$$

$$= 2\epsilon^2 (w_{22}^V)^2 \cdot \frac{1}{N}\sum_{i=1}^N \mathbb{E}_\tau[x_{\tau,i}^\top w_\tau w_\tau^\top x_{\tau,i}] \cdot \underbrace{\mathrm{Tr}\Big[(W_{11}^{KQ} W^E)^\top W_{11}^{KQ} W^E \Lambda\Big]}_{\text{Lemma B.2}}$$

$$= 2\epsilon^2 (w_{22}^V)^2 \cdot \mathrm{Tr}(\Lambda) \cdot \mathrm{Tr}\Big[(W_{11}^{KQ} W^E)^\top W_{11}^{KQ} W^E \Lambda\Big]. \tag{B.10}$$

Finally, by inserting Eqs. (B.9) and (B.10) back into the surrogate AT loss $\tilde{\mathcal{L}}_{\mathrm{LSAE}}^{\mathrm{adv}}(\theta)$ in Eq. (13) and repeatedly applying Lemma B.3 and the commutativity between $\Lambda$ and $\Gamma_N$, we thus have

$$\tilde{\mathcal{L}}_{\mathrm{LSAE}}^{\mathrm{adv}}(\theta) = 2(w_{22}^V)^2 \cdot \mathrm{Tr}\Big[((W^E)^\top W_{11}^{KQ} W^E)^\top \cdot \Gamma_N \Lambda \cdot ((W^E)^\top W_{11}^{KQ} W^E) \cdot \Lambda\Big]$$

$$- 4w_{22}^V \cdot \mathrm{Tr}\Big[\Lambda \cdot ((W^E)^\top W_{11}^{KQ} W^E) \cdot \Lambda\Big] + 2\mathrm{Tr}(\Lambda)$$

$$+ 2\epsilon^2 (w_{22}^V)^2 \cdot \mathrm{Tr}(\Lambda) \cdot \mathrm{Tr}\Big[(W_{11}^{KQ} W^E)^\top W_{11}^{KQ} W^E \Lambda\Big]$$

$$= 2(w_{22}^V)^2 \cdot \mathrm{Tr}\Big[W^E \Gamma_N \Lambda (W^E)^\top \cdot (W_{11}^{KQ} W^E \Lambda^{\frac{1}{2}}) \cdot (W_{11}^{KQ} W^E \Lambda^{\frac{1}{2}})^\top\Big]$$

$$- 4w_{22}^V \cdot \mathrm{Tr}\Big[(W_{11}^{KQ} W^E \Lambda^{\frac{1}{2}}) \cdot \Lambda^{\frac{3}{2}} (W^E)^\top\Big] + 2\mathrm{Tr}(\Lambda)$$

$$+ 2\epsilon^2 (w_{22}^V)^2 \mathrm{Tr}(\Lambda) \cdot \mathrm{Tr}\Big[(W_{11}^{KQ} W^E \Lambda^{\frac{1}{2}}) \cdot (W_{11}^{KQ} W^E \Lambda^{\frac{1}{2}})^\top\Big]$$

$$= 2(w_{22}^V)^2 \cdot \mathrm{Tr}\Big[(W^E \Gamma_N \Lambda (W^E)^\top + \mathrm{Tr}(\Lambda)\epsilon^2 I_d) \cdot (W_{11}^{KQ} W^E \Lambda^{\frac{1}{2}}) \cdot (W_{11}^{KQ} W^E \Lambda^{\frac{1}{2}})^\top\Big]$$

$$- 4w_{22}^V \cdot \mathrm{Tr}\Big[(W_{11}^{KQ} W^E \Lambda^{\frac{1}{2}}) \cdot \Lambda^{\frac{3}{2}} (W^E)^\top\Big] + 2\mathrm{Tr}(\Lambda). \tag{B.11}$$

The proof is completed. $\qquad\square$

Based on the simplified surrogate AT loss, we now calculate the global solution for the surrogate embedding ICL AT problem in the following Lemma B.8.

**Lemma B.8.** *Suppose Assumption 1 holds. Then, $\theta_* := (W_*^E, W_*^{KQ}, W_*^V)$ is a global minimizer for the surrogate embedding ICL AT problem defined in Eq. (13) if and only if*

$$w_{*,22}^V (W_*^E)^\top W_{*,11}^{KQ} W_*^E = (W_*^E)^\top (W_*^E \Gamma_N \Lambda (W_*^E)^\top + \mathrm{Tr}(\Lambda)\epsilon^2 I_d)^{-1} W_*^E \Lambda.$$

*Proof.* Under Assumption 1, by applying Lemma B.3 and Lemma B.7, we can re-organize the surrogate AT loss $\tilde{\mathcal{L}}_{\text{LSAE}}^{\text{adv}}(\theta)$ in Eq. (13) as follows,

$$
\begin{aligned}
&\tilde{\mathcal{L}}_{\text{LSAE}}^{\text{adv}}(\theta) \\
&= 2(w_{22}^V)^2 \cdot \text{Tr}\Big[\big(W^E\Gamma_N\Lambda(W^E)^\top + \text{Tr}(\Lambda)\epsilon^2 I_d\big) \cdot (W_{11}^{KQ}W^E\Lambda^{\frac{1}{2}}) \cdot (W_{11}^{KQ}W^E\Lambda^{\frac{1}{2}})^\top\Big] \\
&\quad - 4w_{22}^V \cdot \text{Tr}\Big[(W_{11}^{KQ}W^E\Lambda^{\frac{1}{2}}) \cdot \Lambda^{\frac{3}{2}}(W^E)^\top\Big] + 2\text{Tr}(\Lambda) \\
&= 2 \cdot \text{Tr}\Big[\big(W^E\Gamma_N\Lambda(W^E)^\top + \text{Tr}(\Lambda)\epsilon^2 I_d\big) \cdot (w_{22}^V W_{11}^{KQ}W^E\Lambda^{\frac{1}{2}}) \cdot (w_{22}^V W_{11}^{KQ}W^E\Lambda^{\frac{1}{2}})^\top\Big] + 2\text{Tr}(\Lambda) \\
&\quad - 4 \cdot \text{Tr}\Big[\big(W^E\Gamma_N\Lambda(W^E)^\top + \text{Tr}(\Lambda)\epsilon^2 I_d\big) \cdot (w_{22}^V W_{11}^{KQ}W^E\Lambda^{\frac{1}{2}}) \cdot \Lambda^{\frac{3}{2}}(W^E)^\top\big(W^E\Gamma_N\Lambda(W^E)^\top + \text{Tr}(\Lambda)\epsilon^2 I_d\big)^{-1}\Big] \\
&= 2 \cdot \text{Tr}\Big[\big(W^E\Gamma_N\Lambda(W^E)^\top + \text{Tr}(\Lambda)\epsilon^2 I_d\big) \\
&\qquad\quad \cdot \Big(w_{22}^V W_{11}^{KQ}W^E\Lambda^{\frac{1}{2}} - \big(W^E\Gamma_N\Lambda(W^E)^\top + \text{Tr}(\Lambda)\epsilon^2 I_d\big)^{-1}W^E\Lambda^{\frac{3}{2}}\Big) \\
&\qquad\quad \cdot \Big(w_{22}^V W_{11}^{KQ}W^E\Lambda^{\frac{1}{2}} - \big(W^E\Gamma_N\Lambda(W^E)^\top + \text{Tr}(\Lambda)\epsilon^2 I_d\big)^{-1}W^E\Lambda^{\frac{3}{2}}\Big)^\top\Big] \\
&\quad - \text{Tr}\Big[\big(W^E\Gamma_N\Lambda(W^E)^\top + \text{Tr}(\Lambda)\epsilon^2 I_d\big)^{-2}W^E\Lambda^3(W^E)^\top\Big] + 2\text{Tr}(\Lambda). \qquad\qquad (\text{B}.12)
\end{aligned}
$$

Note that the second and third summation terms in Eq. (B.12) are constants. Besides, the first term in Eq. (B.12) is non-negative and can achieve zero via setting

$$
w_{*,22}^V W_{*,11}^{KQ}W_*^E\Lambda^{\frac{1}{2}} - (W_*^E\Gamma_N\Lambda(W_*^E)^\top + \text{Tr}(\Lambda)\epsilon^2 I_d)^{-1}W_*^E\Lambda^{\frac{3}{2}} = 0,
$$

which is

$$
w_{*,22}^V(W_*^E)^\top W_{*,11}^{KQ}W_*^E = (W_*^E)^\top(W_*^E\Gamma_N\Lambda(W_*^E)^\top + \text{Tr}(\Lambda)\epsilon^2 I_d)^{-1}W_*^E\Lambda.
$$

The proof is completed. $\qquad\qquad\qquad\qquad\qquad\qquad\qquad\qquad\qquad\qquad\qquad\square$

### B.4 PROOF OF THEOREM 2

We first prove a useful Lemma B.9 that will be frequently used in this section.

**Lemma B.9.** *For the inverse matrix* $(W_*^E\Gamma_N\Lambda(W_*^E)^\top + \text{Tr}(\Lambda)\epsilon^2 I_d)^{-1} \in \mathbb{R}^{d\times d}$ *in the optimal surrogate ICL embedding AT solution in Theorem 1, we have*

$$
\Big\|\big(W_*^E\Gamma_N\Lambda(W_*^E)^\top + \text{Tr}(\Lambda)\epsilon^2 I_d\big)^{-1}\Big\|_2 \le \frac{1}{\sigma_{\min}(\Gamma_N\Lambda) \cdot \sigma_{\min}(W_*^E)^2 + \text{Tr}(\Lambda)\epsilon^2}.
$$

*Proof.* According to the definition of matrix operator norm (*i.e.*, $\|\cdot\|_2$), we have

$$
\begin{aligned}
&\Big\|\big(W_*^E\Gamma_N\Lambda(W_*^E)^\top + \text{Tr}(\Lambda)\epsilon^2 I_d\big)^{-1}\Big\|_2 = \sigma_{\max}\Big[\big(W_*^E\Gamma_N\Lambda(W_*^E)^\top + \text{Tr}(\Lambda)\epsilon^2 I_d\big)^{-1}\Big] \\
&= \frac{1}{\sigma_{\min}\Big[W_*^E\Gamma_N\Lambda(W_*^E)^\top + \text{Tr}(\Lambda)\epsilon^2 I_d\Big]} = \frac{1}{\sigma_{\min}\Big[W_*^E\Gamma_N\Lambda(W_*^E)^\top\Big] + \text{Tr}(\Lambda)\epsilon^2}. \qquad (\text{B}.13)
\end{aligned}
$$

Notice that $W_*^E\Gamma_N\Lambda(W_*^E)^\top$ is a positive semidefinite matrix, we thus have

$$
\sigma_{\min}\Big[W_*^E\Gamma_N\Lambda(W_*^E)^\top\Big] = \lambda_{\min}\Big[W_*^E\Gamma_N\Lambda(W_*^E)^\top\Big]. \qquad\qquad (\text{B}.14)
$$

By further applying Lemma B.5,

$$
\lambda_{\min}\Big[W_*^E\Gamma_N\Lambda(W_*^E)^\top\Big] = \min_{v\in\mathbb{R}^d, \|v\|_2=1} v^\top W_*^E\Gamma_N\Lambda(W_*^E)^\top v. \qquad\qquad (\text{B}.15)
$$

Denote that $u = (W_*^E)^\top v \in \mathbb{R}^d$, then Eq. (B.15) can be re-written as

$$
\begin{aligned}
\lambda_{\min}\left[W_*^E \Gamma_N \Lambda (W_*^E)^\top\right] &= \min_{v \in \mathbb{R}^d, \|v\|_2=1} v^\top W_*^E \Gamma_N \Lambda (W_*^E)^\top v \\
&= \min_{v \in \mathbb{R}^d, \|v\|_2=1} u^\top (\Gamma_N \Lambda) u \\
&= \min_{v \in \mathbb{R}^d, \|v\|_2=1} \left\{ \frac{u^\top (\Gamma_N \Lambda) u}{u^\top u} \cdot u^\top u \right\} \\
&\geq \min_{u' \in \mathbb{R}^d, \|u'\| \neq 0_d} \frac{u'^\top (\Gamma_N \Lambda) u'}{u'^\top u'} \cdot \min_{v \in \mathbb{R}^d, \|v\|_2=1} u^\top u \\
&\overset{(*_1)}{=} \lambda_{\min}(\Gamma_N \Lambda) \cdot \min_{v \in \mathbb{R}^d, \|v\|_2=1} v^T W_*^E (W_*^E)^\top v, \\
&\overset{(*_2)}{=} \lambda_{\min}(\Gamma_N \Lambda) \cdot \lambda_{\min}(W_*^E (W_*^E)^\top) \\
&\overset{(*_3)}{=} \sigma_{\min}(\Gamma_N \Lambda) \cdot \sigma_{\min}(W_*^E (W_*^E)^\top) \\
&= \sigma_{\min}(\Gamma_N \Lambda) \cdot \sigma_{\min}(W_*^E)^2,
\end{aligned}
\tag{B.16}
$$

where both $(*_1)$ and $(*_2)$ are obtained via again applying Lemma B.5, and $(*_3)$ is due to the fact that both $\Gamma_N \Lambda$ and $W_*^E (W_*^E)^\top$ are positive semidefinite matrices. Combining Eqs. (B.13), (B.14), and (B.16), we finally have that

$$
\mathrm{Tr}\left[\left(W_*^E \Gamma_N \Lambda (W_*^E)^\top + \mathrm{Tr}(\Lambda)\epsilon^2 I_d\right)^{-1}\right] \leq \frac{1}{\sigma_{\min}(\Gamma_N \Lambda) \cdot \sigma_{\min}(W_*^E)^2 + \mathrm{Tr}(\Lambda)\epsilon^2},
$$

which completes the proof. $\qquad\square$

With the help of Lemma B.9, we can now start to prove Theorem 2.

*Proof of Theorem 2.* For the converged LSA-E model $f_{\mathrm{LSAE},\theta*}(\cdot)$ trained from the surrogate ICL embedding AT, by inserting its prediction function $\hat{y}_{q,\theta*}(\cdot)$ given in Eq. (14) into the robust risk $\mathcal{R}_{\rho,M}^{\mathrm{adv}}(\cdot)$ defined in Eq. (12) and using the inequality $|a+b|^2 \leq 2(a^2+b^2)$, we have that

$$
\begin{aligned}
&\mathcal{R}_{\rho,M}^{\mathrm{adv}}(\theta_*) \\
&= \mathbb{E}_\tau \max_{\|\Delta_\tau^{O\top}\|_{2,\infty} \leq \rho} \frac{1}{2}\left|\frac{1}{N} Y_\tau (X_\tau + (0_{d_0 \times (N-M)} \quad \Delta_\tau^O))^\top \cdot (W_*^E)^\top \left(W_*^E \Gamma_N \Lambda (W_*^E)^\top + \mathrm{Tr}(\Lambda)\epsilon^2 I_d\right)^{-1} W_*^E \Lambda \cdot x_{\tau,q} - y_{\tau,q}\right|^2 \\
&\leq \underbrace{\frac{1}{N^2} \mathbb{E}_\tau \left|Y_\tau(X_\tau)^\top \cdot (W_*^E)^\top \left(W_*^E \Gamma_N \Lambda (W_*^E)^\top + \mathrm{Tr}(\Lambda)\epsilon^2 I_d\right)^{-1} W_*^E \Lambda \cdot x_{\tau,q}\right|^2}_{:=C_1} \\
&\quad \underbrace{- \frac{2}{N} \mathbb{E}_\tau \left[Y_\tau(X_\tau)^\top \cdot (W_*^E)^\top \left(W_*^E \Gamma_N \Lambda (W_*^E)^\top + \mathrm{Tr}(\Lambda)\epsilon^2 I_d\right)^{-1} W_*^E \Lambda \cdot x_{\tau,q} \cdot y_{\tau,q}\right]}_{:=C_2} + \underbrace{\mathbb{E}_\tau [y_{\tau,q}^2]}_{:=C_3} \\
&\quad + \underbrace{\mathbb{E}_\tau \max_{\|\Delta_\tau^{O\top}\|_{2,\infty} \leq \rho} \left|\frac{1}{N} Y_{\tau,(N-M+1):N}(\Delta_\tau^O)^\top \cdot (W_*^E)^\top \left(W_*^E \Gamma_N \Lambda (W_*^E)^\top + \mathrm{Tr}(\Lambda)\epsilon^2 I_d\right)^{-1} W_*^E \Lambda \cdot x_{\tau,q}\right|^2}_{:=C_4},
\end{aligned}
\tag{B.17}
$$

where $Y_{\tau,(N-M+1):N} := (y_{\tau,N-M+1} \quad \cdots \quad y_{\tau,N}) \in \mathbb{R}^{1 \times M}$.

**For $C_1$ in Eq. (B.17)**, we have

$$C_1 := \frac{1}{N^2} \mathbb{E}_\tau \left| Y_\tau (X_\tau)^\top \cdot (W_*^E)^\top \left( W_*^E \Gamma_N \Lambda (W_*^E)^\top + \text{Tr}(\Lambda)\epsilon^2 I_d \right)^{-1} W_*^E \Lambda \cdot x_{\tau,q} \right|^2$$

$$= \frac{1}{N^2} \mathbb{E}_\tau \left[ x_{\tau,q}^\top \cdot (W_*^E \Lambda)^\top \left( W_*^E \Gamma_N \Lambda (W_*^E)^\top + \text{Tr}(\Lambda)\epsilon^2 I_d \right)^{-1} W_*^E \cdot \mathbb{E}_\tau [X_\tau \cdot X_\tau^\top w_\tau \cdot w_\tau^\top X_\tau \cdot (X_\tau)^\top] \right.$$
$$\left. \cdot (W_*^E)^\top \left( W_*^E \Gamma_N \Lambda (W_*^E)^\top + \text{Tr}(\Lambda)\epsilon^2 I_d \right)^{-1} W_*^E \Lambda \cdot x_{\tau,q} \right]$$

$$= \mathbb{E}_\tau \left[ x_{\tau,q}^\top \cdot (W_*^E \Lambda)^\top \left( W_*^E \Gamma_N \Lambda (W_*^E)^\top + \text{Tr}(\Lambda)\epsilon^2 I_d \right)^{-1} W_*^E \right.$$
$$\cdot \frac{1}{N^2} \left( \sum_i \mathbb{E}_\tau [x_{\tau,i} x_{\tau,i}^\top x_{\tau,i} x_{\tau,i}^\top] + \sum_{i \neq j} \mathbb{E}_\tau [x_{\tau,i} x_{\tau,i}^\top] \cdot \mathbb{E}_\tau [x_{\tau,j} x_{\tau,j}^\top] \right)$$
$$\left. \cdot (W_*^E)^\top \left( W_*^E \Gamma_N \Lambda (W_*^E)^\top + \text{Tr}(\Lambda)\epsilon^2 I_d \right)^{-1} W_*^E \Lambda \cdot x_{\tau,q} \right]$$

$$= \mathbb{E}_\tau \left[ x_{\tau,q}^\top \cdot (W_*^E \Lambda)^\top \left( W_*^E \Gamma_N \Lambda (W_*^E)^\top + \text{Tr}(\Lambda)\epsilon^2 I_d \right)^{-1} W_*^E \cdot \frac{1}{N^2} \left( \underbrace{N(2\Lambda^2 + \text{Tr}(\Lambda)\Lambda)}_{\text{Lemma B.1}} + (N^2 - N)\Lambda^2 \right) \right.$$
$$\left. \cdot (W_*^E)^\top \left( W_*^E \Gamma_N \Lambda (W_*^E)^\top + \text{Tr}(\Lambda)\epsilon^2 I_d \right)^{-1} W_*^E \Lambda \cdot x_{\tau,q} \right]$$

$$= \mathbb{E}_\tau \left[ x_{\tau,q}^\top \cdot (W_*^E \Lambda)^\top \left( W_*^E \Gamma_N \Lambda (W_*^E)^\top + \text{Tr}(\Lambda)\epsilon^2 I_d \right)^{-1} W_*^E \cdot \Gamma_N \Lambda \right.$$
$$\left. \cdot (W_*^E)^\top \left( W_*^E \Gamma_N \Lambda (W_*^E)^\top + \text{Tr}(\Lambda)\epsilon^2 I_d \right)^{-1} W_*^E \Lambda \cdot x_{\tau,q} \right]$$

$$= \underbrace{\text{Tr} \left[ (W_*^E \Lambda)^\top \left( W_*^E \Gamma_N \Lambda (W_*^E)^\top + \text{Tr}(\Lambda)\epsilon^2 I_d \right)^{-1} W_*^E \cdot \Gamma_N \Lambda \cdot (W_*^E)^\top \left( W_*^E \Gamma_N \Lambda (W_*^E)^\top + \text{Tr}(\Lambda)\epsilon^2 I_d \right)^{-1} W_*^E \Lambda \cdot \Lambda \right]}_{\text{Lemma B.2}}.$$

Then, by repeatedly applying Lemma B.3 and Lemma B.4, we further have

$$C_1 := \text{Tr} \left[ (W_*^E \Lambda)^\top \left( W_*^E \Gamma_N \Lambda (W_*^E)^\top + \text{Tr}(\Lambda)\epsilon^2 I_d \right)^{-1} W_*^E \cdot \Gamma_N \Lambda \cdot (W_*^E)^\top \left( W_*^E \Gamma_N \Lambda (W_*^E)^\top + \text{Tr}(\Lambda)\epsilon^2 I_d \right)^{-1} W_*^E \Lambda^2 \right]$$

$$\overset{(*_1)}{=} \text{Tr} \left[ (W_*^E)^\top \left( W_*^E \Gamma_N \Lambda (W_*^E)^\top + \text{Tr}(\Lambda)\epsilon^2 I_d \right)^{-1} W_*^E \cdot \Gamma_N \Lambda \cdot (W_*^E)^\top \left( W_*^E \Gamma_N \Lambda (W_*^E)^\top + \text{Tr}(\Lambda)\epsilon^2 I_d \right)^{-1} W_*^E \cdot \Lambda^3 \right]$$

$$\overset{(*_2)}{\leq} \text{Tr} \left[ (W_*^E)^\top \left( W_*^E \Gamma_N \Lambda (W_*^E)^\top + \text{Tr}(\Lambda)\epsilon^2 I_d \right)^{-1} W_*^E \Gamma_N \Lambda (W_*^E)^\top \left( W_*^E \Gamma_N \Lambda (W_*^E)^\top + \text{Tr}(\Lambda)\epsilon^2 I_d \right)^{-1} W_*^E \right] \cdot \lambda_{\max}(\Lambda^3)$$

$$\overset{(*_3)}{=} \text{Tr} \left[ (W_*^E)^\top \left( W_*^E \Gamma_N \Lambda (W_*^E)^\top + \text{Tr}(\Lambda)\epsilon^2 I_d \right)^{-1} W_*^E (W_*^E)^\top \left( W_*^E \Gamma_N \Lambda (W_*^E)^\top + \text{Tr}(\Lambda)\epsilon^2 I_d \right)^{-1} W_*^E \Gamma_N \Lambda \right] \cdot \lambda_{\max}(\Lambda)^3$$

$$\overset{(*_4)}{\leq} \text{Tr} \left[ (W_*^E)^\top \left( W_*^E \Gamma_N \Lambda (W_*^E)^\top + \text{Tr}(\Lambda)\epsilon^2 I_d \right)^{-1} W_*^E (W_*^E)^\top \left( W_*^E \Gamma_N \Lambda (W_*^E)^\top + \text{Tr}(\Lambda)\epsilon^2 I_d \right)^{-1} W_*^E \right]$$
$$\cdot \lambda_{\max}(\Gamma_N \Lambda) \cdot \lambda_{\max}(\Lambda)^3$$

$$\overset{(*_5)}{\leq} \text{Tr} \left[ W_*^E (W_*^E)^\top \left( W_*^E \Gamma_N \Lambda (W_*^E)^\top + \text{Tr}(\Lambda)\epsilon^2 I_d \right)^{-1} W_*^E (W_*^E)^\top \right] \cdot \lambda_{\max} \left[ \left( W_*^E \Gamma_N \Lambda (W_*^E)^\top + \text{Tr}(\Lambda)\epsilon^2 I_d \right)^{-1} \right]$$
$$\cdot \lambda_{\max}(\Gamma_N) \cdot \lambda_{\max}(\Lambda)^4$$

$$\overset{(*_6)}{\leq} \text{Tr} \left[ W_*^E (W_*^E)^\top W_*^E (W_*^E)^\top \left( W_*^E \Gamma_N \Lambda (W_*^E)^\top + \text{Tr}(\Lambda)\epsilon^2 I_d \right)^{-1} \right] \cdot \frac{\lambda_{\max}(\Gamma_N) \cdot \lambda_{\max}(\Lambda)^4}{\sigma_{\min}(\Gamma_N \Lambda) \cdot \sigma_{\min}(W_*^E)^2 + \text{Tr}(\Lambda)\epsilon^2}$$

$$\overset{(*_7)}{\leq} \text{Tr} \left[ W_*^E (W_*^E)^\top W_*^E (W_*^E)^\top \right] \cdot \frac{\lambda_{\max}(\Gamma_N) \cdot \lambda_{\max}(\Lambda)^4}{[\sigma_{\min}(\Gamma_N \Lambda) \cdot \sigma_{\min}(W_*^E)^2 + \text{Tr}(\Lambda)\epsilon^2]^2}$$

$$= \frac{\sigma_{\max}(\Gamma_N \Lambda^4) \cdot \sum_{i=1}^d \sigma_i(W_*^E)^4}{[\sigma_{\min}(\Gamma_N \Lambda) \cdot \sigma_{\min}(W_*^E)^2 + \text{Tr}(\Lambda)\epsilon^2]^2}, \tag{B.18}$$

where $(*_1)$ and $(*_3)$ is due to Lemma B.3, $(*_2)$, $(*_4)$ is due to Lemma B.4, $(*_5)$ is by Lemma B.3 and Lemma B.4, $(*_6)$ is by Lemma B.3 and Lemma B.9, and $(*_7)$ is by Lemma B.4 and Lemma B.9.

**For $C_2$ in Eq. (B.17)**, we have

$$
\begin{aligned}
C_2 &:= \frac{2}{N} \mathbb{E}_\tau \Big[ Y_\tau(X_\tau)^\top \cdot (W_*^E)^\top \Big( W_*^E \Gamma_N \Lambda (W_*^E)^\top + \text{Tr}(\Lambda)\epsilon^2 I_d \Big)^{-1} W_*^E \Lambda \cdot x_{\tau,q} \cdot y_{\tau,q} \Big] \\
&= \frac{2}{N} \mathbb{E}_\tau \Big[ w_\tau^\top \cdot X_\tau(X_\tau)^\top \cdot (W_*^E)^\top \Big( W_*^E \Gamma_N \Lambda (W_*^E)^\top + \text{Tr}(\Lambda)\epsilon^2 I_d \Big)^{-1} W_*^E \Lambda \cdot x_{\tau,q} \cdot x_{\tau,q}^\tau \cdot w_\tau \Big] \\
&\overset{(*)}{=} \frac{2}{N} \text{Tr}\Big[ \mathbb{E}_\tau [X_\tau(X_\tau)^\top] \cdot (W_*^E)^\top \Big( W_*^E \Gamma_N \Lambda (W_*^E)^\top + \text{Tr}(\Lambda)\epsilon^2 I_d \Big)^{-1} W_*^E \Lambda \cdot \mathbb{E}_\tau [x_{\tau,q} x_{\tau,q}^\tau] \Big] \\
&= \frac{2}{N} \text{Tr}\Big[ N\Lambda \cdot (W_*^E)^\top \Big( W_*^E \Gamma_N \Lambda (W_*^E)^\top + \text{Tr}(\Lambda)\epsilon^2 I_d \Big)^{-1} W_*^E \Lambda \cdot \Lambda \Big] \\
&\overset{(**)}{=} 2 \cdot \text{Tr}\Big[ \Lambda^{\frac{3}{2}} \cdot (W_*^E)^\top \Big( W_*^E \Gamma_N \Lambda (W_*^E)^\top + \text{Tr}(\Lambda)\epsilon^2 I_d \Big)^{-1} W_*^E \cdot \Lambda^{\frac{3}{2}} \Big] \geq 0,
\end{aligned}
\tag{B.19}
$$

where $(*)$ is by Lemma B.2 and $(**)$ is by Lemma B.3.

**For $C_3$ in Eq. (B.17)**, we have

$$
C_3 := \mathbb{E}_\tau [y_{\tau,q}^2] = \mathbb{E}_\tau [w_\tau^\top x_{\tau,q} x_{\tau,q}^\top w_\tau] = \text{Tr}(\mathbb{E}_\tau [x_{\tau,q} x_{\tau,q}^\top]) = \text{Tr}(\Lambda).
\tag{B.20}
$$

**For $C_4$ in Eq. (B.17)**, we have

$$
\begin{aligned}
C_4 &:= \mathbb{E}_\tau \max_{\|\Delta_\tau^{O\top}\|_{2,\infty} \leq \rho} \Big| \frac{1}{N} Y_{\tau,(M-N+1):N}(\Delta_\tau^O)^\top \cdot (W_*^E)^\top \Big( W_*^E \Gamma_N \Lambda (W_*^E)^\top + \text{Tr}(\Lambda)\epsilon^2 I_d \Big)^{-1} W_*^E \Lambda \cdot x_{\tau,q} \Big|^2 \\
&\leq \frac{1}{N^2} \cdot \mathbb{E}_\tau \|Y_{\tau,(M-N+1):N}\|_2^2 \cdot \mathbb{E}_\tau \max_{\|\Delta_\tau^O\|_{2,\infty} \leq \rho} \Big\| (\Delta_\tau^O)^\top \cdot (W_*^E)^\top \Big( W_*^E \Gamma_N \Lambda (W_*^E)^\top + \text{Tr}(\Lambda)\epsilon^2 I_d \Big)^{-1} W_*^E \Lambda \cdot x_{\tau,q} \Big\|_2^2 \\
&= \frac{M}{N^2} \text{Tr}(\Lambda) \cdot \rho^2 \cdot \mathbb{E}_\tau \Big\| (W_*^E)^\top \Big( W_*^E \Gamma_N \Lambda (W_*^E)^\top + \text{Tr}(\Lambda)\epsilon^2 I_d \Big)^{-1} W_*^E \Lambda \cdot x_{\tau,q} \Big\|_2^2 \\
&\overset{(*)}{=} (M\rho^2 \text{Tr}(\Lambda)/N^2) \\
&\quad \cdot \text{Tr}\Big[ \Lambda (W_*^E)^\top \Big( W_*^E \Gamma_N \Lambda (W_*^E)^\top + \text{Tr}(\Lambda)\epsilon^2 I_d \Big)^{-1} W_*^E (W_*^E)^\top \Big( W_*^E \Gamma_N \Lambda (W_*^E)^\top + \text{Tr}(\Lambda)\epsilon^2 I_d \Big)^{-1} W_*^E \Lambda \cdot \Lambda \Big],
\end{aligned}
$$

where $(*)$ follows Lemma B.2. By using a similar derivation as that for Eq. (B.18), we thus have

$$
C_4 \leq \frac{M\rho^2 \text{Tr}(\Lambda)}{N^2} \cdot \frac{\sigma_{\max}(\Lambda)^3 \cdot \sum_{i=1}^d \sigma_i(W_*^E)^4}{[\sigma_{\min}(\Gamma_N \Lambda) \cdot \sigma_{\min}(W_*^E)^2 + \text{Tr}(\Lambda)\epsilon^2]^2}.
\tag{B.21}
$$

Finally, inserting Eqs. (B.18), (B.19), (B.20), and (B.21) into Eq. (B.17), we thus have

$$
\begin{aligned}
\mathcal{R}_{\rho,M}^{\text{adv}}&(\theta_*) \\
&\leq \frac{\sigma_{\max}(\Gamma_N \Lambda^4) \cdot \sum_{i=1}^d \sigma_i(W_*^E)^4}{[\sigma_{\min}(\Gamma_N \Lambda) \cdot \sigma_{\min}(W_*^E)^2 + \text{Tr}(\Lambda)\epsilon^2]^2} - 0 + \text{Tr}(\Lambda) + \frac{M\rho^2 \text{Tr}(\Lambda)}{N^2} \cdot \frac{\sigma_{\max}(\Lambda)^3 \cdot \sum_{i=1}^d \sigma_i(W_*^E)^4}{[\sigma_{\min}(\Gamma_N \Lambda) \cdot \sigma_{\min}(W_*^E)^2 + \text{Tr}(\Lambda)\epsilon^2]^2}. \\
&= \cdot \frac{\Big( \sigma_{\max}(\Gamma_N \Lambda) + \frac{M\rho^2 \text{Tr}(\Lambda)}{N^2} \Big) \cdot \sigma_{\max}(\Lambda)^3 \cdot \sum_{i=1}^d \sigma_i(W_*^E)^4}{[\sigma_{\min}(\Gamma_N \Lambda) \cdot \sigma_{\min}(W_*^E)^2 + \text{Tr}(\Lambda)\epsilon^2]^2} + \text{Tr}(\Lambda) \\
&\leq \mathcal{O}\Big( \frac{(1 + M\rho^2/N^2) \cdot \sum_{i=1}^d \sigma_i(W_*^E)^4}{\sigma_{\min}(W_*^E)^4 + \epsilon^4} \Big) + \mathcal{O}(1).
\end{aligned}
$$

The proof is completed. $\qquad\square$

## C ADDITIONAL EXPERIMENTAL DETAILS

This section collects experimental details omitted from Section 5.

Table 5: LC-WinRate on models trained via CAT with different toward/away cut-off thresholds.

| Type (Toward / Away Cut-offs) | (Utility) LC-WinRate (%) ↑ | | | | | |
|---|---|---|---|---|---|---|
| | Vicuna-7B | Mistral-7B | Llama-2-7B | Llama-3.1-8B | Qwen2.5-7B | Gemma-2B |
| Original | 76.86 | 90.96 | 86.70 | 85.99 | 91.14 | 63.96 |
| CAT (1.0 / −3.0) | 36.66 | 15.76 | 67.51 | 45.71 | 77.07 | 41.75 |
| CAT (0.5 / −5.0) | 23.60 | 12.12 | 67.65 | 52.46 | 71.11 | 38.91 |

### C.1 ADVERSARIAL TRAINING

**Searching embedding space adversarial perturbation.** We leverage the projected gradient descent (PGD) (Madry et al., 2018) to solve the following embedding space adversarial perturbation searching problem for any harmful input-output pair $(x, y)$,

$$\delta^* = \left\lfloor \underset{\|\delta_1\|_2, \cdots, \|\delta_{|x|}\|_2 \leq \epsilon}{\arg\max} \log p_\theta(y|\mathcal{E}(x) + \delta) \right\rfloor.$$

Specifically, PGD will first initialize a starting perturbation as $\delta^{(0)} := 0_{d \times |x|}$, and then iteratively update it for $K$ times. In the $k$-th iteration, the update is as follows,

$$\delta_i^{(k)} = \prod_{\|\delta_i\|_2 \leq \epsilon} \left[ \delta_i^{(k-1)} + \eta \cdot \frac{\partial_{\delta_i} \log p_\theta(y|\mathcal{E}(x) + \delta^{(k-1)})}{\|\partial_{\delta_i} \log p_\theta(y|\mathcal{E}(x) + \delta^{(k-1)})\|_2} \right], \quad \forall i \in \{1, \cdots, |x|\},$$

where $\delta^{(k)}$ is the intermediate adversarial perturbation found in the $k$-th iteration, $\delta_i^{(k)}$ is the perturbation for the $i$-th token embedding, $\eta > 0$ is the step size, and $\prod_{\|\delta_i\|_2 \leq \epsilon}$ means that the perturbation $\delta_i$ for the $i$-th input token embedding each token embedding is projected in a ball sphere centered at $0_{d \times 1}$. The eventual adversarial perturbation is $\delta^* := \delta^{(K)}$.

In all experiments, the perturbation radius $\epsilon$ is set as 0.05, the update number $K$ is set as 10, and the learning rate is set as $1 \times 10^{-2}$.

**Loss cut-off technique in LLM AT.** The adversarial loss in CAT and ER-CAT can be decomposed into a *toward* loss and an *away* loss as follows,

$$\underset{(x,y,\tilde{y}) \in D^{(h)}}{\mathbb{E}} \left[ \underbrace{(-\log p_\theta(y|\mathcal{E}(x) + \delta^*))}_{\text{Toward Loss}} + \underbrace{\log p_\theta(\tilde{y}|\mathcal{E}(x) + \delta^*)}_{\text{Away Loss}} \right].$$

Then, the original CAT paper (Xhonneux et al., 2024) suggest "cut-off" each loss function $\mathcal{L}'$ as

$$\mathcal{L} = \mathbb{I}[\mathcal{L}'] \cdot 0.999c + (\mathbb{I}[\mathcal{L}' > c] \cdot 0.001 + \mathbb{I}[\mathcal{L}' \leq c]) \cdot \mathcal{L}'$$

To prevent over-optimizing both the toward and away losses, we use a cut-off parameter $c$ and the indicator function $\mathbb{I}[\cdot]$. Xhonneux et al. (2024) originally set the cut-off value as $0.5$ for the toward loss and $-5.0$ for the away loss. However, we have empirically found that such a hyperparameter choice significantly reduces the utility of trained LLMs. Specifically, we calculate the LC-WinRate of a model trained from CAT with a toward cut-off value of $0.5$ and an away cut-off value of $-5.0$ in Table 5. From the table, we find that, for example, the original cut-off setting from Xhonneux et al. (2024) can result in the LC-WinRate of Vicuna being $23.60\%$, which is significantly lower when compared with its original LC-WinRate of $76.86\%$ before finetuning.

As a result, in our experiments, we relax the cut-off values to $1.0$ for the toward loss and $-3.0$ for the away loss in both CAT and ER-CAT to help trained LLMs better preserve their utility.

**LoRA setting.** We use the PEFT library (Mangrulkar et al., 2022) to apply LoRA (Hu et al., 2022) to the embedding layer and all query and key projection matrices in attention layers of LLMs. For the embedding layer, we set its PEFT hyperparmeters as `r=1024`, `lora_alpha=32`, and `lora_dropout=0.1`. Besides, for the remaining layers, we set their PEFT hyperparameters as `r=64`, `lora_alpha=32`, and `lora_dropout=0.1`.

**Adversarial training.** We use AdamW to train each model via CAT in Eq. (4) or our ER-CAT in Eq. (15), where the embedding space perturbation radius $\epsilon$ is fixed to 0.05. To improve the efficiency

of tuning LLMs, LoRA (Hu et al., 2022) is applied to the embedding layer and all query and key projection matrices in attention layers.

**Other AT hyperparameters.** In every AT experiment, we perform LLM AT with AdamW for $60$ iterations, where the learning rate is fixed to $2 \times 10^{-4}$. The batch size is set as $64$, where $8$ samples are adversarial inputs and the remaining $56$ samples are utility inputs. We always use "`Sorry, I can't do that.`" as the safe response for harmful inputs. For the hyperparameter $\alpha$ of CAT, we follow Xhonneux et al. (2024) to set it as $0.5$. For the hyperparameters $\alpha$ and $\beta$ of our ER-CAT, we set them to $0.1$ and $0.5$, respectively.

## C.2 JAILBREAK ATTACKS

We adopt six jailbreak attacks to assess the jailbreak robustness of LLMs. Four of them are token-level suffix attacks, which are: GCG (Zou et al., 2023), BEAST (Sadasivan et al., 2024), GCQ (Hayase et al., 2024), and Zhu's AutoDAN (Zhu et al., 2024). The remaining two are prompt-level attacks, which are: DeepInception (Li et al., 2023) and PAIR (Chao et al., 2023). We re-implemented all six attacks by ourselves to enable efficient and fair jailbreak evaluations. Additionally, for every suffix attack, the length of adversairal suffix token length is set as $20$. Other hyperparameters of jailbreak attacks are set as follows:

- **GCG:** According to Algorithm 1 in Zou et al. (2023), hyperparameters that we need to set for GCG include the iteration number $T$, the top-k parameter $k$, and the "batch-size" $B$. We set $T$ as $500$, $k$ as $256$, and $T$ as $64$.

- **BEAST:** According to Algorithm 1 in Sadasivan et al. (2024), hyperparameters that we need to set for BEAST are two beam-search parameters $k_1$ and $k_2$. We set $k_1$ as $64$ and $k_2$ as $16$.

- **GCQ:** According to Algorithm 1 in Hayase et al. (2024), hyperparameters that we need to set for GCQ include the iteration number $T$, the proxy batch size $b_p$, the query batch size $b_q$, and the buffer size $B$. We set $T = 200$ and $b_p = b_q = B = 128$.

- **Zhu's AutoDAN:** According to Algorithm 1 and Algorithm 2 in Zhu et al. (2024), hyper-parameters that we need to set for Zhu's AutoDAN are the iteration number $T$ in each step, objective weights $w_1$ and $w_2$, the top-$B$ parameter $B$, and the temperature $\tau$. We set $T$ as $3$, $w_1$ as $10$, $w_2$ as $100$, $B$ as $256$, and $\tau$ as $2$.

- **DeepInception:** According to Li et al. (2023), DeepInception attack leverages human-crafted jailbreak prompts to perform attacks. Therefore, no hyperparameter need to be set for DeepInception. Additionally, we use the role play-based prompt from Li et al. (2023) to conduct the attck.

- **PAIR:** According to Chao et al. (2023), PAIR leverages an LLM-based attacker and an LLM-based judger to iteratively synthesize, evaluate, and refine jailbreak prompts. We use Mistral-8x7B-Instruct-v0.1 as the base model for attacker and Llama-3-70B-Instruct as the base model for judger. Besides, the number of iteratively refining is fix to $10$.

## C.3 ADDITIONAL RESULTS

This section collects experimental results that omitted from Section 5.

**Evolutions of singular values of $W^E$.** As illustrated in Section 4.4, our proposed ER-CAT aims to regularize the embedding matrix of trained LLMs so that its singular values be neither too large nor too small. Here we empirically show that the introduced regularization term in ER-CAT can indeed help to achieve such a training goal. Specifically, we plot the maximum singular value, the minimum singular value, the standard deviation of the singular values, and the mean of the singular values of the embedding matrix over the training progress of CAT and ER-CAT in Figure 1. From the figure, we find that when compared with the original CAT method, our ER-CAT can optimize the LLM embedding matrix to: (1) reduce its maximum singular value, (2) increase its minimum singular value, (3) reduce the standard deviation of all its singular values, and (4) do not change the mean of singular values too much (the change of mean is less than $2\%$ on every base model). In other words, the newly introduced regularization term in ER-CAT can make these singular values more concentrated. However, we also notice that the reduction in singular values' standard deviation

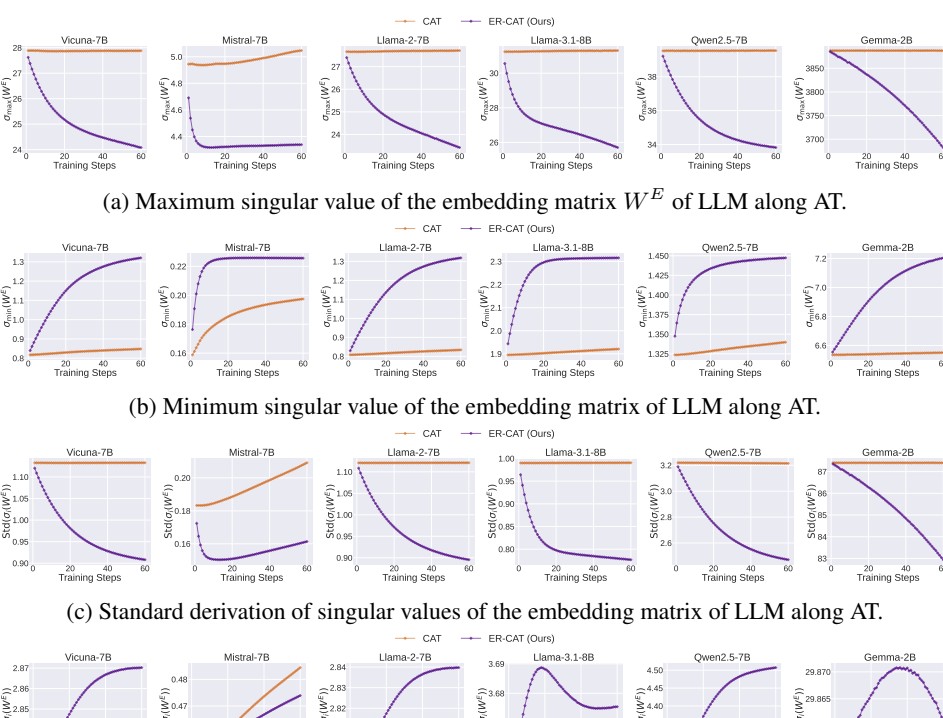

(a) Maximum singular value of the embedding matrix $W^E$ of LLM along AT.

(b) Minimum singular value of the embedding matrix of LLM along AT.

(c) Standard derivation of singular values of the embedding matrix of LLM along AT.

(d) Mean of singular values of the embedding matrix of LLM along AT.

Figure 1: Evolutions of singular values of the embedding matrix of LLMs along AT.

Table 6: 1@5 ASR on different models and attacks. A low ASR indicates a model robustness.

|  |  | GCG | BEAST | GCQ | Zhu's AutoDAN | DeepInception | PAIR |
|---|---|---|---|---|---|---|---|
| Vicuna-7B | Original | 95.0 | 92.0 | 94.0 | 26.0 | 77.0 | 82.0 |
|  | CAT | **31.0** | 31.0 | **14.0** | **2.0** | **22.0** | **14.0** |
|  | ER-CAT (Ours) | 34.0 | **30.0** | 18.0 | 5.0 | 25.0 | 30.0 |
| Mistral-7B | Original | 94.0 | 89.0 | 92.0 | 64.0 | 84.0 | 73.0 |
|  | CAT | 25.0 | **17.0** | 14.0 | 4.0 | 4.0 | **33.0** |
|  | ER-CAT (Ours) | **20.0** | 18.0 | **8.0** | **2.0** | **3.0** | 40.0 |
| Llama-2-7B | Original | 61.0 | 30.0 | 9.0 | 11.0 | 70.0 | 44.0 |
|  | CAT | 45.0 | 30.0 | 15.0 | 8.0 | 24.0 | 25.0 |
|  | ER-CAT (Ours) | **32.0** | **19.0** | **5.0** | **1.0** | **8.0** | **12.0** |
| Llama-3.1-8B | Original | 24.0 | 39.0 | 12.0 | 10.0 | 75.0 | 60.0 |
|  | CAT | **9.0** | **10.0** | **0.0** | **0.0** | **0.0** | 12.0 |
|  | ER-CAT (Ours) | **9.0** | 17.0 | **0.0** | **0.0** | **0.0** | **9.0** |
| Qwen2.5-7B | Original | 86.0 | 86.0 | 82.0 | 26.0 | 90.0 | 63.0 |
|  | CAT | 41.0 | 43.0 | 40.0 | **0.0** | 1.0 | 26.0 |
|  | ER-CAT (Ours) | **30.0** | **31.0** | **14.0** | **0.0** | 4.0 | **21.0** |
| Gemma-2B | Original | 79.0 | 68.0 | 21.0 | 8.0 | 45.0 | 34.0 |
|  | CAT | **40.0** | **27.0** | **8.0** | 1.0 | 1.0 | **10.0** |
|  | ER-CAT (Ours) | **40.0** | **27.0** | 15.0 | 3.0 | **0.0** | **10.0** |

might not be large enough, which means that there is still room for improving the effectiveness of concentrating embedding matrix singular values.

**"Worst-case" (1@5 ASR) robustness analysis.** In Section 5, we leverage the metric Avg@5 ASR to evaluate the jailbreak robustness of LLMs. Here we focus on a more challenging setting to assess LLMs' robustness via the 1@5 ASR metric. Specifically, under the 1@5 ASR metric, each jailbreak prompt needs to be repeatedly fed into the targeted model 5 times. If any of the 5 responses is judged

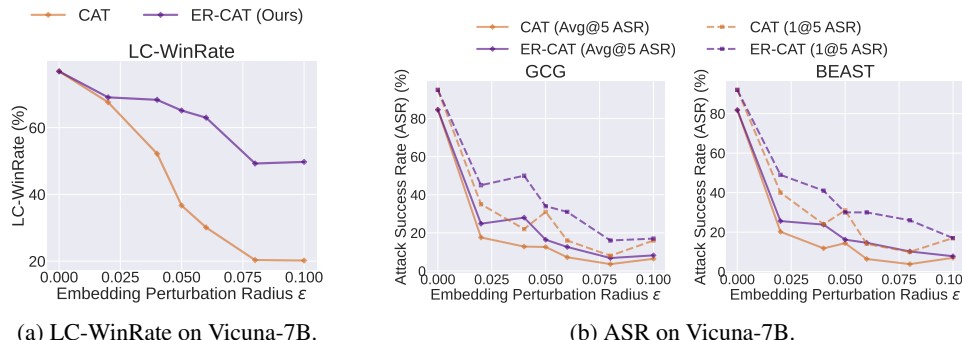

(a) LC-WinRate on Vicuna-7B.  (b) ASR on Vicuna-7B.

Figure 2: Utility (measured by LC-WinRate) and jailbreak robustness (measured by ASR) on Vicunna-7B trained with different embedding perturbation radius $\epsilon$ within the range $[0, 0.1]$. A high LC-WinRate indicates a better utility, while a low ASR indicates a strong jailbreak robustness.

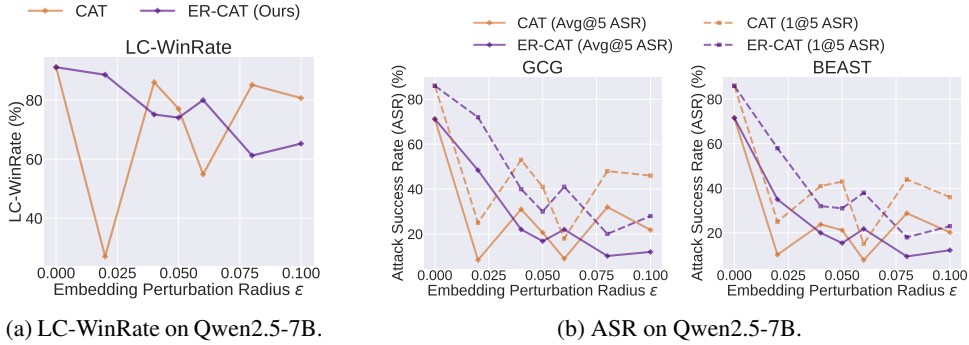

(a) LC-WinRate on Qwen2.5-7B.  (b) ASR on Qwen2.5-7B.

Figure 3: Utility (measured by LC-WinRate) and jailbreak robustness (measured by ASR) on Qwen2.5-7B trained with different embedding perturbation radius $\epsilon$ within the range $[0, 0.1]$. A high LC-WinRate indicates a better utility, while a low ASR indicates a strong jailbreak robustness.

as a harmful response, then this jailbreak prompt is considered to have successfully jailbroken the targeted model. Results of 1@5 ASR are collected and presented in Table 6. From the table, we have similar observations as those for Avg@5 ASR (see Figure 1 in Section 5), *i.e.*, ER-CAT achieves significantly better jailbreak robustness on Llama-2 and Qwen2.5 models, while maintaining similar robustness on Vicuna, Mistral, Llama-3.1 and Gemma models. It is also worth noting that while ER-CAT and CAT achieve similar robustness on Vicuna and Mistral, according to Table 2 in Section 5, the utility achieved by ER-CAT is significantly better than that by CAT.

**Ablation studies on the embedding space perturbation radius $\epsilon$ in AT.** In our main experiments, the embedding space perturbation $\epsilon$ is fixed to $0.05$ for both CAT and our ER-CAT. We now analyze how different radii $\epsilon$ affect the performance of adversarially trained models. Specifically, we train LLMs with different embedding perturbation radii $\epsilon$ from the set $\{0, 0.02, 0.04, 0.05, 0.06, 0.08, 0.1\}$ and then calculate their utility (*i.e.*, LC-WinRate) and robustness (*i.e.*, GCG and BEAST). Preliminary results are shown in Figure 2 for Vicuna-7B and Figure 3 for Qwen2.5-7B. From Figure 2, we observe that as the radius $\epsilon$ increases, ER-CAT maintains similar jailbreak robustness to that of CAT, but the utility of CAT models degenerates rapidly. Besides, from Figure 3 we find that CAT is very sensitive to the change of radius $\epsilon$ while ER-CAT is less sensitive. We also observe that when the radius is large (*i.e.*, $\epsilon > 0.02$), the jailbreak robustness of ER-CAT is better than that of CAT in most cases.

