# OpenReview forum: "Understanding and Improving Continuous LLM Adversarial Training via In-context Learning Theory"
_ICLR.cc/2026/Conference — ICLR 2026 Poster_

### Official Review · Reviewer_3RMS · 2025-10-25

**Soundness:** 3
**Presentation:** 2
**Contribution:** 3
**Rating:** 4
**Confidence:** 3

**Summary:**

The authors investigate the ability of continuous adversarial training (CAT) to improve the robustness of LLMs using in-context learning (ICL) theory. The authors formalize a simplified surrogate problem based on linear transformers trained on in-context linear regression tasks. They prove that the robust generalization bound of models trained with CAT negatively correlates with the perturbation radius in embedding space and depends on the singular values of the embedding matrix. Based on their theoretical observations, they argue that the singular values should be constrained in the embedding matrix and provide a new version of CAT that achieves better accuracy robustness tradeoffs.

**Strengths:**

- The beginning of the paper is easy to follow and motivates the proposed analysis intuitively with prior work. The relevant context is clearly explained and provides a better formal description than the original works in some cases
- Provides the first theoretical analysis of CAT, explaining why embedding-space perturbations can improve robustness to input-space jailbreaks, a contribution missing from the large number of adversarial training works in the LLM space.
- Based on their theory, the authors are able to provide a practical algorithm (even if empirical results are somewhat inconclusive in my opinion)

**Weaknesses:**

- I found the description of the ICL theory hard to follow and would have appreciated additional text motivating / explaining the provided notation. Section 4.1 is hard to read.
- The results provided regarding the performance of ER-CAT are somewhat inconclusive. For LLAMA-2-7B there is no clear advantage visible for ER-CAR. Moreover, it remains unclear if the benefits stem from the regularization approach or just from insufficient hyperparameter tuning of both methods. The large number of hyperparameters in LLM adversarial training appear to make it difficult to compare two appraoches (specifically if results are somewhat close)
- Instead of averaging the ASR for the @5 approach, I would argue it's more sensible to compute the "max" harmfulness over all trials, as we are generally interested in lower bounds on the robustness.
- Could the authors provide an "ensemble ASR" value that describes the lower bound robustness against all attacks (e.g., if one attack out of all succeeds for a prompt it can be considered broken). This would enable easier comparisons between CAT and ER-CAT regarding utility robustness trade-offs
- Assumptions made in the theoretical part could be more clearly highlighted / summarized
- Relevant hyperparameters that are important in the theoretical part are not analyzed (e.g., the perturbation magnitude constraint)

**Questions:**

- The theory provided in the paper provides some connections between the operator norm of the embedding function and the robustness of LLMs. Do the authors think that there is a general connection concerning the Lipschitzness of the function (LLM) and its robustness in the context of adversarial robustness in LLMs? (e.g., similar to [1])
- Limited empirical analysis regarding model properties. How robust are models to continuous attacks? Do the regularized embedding matrices successfully reduce the effect of perturbations? Are the singular values of the final matrices considerably lower? Is there a connection between the magnitude of the singular values and the robustness? Do more robust existing models generally have lower singular values?
- The surrogate setting appears somewhat simplified and not able to capture the complexity of the actual problem. Could the authors motivate why this might not be the case?

[1] Roth, Kevin, Yannic Kilcher, and Thomas Hofmann. "Adversarial training is a form of data-dependent operator norm regularization." Advances in Neural Information Processing Systems 33 (2020): 14973-14985.

---

> ### Author Response · Authors · 2025-11-25
> **Rebuttal Part [1/4]**
>
> Thank you for your thorough comments.
>
> **Q1: I found the description of the ICL theory hard to follow and would have appreciated additional text motivating / explaining the provided notation. Section 4.1 is hard to read.**
>
> **A1:** Thanks. As you suggested, we have added a new Section 4.2 in our revised paper to discuss and bridge the gap between the ICL embedding AT for theoretical LSA-E models and the CAT for real-world LLMs. Concretely, this section explains: (1) why the theoretical LSA-E models are similar to real-world LLMs, (2) why the two AT problems are similar to each other, and (3) why the ICL adversarial robustness assessed by the ICL (suffix) adversarial attacks is similar to LLM jailbreak robustness. Please refer to this new section in the revision for details.
>
> **Q2.1: The results provided regarding the performance of ER-CAT are somewhat inconclusive. For LLAMA-2-7B there is no clear advantage visible for ER-CAT.**
>
> **A2.1:** For Llama-2-7B, the advantage of adopting ER-CAT is two-fold. Firstly, from the robustness analysis in Table 1 in our original paper, one can find that for those "strong" jailbreak attacks such as GCG, BEAST, and PAIR, ER-CAT can realize 7% to 9% lower ASR when compared with CAT. Besides, for those "weak" jailbreak attacks, ER-CAT can maintain their ASRs at almost zero. All these results indicate that ER-CAT can realize stronger jailbreak robustness.
>
> Secondly, from the utility analysis in Table 2 in our original paper, we find that ER-CAT only reduces the LC-WinRate by 2% when compared with the original CAT method. This suggests that ER-CAT significantly improves the robustness at very little cost compared with CAT.
>
> **Q2.2: Moreover, it remains unclear if the benefits stem from the regularization approach or just from insufficient hyperparameter tuning of both methods. The large number of hyperparameters in LLM adversarial training appear to make it difficult to compare two appraoches (specifically if results are somewhat close)**
>
> **A2.2:** Thanks. For the hyperparameters of the original CAT, we directly follow its original paper [r6] to set the $\alpha$ value in Eq.(4) in our original paper as $0.5$. We believe [r6] has already well tuned this $\alpha$ hyperparameter.
>
> Besides, for the new hyperparameter $\beta$ in our ER-CAT, we have conducted a preliminary ablation study on it in our revision. Please see Table 4 in Section 5.2 in the revised paper. We carefully varied $\beta$ with the range $[0,1]$, but our current results show that varying this coefficient $\beta$ does not change the utility or the robustness of the model trained with ER-CAT too much.
>
> In summary, we believe our updated experiments have already performed sufficient hyperparameter tuning.
>
> **Q3: Instead of averaging the ASR for the @5 approach, I would argue it's more sensible to compute the "max" harmfulness over all trials, as we are generally interested in lower bounds on the robustness.**
>
> **A3:** Thanks for your suggestion. We have computed the "max" harmfulness over all trials, denoted as "1@5 ASR", in our revised paper. Please see Table 6 in Appendix C.3 in the revision.
>
> From this new table and combined with our previous utility analysis in Section 5, we observe that: (1) ER-CAT maintains similar utility as that of CAT on Llama-2-7B and Qwen2.5-7B models, but achieves significantly better 1@5 ASR, and (2) ER-CAT maintains similar 1@5 ASR as that of CAT on Vicuna-7B and Mistral-7B models, but achieves significantly better utility.
>
> **Q4: Could the authors provide an "ensemble ASR" value that describes the lower bound robustness against all attacks (e.g., if one attack out of all succeeds for a prompt it can be considered broken). This would enable easier comparisons between CAT and ER-CAT regarding utility robustness trade-offs.**
>
> **A4:** Thank you for your suggestion. We commit to reporting such an ensemble ASR in our final paper but not for now due to the limited rebuttal period. Specifically, since we did not record the jailbreak judging results for individual jailbreak prompts, we will need to re-evaluate every jailbreak prompt synthesized by all six jailbreak attacks, which is time-consuming (we can report "1@5 ASR" in **A3** simply because we have previously calculated it). Nevertheless, we deduce one would have similar observations from the "ensemble ASR" as those of the "1@5 ASR" in **A3**, i.e., ER-CAT can maintain utility while improving robustness or maintain robustness while improving utility.

---

> ### Author Response · Authors · 2025-11-25
> **Rebuttal Part [2/4]**
>
> **Q5: Assumptions made in the theoretical part could be more clearly highlighted / summarized.**
>
> **A5:** Firstly, we respectfully note that we only explicitly made a single assumption, i.e., **Assumption 1** in the original Section 4.2. This Assumption 1 is widely adopted in existing ICL theoretical studies. The idea behind it is to ensure the symmetric initialization of the model parameters of ICL linear transformers. While the motivation for Assumption 1 was explained in our original paper, we have reorganized the explanation as **Remark 1** in the new Section 4.3 in the revision. Please refer to our revised paper for details.
>
> Besides, in our Theorem 2 in the original Section 4.2, we additionally require that the embedding space dimension $d$ of the LSA-E model be no larger than the input in-context sample dimension $d\_0$. Such a requirement means that the LSA-E models implicitly compress the input data. We have added a new **Remark 2** in the new Section 4.3 to illustrate this. Please also refer to our revised paper for details.
>
> **Q6: Relevant hyperparameters that are important in the theoretical part are not analyzed (e.g., the perturbation magnitude constraint)**
>
> **A6:** For the embedding perturbation radius $\epsilon$, we respectfully note that how to appropriately choose it in practice is still an open problem, and our empirical analysis just follows existing studies on continuous LLM adversarial training [r6, r7] to set it to a common value of $0.05$. We feel that addressing the hyperparameter tuning problem for this radius $\epsilon$ might be beyond the scope of our study, as our paper focuses on theoretically explaining the mechanism behind the empirical success of LLM CAT, i.e., explaining why adversarial perturbation in the token embedding space helps LLM defend against adversarial inputs from the original token space.
>
> Besides, for another hyperparameter $\beta$ in ER-CAT, we have conducted a preliminary ablation study on it in our revision. Please see Table 4 in Section 5.2 in the revised paper. Our current results show that varying this coefficient $\beta$ does not change the utility or the robustness of the model trained with ER-CAT too much. We deduce that this is because we use the AdamW optimizer to perform the training, and the gradient normalization procedure in AdamW implicitly performs reweighting on the ER-CAT objective to mitigate the effect of tuning coefficients for different terms.
>
> **Q7: The theory provided in the paper provides some connections between the operator norm of the embedding function and the robustness of LLMs. Do the authors think that there is a general connection concerning the Lipschitzness of the function (LLM) and its robustness in the context of adversarial robustness in LLMs? (e.g., similar to [r1])**
>
> **A7:** Thanks for this good question. We indeed believe that the Lipschitzness of LLMs and the jailbreak robustness (adversarial robustness) of LLMs are related. Specifically, in previous studies on the adversarial robustness of image classification models, many works have already found that a network with a low Lipschitz constant can enjoy strong certified/provable robustness against adversarial images [r2, r3, r4]. As a result, it would be expected that an LLM with a bounded Lipschitz constant would also enjoy robustness (against jailbreak attacks) to some extent. We will leave the study on the relationship between the Lipschitzness and robustness of LLMs for future work.

---

> > ### Author Response · Authors · 2025-11-25
> > **Rebuttal Part [3/4]**
> >
> > **Q8.1: Limited empirical analysis regarding model properties. How robust are models to continuous attacks?**
> >
> > **A8.1:** Thanks for your suggestion. We commit to adding robustness evaluation with continuous attacks in the final version of our paper, but not for now due to the limited rebuttal period. Nevertheless, we would also like to note that discrete jailbreak attacks more commonly exist in real-world scenarios than continuous attacks, and that is why we think we should put more effort into analyzing these practical discrete jailbreak attacks.
> >
> > **Q8.2: Do the regularized embedding matrices successfully reduce the effect of perturbations?**
> >
> > **A8.2:** Thanks. If the "perturbations" here mean the perturbations in the token space, then the answer to this question is YES, as our experiments in Section 5 indeed show that ER-CAT can improve LLM robustness compared to CAT.
> >
> > However, if the "perturbations" here mean perturbations in the embedding space, then we would like to respectfully argue that the goal of regularizing the LLM embedding matrix in ER-CAT is not to reduce the effect of embedding space perturbations. In fact, its goal is to improve the robustness of LLMs against inputs from their original token space. The idea behind this follows from our Theorem 2.
> >
> > **Q8.3: Are the singular values of the final matrices considerably lower?**
> >
> > **A8.3:** We respectfully argue that, as explained in Section 4.4 of our original paper, our ER-CAT method aims to make the singular values of the embedding matrix of trained LLMs **neither too large nor too small**. In other words, **our ER-CAT does not encourage all singular values to be small**.
> >
> > Besides, we have visualized the evolution of the embedding matrix's singular values during training in Figure 1 of Appendix C.3 in our revised paper. Specifically, we plotted the maximum singular value, the minimum singular value, and the standard deviation of the singular values of the embedding matrix over the training progress of CAT and ER-CAT. We find that, compared with the original CAT method, our ER-CAT can optimize the LLM embedding matrix to: (1) reduce its maximum singular value, (2) increase its minimum singular value, and (3) reduce the standard deviation of all its singular values. All of these findings suggest that the newly introduced regularization term in ER-CAT can indeed make these singular values more concentrated.
> >
> > **Q8.4: Is there a connection between the magnitude of the singular values and the robustness?**
> >
> > **A8.4:** Thanks. Our proven robust generalization upper bound in Theorem 2 shows that the magnitude of singular values of the LLM embedding matrix should not be too large or too small to lead to better LLM jailbreak robustness. Following this idea, we designed an ER-CAT algorithm for real-world LLMs that performs LLM AT by concentrating the singular values of the LLM embedding matrix. Experiments on LLMs show that ER-CAT can indeed improve LLM jailbreak robustness (especially on Llama-2-7B and Qwen2.5-7B), which therefore empirically justifies the connection between the magnitude of singular values and LLM robustness.
> >
> > **Q8.5: Do more robust existing models generally have lower singular values?**
> >
> > **A8.5:** Thanks. Firstly, we would like to again respectfully note that we did not say that lower singular values of the LLM embedding matrix lead to stronger jailbreak robustness. Our proven robust generalization bound in Theorem 2 in Section 4 suggests that to achieve better jailbreak robustness, the singular values of the LLM embedding matrix should not be too large or too small. Besides, the introduced regularization term in our proposed ER-CAT aims to make the singular values of the LLM embedding matrix concentrate toward each other.
> >
> > Secondly, we note that for different model families, the magnitude of their embedding matrix singular values may also be very different. For example, in our new Figure 1(a) in Appendix C.3 in the revision, the maximum singular value of the base model Mistral-7B can be as low as around 5.0, while for Gemma-2B it can be as large as almost 4,000. Therefore, we are afraid that it might be inappropriate to directly compare the magnitude of embedding matrix singular values across different types of LLMs.
> >
> > Finally, we would also like to note that our paper is the very first work to connect the singular values of the embedding matrix of LLMs with jailbreak robustness. Before our paper, there were few studies on LLM jailbreak robustness that reported the magnitude of embedding matrix singular values. While we cannot know what the magnitude of the embedding matrix singular values in previous robust models is, we hope that our paper can let the community focus more on analyzing the embedding matrix when studying jailbreak robustness.

---

> > > ### Author Response · Authors · 2025-11-25
> > > **Rebuttal Part [4/4]**
> > >
> > > **Q9: The surrogate setting appears somewhat simplified and not able to capture the complexity of the actual problem. Could the authors motivate why this might not be the case?**
> > >
> > > **A9:** Thanks for this good question. Firstly, we note that to conduct a robust generalization analysis for the original ICL embedding AT, the key step is to obtain the closed-form solution of a LSA-E model that enjoys **a small AT loss $L^{\mathrm{adv}}\_{\mathrm{LSAE}}(\theta)$**. As explained in Lemma 1, the surrogate AT loss $\tilde L^{\mathrm{adv}}\_{\mathrm{LSAE}}(\theta)$ in the surrogate ICL embedding AT is a closed-form upper bound of the original AT loss $L^{\mathrm{adv}}\_{\mathrm{LSAE}}(\theta)$. Therefore, since the closed-form optimal model parameter $\theta_*$ in Theorem 1 is the minimizer of the surrogate AT loss $\tilde L^{\mathrm{adv}}\_{\mathrm{LSAE}}(\theta)$, its corresponding original AT loss $L^{\mathrm{adv}}\_{\mathrm{LSAE}}(\theta\_\*)$ should also be small due to the fact that $L^{\mathrm{adv}}\_{\mathrm{LSAE}}(\theta\_\*) \leq \tilde L^{\mathrm{adv}}\_{\mathrm{LSAE}}(\theta\_\*)$. In other words, the optimal LSA-E model $f\_{\mathrm{LSAE}, \theta\_*}$ obtained from the surrogate AT problem also enjoys a small AT loss in the original AT problem and thus is suitable for our theoretical robustness generalization analysis of ICL embedding AT.
> > >
> > > Secondly, while the robust generalization upper bound in Theorem 2 is proved for the surrogate problem, this bound is empirically justified by our experiments on real-world LLMs and jailbreak attacks in Section 5. Specifically, the main observation from this bound is that the robustness of LSA-E models/LLMs trained with embedding AT is closely related to the eigenvalues of the embedding matrix $W^E$. For the trained model to have strong adversarial robustness, the eigenvalues of $W^E$ should be neither too large nor too small. Based on this observation, we design an improved CAT algorithm for LLMs, named ER-CAT, by enforcing the eigenvalues of $W^E$ to concentrate during training. Experimental results in Table 1 in Section 5 show that ER-CAT improved the jailbreak robustness of real-world LLMs, which justifies our theoretical findings. This indicates that our theoretical surrogate setting is a good characterization of real-world complex AT settings.
> > >
> > > Thirdly, we would also like to note that [r5] leverages a similar surrogate setting (i.e., theoretically studying the optimization of an upper bound of the original AT loss) to analyze AT for LLMs. According to [r5], the obtained robust generalization upper bound is also empirically justified on real-world LLMs against jailbreak attacks. As a result, we believe this surrogate technique is strong enough for us to analyze the robust generalization of the original ICL embedding AT as well as the real-world LLM CAT.
> > >
> > > **References**
> > >
> > > [r1] Roth, Kevin, Yannic Kilcher, and Thomas Hofmann. "Adversarial training is a form of data-dependent operator norm regularization." Advances in Neural Information Processing Systems 33 (2020): 14973-14985.
> > >
> > > [r2] Meunier et al. A Dynamical System Perspective for Lipschitz Neural Networks. ICML 2022.
> > >
> > > [r3] Singla et al. Improved deterministic l2 robustness on CIFAR-10 and CIFAR-100. ICLR 2022.
> > >
> > > [r4] Zhang et al. Rethinking Lipschitz Neural Networks and Certified Robustness: A Boolean Function Perspective. NeurIPS 2022.
> > >
> > > [r5] Fu et al. Short-length Adversarial Training Helps LLMs Defend Long-length Jailbreak Attacks: Theoretical and Empirical Evidence. NeurIPS 2025.
> > >
> > > [r6] Xhonneux et al. Efficient Adversarial Training in LLMs with Continuous Attacks. NeurIPS 2024.
> > >
> > > [r7] Dékány et al. MixAT: Combining Continuous and Discrete Adversarial Training for LLMs. NeurIPS 2025.

---

> > > > ### Comment · Reviewer_3RMS · 2025-11-25
> > > > **Thanks for the response**
> > > >
> > > > I thank the authors for their detailed response.
> > > >
> > > > The majority of my concerns have been addressed, and additional experiments have been provided in some cases (e.g., analyzing singular values in more detail, which has also been a concern of other reviewers or worst-case 1@5 ASR).
> > > >
> > > > Some misunderstandings on my side have also been clarified, and I thank the authors for their patience.
> > > >
> > > > Under the assumption that the final paper will include:
> > > > - an analysis of worst-case robustness against all attacks (ensemble)
> > > > - a sanity check regarding the robustness against continuous attacks
> > > >
> > > > I raised my score to 6.
> > > >
> > > > Why I did not raise my score further: I understand that conducting more hyperparameter tuning can be expensive, but for a higher score, I would expect the authors to conduct more in-depth studies on the effectiveness of different methods across a range of hyperparameters. In machine learning, gains have often been misattributed to new methods when, in reality, differences were the result of insufficient tuning. Given this history, I do not think it is sufficient to simply adopt fixed hyperparameter values from a previous paper (if this paper does not provide a detailed study on hyperparameters, which is the case for [r6]).

---

> > > > > ### Author Response · Authors · 2025-11-26
> > > > > **Thank you for your feedback & Further response**
> > > > >
> > > > > Thank you very much for your feedback and kind support! We commit that the experiments of (1) ensemble ASR and (2) continuous attack will be added in over final revision.
> > > > >
> > > > > Besides, after reading your and **Reviewer#hwXR**'s follow-up comments, we now realize that it is indeed important and meaningful to analyze **how different embedding space perturbation radii $\epsilon$ affect the performance of adversarially trained models**. To this end, **we conducted a preliminary ablation study on this with Vicuna-7B and Qwen2.5-7B in our latest revision**. Please see Figure 2 (for Vicuna-7B) and Figure 3 (for Qwen2.5-7B) in Appendix C.3 of the revision.
> > > > >
> > > > > Our current results show that: (1) For Vicuna-7B, when the radius $\epsilon$ becomes large, ER-CAT maintains jailbreak robustness similar to CAT, but the utility of CAT significantly decreases. (2) For Qwen2.5-7B, varying the radius $\epsilon$ can cause the utility of CAT to fluctuate significantly, and ER-CAT can achieve better jailbreak robustness in most settings. Additionally, these figures show that the chosen radius of 0.05 in our main experiments balances utility and robustness well for ER-CAT, i.e., making the utility of ER-CAT not too weak and the ASR achieved by ER-CAT not too high.

---

### Official Review · Reviewer_iaCj · 2025-10-30

**Soundness:** 3
**Presentation:** 3
**Contribution:** 3
**Rating:** 6
**Confidence:** 3

**Summary:**

This paper addresses the problem of Continuous Adversarial Training (CAT) for LLMs, an efficient defense against jailbreak attacks that operates in the continuous embedding space. While CAT is empirically effective, the theoretical reason why perturbations in the embedding space can confer robustness against attacks in the discrete token space has remained unknown.

The authors provide the first theoretical analysis to bridge this gap. They leverage In-Context Learning (ICL) theory, modeling the problem using a Linear Self-Attention with Embedding (LSA-E) model trained on linear regression tasks. Their key theoretical contribution is a robust generalization bound for this model. This bound demonstrates that the model's robust risk (against input-space perturbations) is negatively correlated with the perturbation radius (epsilon) used during embedding-space AT. This result provides a formal explanation for CAT's effectiveness.

Furthermore, the bound reveals that model robustness is closely tied to the singular values of the embedding matrix. Based on this insight, the authors propose a novel method, Embedding-Regularized Continuous AT (ER-CAT), which adds a regularization term to the CAT objective to minimize the variance of the embedding matrix's singular values. Empirical results on six real-world LLMs and six jailbreak attacks show that ER-CAT generally achieves a better robustness-utility tradeoff than standard CAT.

**Strengths:**

**Originality**: The paper's primary contribution is highly original. It tackles a novel and important theoretical question (i.e., "Why does CAT work?") that sits at the intersection of LLM robustness and efficiency. The choice of using ICL as the analytical framework is creative and non-trivial, establishing a new and insightful link between these fields. The development of a new defense (ER-CAT) directly from the theoretical findings is a good example of theory-driven research.

**Significance**: The work is of high significance to the ICLR community. As LLMs become more pervasive, efficient and principled defense mechanisms are critical. CAT is a pragmatically important method, and this paper provides the first solid theoretical grounding for it. This understanding can lead to more principled improvements beyond simple heuristics, strengthening the foundation for future work on robust LLMs.

**Clarity**: The paper is well-written. The abstract and introduction clearly articulate the problem, the gap in existing knowledge, and the paper's contributions. The logical flow from the core research question to the theoretical setup, the key findings (Theorem 2), the implications (Sec 4.3), and the proposed method (ER-CAT, Eq. 15) is clear and easy to follow.

**Weaknesses:**

**The Theory-Practice Gap**: The primary weakness of this paper is the gap between the theoretical model and the practical application. The analysis relies on a Linear Self-Attention (LSA-E) model trained on a linear regression task. This is a massive simplification of a multi-billion parameter, highly non-linear Transformer LLM performing the complex task of "refusal." While the authors provide some justification (Sec 4.1), the leap of faith required is substantial. The paper would be strengthened by a more explicit discussion of these limitations and the potential risks of generalizing these linear insights to the non-linear deep learning regime.


**Missing Ablation for New Hyperparameter**: The proposed ER-CAT method introduces a new hyperparameter, beta (the coefficient for the singular value regularization), which is set to 0.2. The paper provides no ablation study or sensitivity analysis for this hyperparameter. How was this value chosen? Is the method's performance robust to changes in beta? This is a crucial missing piece of the empirical validation.

**Questions:**

1. How was the regularization coefficient beta = 0.2 for ER-CAT selected? Could you provide a sensitivity analysis or ablation study on at least one model to show how performance (both ASR and LC-WinRate) changes with different values of beta?

2. What is the practical computational overhead (e.g., increase in training time per step/epoch) of calculating and backpropagating through the singular value variance regularization in ER-CAT compared to the standard CAT baseline?

---

> ### Author Response · Authors · 2025-11-24
> **Rebuttal Part [1/2]**
>
> Thank you for your thorough comments and kind support!
>
> **Q1: The Theory-Practice Gap: The primary weakness of this paper is the gap between the theoretical model and the practical application. The analysis relies on a Linear Self-Attention (LSA-E) model trained on a linear regression task. This is a massive simplification of a multi-billion parameter, highly non-linear Transformer LLM performing the complex task of "refusal." While the authors provide some justification (Sec 4.1), the leap of faith required is substantial. The paper would be strengthened by a more explicit discussion of these limitations and the potential risks of generalizing these linear insights to the non-linear deep learning regime.**
>
> **A1:** Thanks for your suggestion. We have added a new Section 4.2 in our revised paper to discuss and bridge the gap between the ICL embedding AT for theoretical LSA-E models and the CAT for real-world LLMs, as you suggested. Please refer to this new section for details.
>
> Besides, your specific concerns about the theory-practice gap are carefully addressed as follows:
>   - **Regarding the gap between simple linear transformers (i.e., LSA-E models) and highly non-linear LLMs.** [r1] has empirically shown that the linear self-attention module in LSA-E models shares similar properties with the non-linear ones in LLMs and thus is useful for theoretical understandings of LLMs. Furthermore, [r2] has already succeeded in using simple single-layer linear transformers to explain the robustness of LLMs in certain aspects. As a result, while LSA-E models are significantly simplified when compared with LLMs, we believe they are still very good artifacts for theoretically understanding the properties of LLMs.
>
>   - **Regarding the gap between the simple ICL linear regression task for LSA-E models and the complex "refusal" task for real-world LLMs.** We note that [r2] also uses this ICL linear regression task to theoretically analyze the robustness of adversarially trained ICL models, and the theoretical finding in [r2] is empirically justified on real-world "refusal" task-based LLM AT. Therefore, we believe such a gap would not be a major issue.
>
> **Q2: Missing Ablation for New Hyperparameter: The proposed ER-CAT method introduces a new hyperparameter, beta (the coefficient for the singular value regularization), which is set to 0.2. The paper provides no ablation study or sensitivity analysis for this hyperparameter. How was this value chosen? Is the method's performance robust to changes in beta? This is a crucial missing piece of the empirical validation.**
>
> **A2:** Thanks. We apologize that the regularization coefficient $\beta$ should be $0.5$ and not $0.2$. It was a typo in our original paper and has been fixed in the revision.
>
> Besides, follow your suggestion, we have conducted a preliminary ablation study on it in our revision. Results are presented in Table 4 in Section 5.2 in the revised paper. Our current results show that varying this coefficient $\beta$ does not change the utility or the robustness of the model trained with ER-CAT too much. We deduce that this is because we use the AdamW optimizer to perform the training, and the gradient normalization procedure in AdamW implicitly performs reweighting on the ER-CAT objective to mitigate the effect of tuning coefficients for different terms.
>
>
> **Q3: How was the regularization coefficient beta = 0.2 for ER-CAT selected? Could you provide a sensitivity analysis or ablation study on at least one model to show how performance (both ASR and LC-WinRate) changes with different values of beta?**
>
> **A3:** We apologize again that the regularization coefficient $\beta$ should be $0.5$ and not $0.2$. It was a typo in our original paper and has been fixed in the revision. Besides, we have added preliminary ablation studies on this coefficient $\beta$. Please see **A2** for details.

---

> > ### Author Response · Authors · 2025-11-24
> > **Rebuttal Part [2/2]**
> >
> > **Q4: What is the practical computational overhead (e.g., increase in training time per step/epoch) of calculating and backpropagating through the singular value variance regularization in ER-CAT compared to the standard CAT baseline?**
> >
> > **A4:** Thanks. We have added an empirical (relative) time cost analysis for ER-CAT. Specifically, we have collected and presented the time cost of performing CAT and ER-CAT on different base models in the following Table r1:
> >
> > |               | Vicuna-7B | Mistral-7B | Llama-2-7B | Llama-3.1-8B | Qwen2.5-7B | Gemma-2B |
> > | ------------- | --------- | ---------- | ---------- | ------------ | ---------- | -------- |
> > | CAT           | 987.81    | 933.00     | 934.16     | 904.59       | 801.42     | 335.99   |
> > | ER-CAT (ours) | 1074.87   | 1052.12    | 1100.33    | 1094.90      | 1004.30    | 456.81   |
> >
> > *Table r1: Time cost (s) on different AT methods on different models.*
> >
> > From Table r1, we find that ER-CAT only increases the time cost by 100 to 200 seconds when compared with the original CAT method. This suggests that the relative time overhead of ER-CAT is low.
> >
> > This time cost analysis has also been added to our revised paper. Please refer to Section 5.2 in the revision for details.
> >
> > **References**
> >
> > [r1] Ahn et al. Linear attention is (maybe) all you need (to understand transformer optimization). ICLR 2024.
> >
> > [r2] Fu et al. Short-length Adversarial Training Helps LLMs Defend Long-length Jailbreak Attacks: Theoretical and Empirical Evidence. NeurIPS 2025.

---

> > > ### Comment · Reviewer_iaCj · 2025-11-25
> > >
> > > I thank the authors for the detailed response. The majority of my concerns and questions have been addressed, so I've raised my score accordingly.

---

> > > > ### Author Response · Authors · 2025-11-26
> > > > **Thanks for your support and suggestion!**
> > > >
> > > > Thank you very much for recognizing our contributions and kind support!

---

### Official Review · Reviewer_hwXR · 2025-10-31

**Soundness:** 2
**Presentation:** 3
**Contribution:** 3
**Rating:** 4
**Confidence:** 3

**Summary:**

This paper performs a theoretical analysis on the Continuous Adversarial Training (CAT) method for defending LLMs against jailbreak attacks. Building upon ICL theory, the authors leverage a systematic derivation using a linear Transformer framework and propose a deference approach, termed Embedding-Regularized Continuous Adversarial Training (ER-CAT).

**Strengths:**

1. Improving Adversarial Training (AT) efficiency through embedding-space perturbations offers a promising and theoretically grounded direction for enhancing model robustness.
2. The authors provide a comprehensive theoretical proof under the in-context learning (ICL) framework, establishing clear connections between embedding-space adversarial perturbations and robustness in input space. Based on this analysis, they ultimately propose a new CAT method.
3. ER-CAT extends CAT by introducing an additional embedding regularization term, which constrains the variance of singular values in the embedding matrix. It can be seamlessly implemented within existing CAT training pipelines.

**Weaknesses:**

1. The theoretical analysis is derived using Linear Self-Attention (LSA) models. While this is a common in ICL, the gap between the linearized setting and real large-scale LLMs is neither fully explored nor quantitatively characterized. A discussion or empirical validation of how the derived theory translates to practical LLM is preferred.
2. The paper does not report ablation studies on key hyperparameters. For instance, it remains unclear how sensitive the results are to the regularization strength (β) and perturbation radius (ε).
3. All evaluations are conducted using known jailbreak attack benchmarks. It is unclear for other types of attacks.
4. One of CAT’s main motivations is to improve training efficiency over standard adversarial training. However, ER-CAT’s computational overhead relative to CAT is not analyzed.
5. Since the proposed regularization explicitly constrains the singular value distribution of the embedding matrix, it would be insightful to show how these singular values evolve during training. A visualization could empirically support the theoretical claim.

**Questions:**

See the Weaknesses part.

---

> ### Author Response · Authors · 2025-11-25
> **Rebuttal Part [1/2]**
>
> Thank you for your thorough comments.
>
> **Q1: The theoretical analysis is derived using Linear Self-Attention (LSA) models. While this is a common in ICL, the gap between the linearized setting and real large-scale LLMs is neither fully explored nor quantitatively characterized. A discussion or empirical validation of how the derived theory translates to practical LLM is preferred.**
>
> **A1:** Thanks for your suggestion. We have added a new Section 4.2 in our revised paper to discuss and bridge the gap between the ICL embedding AT for theoretical LSA-E models and the CAT for real-world LLMs, as you suggested. Please refer to this new section for details.
>
> Besides, for your specific concerns about **the gap between simple linear transformers (i.e., LSA-E models) and real large-scale LLMs**, we note that [r1] has empirically shown that the linear self-attention module in LSA-E models shares similar properties with the non-linear ones in LLMs and thus is useful for the theoretical understanding of LLMs. [r2] has further succeeded in using simple single-layer linear transformers to explain the robustness of LLMs in certain aspects. As a result, while LSA-E models are significantly simplified when compared with real large-scale LLMs, we believe they are still very good artifacts for theoretically understanding the properties of LLMs.
>
> Furthermore, our new theory-inspired ER-CAT algorithm for LLMs also empirically justifies that LSA-E models are suitable for theoretically studying the AT problems of real-world LLMs. Specifically, our main theoretical result, i.e., the robust generalization upper bound in Theorem 2, suggests that the robustness of LSA-E models trained with embedding AT is closely related to the eigenvalues of the embedding matrix $W^E$. For the trained model to have strong adversarial robustness, the eigenvalues of $W^E$ should be neither too large nor too small. Based on this observation, we designed our ER-CAT algorithm for LLMs by enforcing the eigenvalues of their embedding matrices to concentrate during training. Experimental results in Table 1 in Section 5 show that ER-CAT improved the jailbreak robustness of real-world LLMs, which in turn justifies our theoretical findings. This indicates that our LSA-E models are good characterizations of real-world complex LLMs.
>
> **Q2: The paper does not report ablation studies on key hyperparameters. For instance, it remains unclear how sensitive the results are to the regularization strength (β) and perturbation radius (ε).**
>
> **A2:** Thanks. For the embedding regularization term $\beta$ in ER-CAT, we have conducted a preliminary ablation study on it in our revision. Please see Table 4 in Section 5.2 in the revised paper. Our current results show that varying this coefficient $\beta$ does not change the utility or the robustness of the model trained with ER-CAT too much. We deduce that this is because we use the AdamW optimizer to perform the training, and the gradient normalization procedure in AdamW implicitly performs reweighting on the ER-CAT objective to mitigate the effect of tuning coefficients for different terms.
>
> Besides, for the embedding perturbation radius $\epsilon$, we respectfully note that how to appropriately choose it in practice is still an open problem, and our empirical analysis just follows existing studies on continuous LLM adversarial training [r3, r4] to set it to a common value of $0.05$. We feel that addressing the hyperparameter tuning problem for this radius $\epsilon$ might be beyond the scope of our study, as our paper focuses on theoretically explaining the mechanism behind the empirical success of LLM CAT, i.e., explaining why adversarial perturbation in the token embedding space helps LLM defend against adversarial inputs from the original token space.
>
> **Q3: All evaluations are conducted using known jailbreak attack benchmarks. It is unclear for other types of attacks.**
>
> **A3:** We respectfully argue that the five jailbreak attacks adopted in our experiments are representative attacks that have been widely used for evaluating/benchmarking LLMs' jailbreak robustness in a variety of existing LLM AT literature such as [r2, r3, r4]. Since our paper aims to theoretically analyze the mechanism of LLM AT **but not to design new jailbreak attacks**, we believe it is appropriate for us to use known and **representative** jailbreak attacks to assess trained LLMs' jailbreak robustness.
>
> Additionally, we would also like to note that the adopted five attacks have already covered the major two families of jailbreak attacks, i.e., the token-level family (GCG, BEAST, GCQ, and Zhu's AutoDAN) and the prompt-level family (DeepInception and PAIR).

---

> ### Author Response · Authors · 2025-11-25
> **Rebuttal Part [2/2]**
>
> **Q4: One of CAT’s main motivations is to improve training efficiency over standard adversarial training. However, ER-CAT’s computational overhead relative to CAT is not analyzed.**
>
> **A4:** Thanks. We have performed an additional (relative) time cost analysis for ER-CAT. Specifically, we have collected and presented the time cost of performing CAT and ER-CAT on different base models in the following Table r1:
>
> |               | Vicuna-7B | Mistral-7B | Llama-2-7B | Llama-3.1-8B | Qwen2.5-7B | Gemma-2B |
> | ------------- | --------- | ---------- | ---------- | ------------ | ---------- | -------- |
> | CAT           | 987.81    | 933.00     | 934.16     | 904.59       | 801.42     | 335.99   |
> | ER-CAT (ours) | 1074.87   | 1052.12    | 1100.33    | 1094.90      | 1004.30    | 456.81   |
>
> *Table r1: Time cost on different AT methods on different models.*
>
> From Table r1, we find that ER-CAT only increases the time cost by 100 to 200 seconds when compared with the original CAT method. This suggests that the relative time overhead of ER-CAT is low.
>
> The time cost analysis has also been added to our revised paper. Please refer to Section 5.2 in the revision for details.
>
> **Q5: Since the proposed regularization explicitly constrains the singular value distribution of the embedding matrix, it would be insightful to show how these singular values evolve during training. A visualization could empirically support the theoretical claim.**
>
> **A5:** Thanks for your suggestion. We have visualized the evolution of the embedding matrix's singular values during training, as you suggested. Please see Figure 1 in Appendix C.3 in our revised paper.
>
> Specifically, we plotted the maximum singular value, the minimum singular value, and the standard deviation of the singular values of the embedding matrix over the training progress of CAT and ER-CAT. We find that, compared with the original CAT method, our ER-CAT can optimize the LLM embedding matrix to: (1) reduce its maximum singular value, (2) increase its minimum singular value, and (3) reduce the standard deviation of all its singular values. In other words, the newly introduced regularization term in ER-CAT can make these singular values more concentrated.
>
> **References**
>
> [r1] Ahn et al. Linear attention is (maybe) all you need (to understand transformer optimization). ICLR 2024.
>
> [r2] Fu et al. Short-length Adversarial Training Helps LLMs Defend Long-length Jailbreak Attacks: Theoretical and Empirical Evidence. NeurIPS 2025.
>
> [r3] Xhonneux et al. Efficient Adversarial Training in LLMs with Continuous Attacks. NeurIPS 2024.
>
> [r4] Dékány et al. MixAT: Combining Continuous and Discrete Adversarial Training for LLMs. NeurIPS 2025.

---

> ### Comment · Reviewer_hwXR · 2025-11-26
> **Response to authors**
>
> Dear authors,
>
> Thank you for your detailed responses. I appreciate that, in your response, the motivation and assumptions are now explained more clearly (in Q1 and Q3). However, I would like to respectfully raise the following concerns (again) regarding the experimental setup.
>
> 1. In the experiments, the embedding perturbation radius $\epsilon$ is set to 0.05, but this choice feels slightly biased toward a particular setting. I have expected at least some ablation results or variants. For example, in [r1, Table 2] they use four  $\epsilon$ values for different datasets, which suggests  $\epsilon$ is not fully consistent that will have a **non-trivial effect**.
>
> 2. From the current writing, **it is not completely clear which one you treat as the main CAT baseline**. I will assume that is [r1] as Eq.(4) in your paper. Please note that in [r1] there are two related methods, CAT and CAPO. Could you please clarify exactly which variant you compare against? Also, if some of your experimental settings are aligned with [r1] (like CAT), then it would be good just to state this explicitly and please consider to **report your results under exactly the same conditions**. That would make the comparison more **apple-to-apple**. Also, whether the Table 3 time in your paper comes directly from [r1] or from your own implementation?
>
> 3. For the plots of the singular value distribution, I suggest adding average value just to provide a complete version.
>
> 4. In some places (e.g., Table 1 caption ) the paper says “A high ASR indicates a strong model robustness.” However, should it be, a lower ASR  means the model is more robust?
>
> [r1] Xhonneux et al. Efficient Adversarial Training in LLMs with Continuous Attacks. NeurIPS 2024.

---

> ### Author Response · Authors · 2025-11-26
> **Thank you for your feedback & Further response (1/2)**
>
> Dear Reviewer#hwXR,
>
> Thank you very much for your kind feedback! We are glad that our rebuttal addressed part of your concerns.
>
> For your remaining/additional concerns, we would like to further address them as follows:
>
> **Q6: In the experiments, the embedding perturbation radius is set to 0.05, but this choice feels slightly biased toward a particular setting. I have expected at least some ablation results or variants. For example, in [r1, Table 2] they use four values for different datasets, which suggests is not fully consistent that will have a non-trivial effect.**
>
> **A6:** Thanks. After reading your and **Reviewer#3RMS**'s follow-up comments, we now realize that it is indeed important and meaningful to analyze **how different embedding space perturbation radii $\epsilon$ affect the performance of adversarially trained models**. To this end, **we conducted a preliminary ablation study on this with Vicuna-7B and Qwen2.5-7B in our latest revision**. Please see Figure 2 (for Vicuna-7B) and Figure 3 (for Qwen2.5-7B) in Appendix C.3 of the revision.
>
> Our current results show that: (1) For Vicuna-7B, when the radius $\epsilon$ becomes large, ER-CAT maintains jailbreak robustness similar to CAT, but the utility of CAT significantly decreases. (2) For Qwen2.5-7B, varying the radius $\epsilon$ can cause the utility of CAT to fluctuate significantly, while ER-CAT can achieve better jailbreak robustness in most settings. Additionally, these figures show that the chosen radius of 0.05 in our main experiments balances utility and robustness well for ER-CAT, i.e., making the utility of ER-CAT not too weak and the ASR achieved by ER-CAT not too high.
>
> **Q7.1: From the current writing, it is not completely clear which one you treat as the main CAT baseline. I will assume that is [r1] as Eq.(4) in your paper. Please note that in [r1] there are two related methods, CAT and CAPO. Could you please clarify exactly which variant you compare against?**
>
> **A7.1:** We would like to clarify that the baseline we compared to in our experiments is the CAT method in [r1] (i.e., Eq.(4) in [r1]). The reason for using CAT instead of CAPO is that our current theoretical framework is for the supervised learning setting, so our empirical analysis should also focus on the same setting to justify our theory (CAT is a supervised learning method); however, CAPO is a (preference-based) reinforcement learning method.
>
> **Q7.2: Also, if some of your experimental settings are aligned with [r1] (like CAT), then it would be good just to state this explicitly and please consider to report your results under exactly the same conditions. That would make the comparison more apple-to-apple.**
>
> **A7.2:** Thanks. We would like to clarify that for the implementation of the CAT baseline, we follow most of the settings as in [r1] except:
>
> - **Number of epochs.** We train every model using CAT or ER-CAT for $60$ iterations, which is approximately $4.8$ epochs (since $60*8/100 = 4.8$), while [r1] trains the model using CAT for $5$ or $2$ epochs. However, we note that the "$4.8$ epochs" is close to "$5$ epochs".
>
> - **Learning rate schedule.** We use a fixed learning rate ($2\times 10^{-4}$) for both CAT and ER-CAT, while [r1] initializes the learning rate for CAT as $2\times 10^{-4}$ and anneals it following a cosine schedule.
>
> - **Embedding space perturbation radius $\epsilon$.** We fix this radius to $0.05$ for all models, while [r1] may set different radii for different models. Nevertheless, we have added an ablation study on how this radius affects the performance of trained models. Please see **A6** for details.
>
> - **Utility data max sequence length.** We set this max sequence length to $1,024$, while in [r1] this sequence length is set to $256$. However, we note that a larger max sequence length would help the trained model preserve better utility.
>
> - **Cut-off values for toward and away losses.** We set the cut-off values for toward and away losses to $1.0$ and $-3.0$, while [r1] sets them to $0.5$ and $-5.0$. The reason for doing this is that we found that the original cut-off values from [r1] would result in the utility of a model trained with CAT being too weak. Please see Table 5 and the discussion in Appendix C.1 in our paper for details.
>
> **Q7.3: Also, whether the Table 3 time in your paper comes directly from [r1] or from your own implementation?**
>
> **A7.3:** Thanks. We would like to clarify that (1) the CAT baseline [r1] adopted in our experiments was completely implemented by ourselves, and (2) the time cost in Table 3 was calculated with our self-implemented CAT code and ER-CAT code with the same computation resources (2xA100@80G for Llama-3.1-8B experiments and 1xA100@80G for other experiments).

---

> ### Author Response · Authors · 2025-11-26
> **Thank you for your feedback & Further response (2/2)**
>
> **Q8: For the plots of the singular value distribution, I suggest adding average value just to provide a complete version.**
>
> **A8:** Thanks for your suggestion. We have revised our paper and plotted the averaged singular values along training in Figure 1(d) in Appendix C.3. From the figure, we find that our ER-CAT does not change the mean of singular values too much (the change in the mean is less than 2% on every base model). This does not contradict the training goal of ER-CAT, as ER-CAT aims to concentrate the singular values but not reduce their mean.
>
> **Q9: In some places (e.g., Table 1 caption ) the paper says "A high ASR indicates a strong model robustness." However, should it be, a lower ASR means the model is more robust?**
>
> **A9:** Yes, it should be "a **low** ASR indicates a strong model robustness". We are really sorry for these typos that may cause potential misunderstandings… We have fixed these kinds of typos in Table 1 (caption; in Section 5.2) and Table 6 (caption; in Appendix C.3). Please refer to our latest revision for details.
>
> Thanks again for pointing this out!
>
> **References**
>
> [r1] Xhonneux et al. Efficient Adversarial Training in LLMs with Continuous Attacks. NeurIPS 2024.

---

### Official Review · Reviewer_8RQd · 2025-11-01

**Soundness:** 3
**Presentation:** 2
**Contribution:** 2
**Rating:** 6
**Confidence:** 4

**Summary:**

This paper takes a step toward understanding why adversarial training (AT) in the embedding space provides robustness to attacks in the token space. To enable analysis, they focus on linear self-attention with embedding module (LSA-E) trained on linear regression ICL tasks and prove that embedding-space AT on LSA-E yields an upper bound on the robust generalization risk of LSA-E models under ICL suffix adversarial attacks, an attack in the token space.

They show that this upper bound (1) has a negative correlation with the adversarial perturbation during AT; therefore, a larger perturbation radius provides better input-space robustness, and (2) links robustness to singular values of the embedding matrix. Motivated by this, they propose Embedding-Regularized CAT (ER-CAT), which adds a penalty on the variance of the singular values of the embedding matrix to push them away from being too large or too small. They compare ER-CAT with CAT on several open-source LLMs and jailbreak attacks to show that ER-CAT achieves a better robustness-utility tradeoff.

**Strengths:**

1. The theoretical analysis on LSA-E models and the robust generalization upper bound is an interesting contribution that sheds light on how robustness of embedding-space adversarial training is connected to the singular values of the embedding matrix and the adversarial perturbation radius.

2. Motivated by the theoretical findings, the paper proposes ER-CAT, a singular-value-based regularization for CAT, and presents experiments where ER-CAT provides some improvements in model robustness (lower ASR) while maintaining approximately similar utility. Although the evaluation setup could be improved (see below), this shows some promise in using embedding-based regularization within CAT.

**Weaknesses:**

1. The connection between ICL embedding AT for LSA-E models and continuous AT in LLMs is underexplained. The theoretical analysis relies on a linear self-attention surrogate and does not capture non-linearities in modern LLMs. Although the paper claims ICL embedding AT is a good approximation for real-world LLM CAT, this is insufficiently supported, and while the empirical results are encouraging, it's unclear when and why robustness gains proven for LSA-E transfer to CAT on large non-linear multilayer transformers.

2. The empirical results present robustness (Table 1) and utility (Table 2) separately, making it difficult to evaluate whether ER-CAT truly provides a better robustness-utility tradeoff than CAT. When ER-CAT achieves lower ASR (better robustness), it often comes at the cost of reduced utility. The paper would benefit from visualizations such as ROC-style curves that plot utility against robustness, allowing direct comparison of the methods at equivalent operating points.

3. Writing/notation issues:
- The loss defined for continuous AT in equation (3) has a mistake. Given that in the text, $y$ is the target harmful resposne and $\tilde{y}$ is the safe response, the current version of the loss in equation (3) is encouraging the harmful response and decreasing the likelihood of the safe response. $y$ and $\tilde{y}$ should be swapped in the Adversarial Loss. This holds similarly for equation (4).

- (minor point) In several of the equations (such as equations (7) and (9), dot (".") is used for matrix multiplication, which to me is a non-standard and confusing notation. I suggest using dots only for dot products and not matrix multiplication.

**Questions:**

1. In the ICL linear-regression surrogate, what is the intended correspondence between ($x_{\tau, i}$, $y_{\tau,i}$) and standard LLM training data? Specifically, should $x_{\tau, i}$ be thought of as a single-token feature vector or a compressed/pooled representation of a multi-token prompt example, and if the latter, how is it modeled, and at which stage relative to $W_E$? Also, in this case, how realistic is the modeling choice that $x_{\tau, i}, x_{\tau,q} \sim N(0, \Lambda)$? A clarification on this helps in understanding the modeling better.

2. What is the reason behind choosing ICL suffix embedding attack for the robust generalization risk analysis? What other types of token-space adversarial attacks do you expect your analysis to extend to?

---

> ### Author Response · Authors · 2025-11-25
> **Rebuttal Part [1/3]**
>
> Thank you for your thorough comments and kind support!
>
> **Q1: The connection between ICL embedding AT for LSA-E models and continuous AT in LLMs is underexplained. The theoretical analysis relies on a linear self-attention surrogate and does not capture non-linearities in modern LLMs. Although the paper claims ICL embedding AT is a good approximation for real-world LLM CAT, this is insufficiently supported, and while the empirical results are encouraging, it's unclear when and why robustness gains proven for LSA-E transfer to CAT on large non-linear multilayer transformers.**
>
> **A1:** Thanks for this good question. Firstly, we respectfully note that many existing works have empirically shown that a linear self-attention surrogate is a good theoretical artifact for approximating and understanding the properties of large, complex, highly non-linear LLMs. Specifically, [r1] has empirically shown that the linear self-attention module in LSA-E models shares similar properties with the non-linear ones in LLMs and thus is useful for theoretically understanding LLMs. [r2] has further succeeded in using simple single-layer linear transformers to explain the robustness of LLMs in certain aspects. Because of the advances in these works, we believe that while LSA-E models are significantly simplified when compared with real large-scale LLMs, they are still a good enough surrogate for theoretically understanding the mechanism behind real-world LLM CAT.
>
> Secondly, our new theory-inspired ER-CAT algorithm for LLMs also empirically justifies that LSA-E models are suitable for theoretically studying the AT problems of real-world LLMs. Specifically, our main theoretical result, i.e., the robust generalization upper bound in Theorem 2, suggests that the robustness of LSA-E models trained with embedding AT is closely related to the eigenvalues of the embedding matrix $W^E$. For the trained model to have strong adversarial robustness, the eigenvalues of $W^E$ should be neither too large nor too small. Based on this observation, we designed our ER-CAT algorithm for LLMs by enforcing the eigenvalues of their embedding matrices to concentrate during training. Experimental results in Table 1 in Section 5 show that ER-CAT improved the jailbreak robustness of real-world LLMs, which in turn justifies our theoretical findings. This indicates that our LSA-E models are good characterizations of real-world complex LLMs.
>
> Additionally, we have added a new Section 4.2 in our revised paper to discuss and bridge the gap between the ICL embedding AT for theoretical LSA-E models and the CAT for real-world LLMs. Please refer to our revision for more details.
>
> **Q2: The empirical results present robustness (Table 1) and utility (Table 2) separately, making it difficult to evaluate whether ER-CAT truly provides a better robustness-utility tradeoff than CAT. When ER-CAT achieves lower ASR (better robustness), it often comes at the cost of reduced utility. The paper would benefit from visualizations such as ROC-style curves that plot utility against robustness, allowing direct comparison of the methods at equivalent operating points.**
>
> **A2:** Thank you for your suggestion. We have conducted a preliminary analysis of the utility-robustness tradeoff for ER-CAT. Specifically, we vary the embedding regularization coefficient $\beta$ in ER-CAT within the range $[0,1]$ and calculate the utility (LC-WinRate) and ASRs of the correspondingly trained LLMs. Please see Table 4 in Section 5.2 of our revised paper. Our current results show that varying this coefficient $\beta$ does not significantly change the utility or the robustness of the model trained with ER-CAT. We deduce that this is because we use the AdamW optimizer for training, and the gradient normalization procedure in AdamW implicitly reweights the ER-CAT objective to mitigate the effect of tuning coefficients for different terms.
>
> We will add a more comprehensive analysis of the utility-robustness tradeoff for both CAT and ER-CAT to the final version of our paper and will better compare their performance.

---

> > ### Author Response · Authors · 2025-11-25
> > **Rebuttal Part [2/3]**
> >
> > **Q3.1: The loss defined for continuous AT in equation (3) has a mistake. Given that in the text, $y$ is the target harmful response and $\tilde y$ is the safe response, the current version of the loss in equation (3) is encouraging the harmful response and decreasing the likelihood of the safe response. $y$ and $\tilde y$ should be swapped in the Adversarial Loss. This holds similarly for equation (4).**
> >
> > **A3.1:** Thanks for pointing out this. We indeed mistakenly reversed the positions of the safe response $\tilde y$ and the harmful response $y$ in Eq.(3) and Eq.(4). We have fixed this mistake in our revision.
> >
> > **Q3.2: (minor point) In several of the equations (such as equations (7) and (9), dot (".") is used for matrix multiplication, which to me is a non-standard and confusing notation. I suggest using dots only for dot products and not matrix multiplication.**
> >
> > **A3.2:** Thanks for your suggestion. We have removed all "dot notations" that originally denoted matrix multiplication from the main text. Please see our revised paper for details.
> >
> > **Q4.1: In the ICL linear-regression surrogate, what is the intended correspondence between $(x_{\tau,i}, y_{\tau,i})$ and standard LLM training data? Specifically, should $x_{\tau,i}$ be thought of as a single-token feature vector or a compressed/pooled representation of a multi-token prompt example, and if the latter, how is it modeled, and at which stage relative to $W_E$?**
> >
> > **A4.1:** Thanks. We clarify that for the in-context sample $(x_{\tau,i}, y_{\tau,i})$ in the ICL (sequential) input $Z_\tau$, (1) the point $x_{\tau,i}$ corresponds to the notion of **"token"** in real-world LLMs and (2) the label $y_{\tau,i}$ corresponds to the **"next-token prediction label" (i.e., the index of the next-token in the discrete token space)** in LLMs. Furthermore, the ICL (sequential) input $Z_\tau$ is what corresponds to the notion of **"(multi-token) prompt"** in LLMs.
> >
> > Such correspondences aim to enable the ICL prediction process of LSA-E models to approximate the next-token prediction process of real-world LLMs. The discrepancy is that in LLMs each "token label" is fused with its corresponding "next-token", while in ICL theory each "token label $y_{\tau,i}$" is explicitly retained.
> >
> > Additionally, we note that the embedding matrix $W^E$ in theoretical LSA-E models can well approximate the embedding matrix in real-world LLMs. Specifically, if we replace each token in an LLM prompt with its one-hot encoding vector defined over the token vocabulary space, the embedding process for each token in an LLM can be seen as a matrix multiplication between the LLM's embedding matrix and the corresponding token's one-hot encoding.
> > This matrix multiplication-based LLM prompt embedding process is almost identical to the ICL input embedding process in LSA-E models, where input features are also linearly transformed by the LSA-E model's embedding matrix $W^E$.
> >
> > **Q4.2: Also, in this case, how realistic is the modeling choice that $x_{\tau,i}, x_{\tau,q} \sim N(0,\Lambda)$? A clarification on this helps in understanding the modeling better.**
> >
> > **A4.2:** As explained in **A4.1**, the in-context point $x_{\tau,i}$ or $x_{\tau,q}$ corresponds to a single token feature in LLMs, which is usually set as a one-hot encoding vector defined over the token space. We admit that there is a discrepancy between a Gaussian vector and a one-hot encoding token feature. However, we also note that it is a standard practice to adopt Gaussian in-context samples in ICL theoretical analysis, and existing works have already succeeded in using such an ICL theoretical framework to explain different properties of real-world LLMs [r2, r3]. As a result, we believe using $x_{\tau,i}, x_{\tau,q} \sim N(0,\Lambda)$ in our theoretical analysis is strong enough to approximate real-world LLMs and reveal the mechanism behind LLM CAT.

---

> > > ### Author Response · Authors · 2025-11-25
> > > **Rebuttal Part [3/3]**
> > >
> > > **Q5: What is the reason behind choosing the ICL suffix embedding attack for the robust generalization risk analysis? What other types of token-space adversarial attacks do you expect your analysis to extend to?**
> > >
> > > **A5:** Firstly, we respectfully argue that the attack that we use for the robust generalization risk analysis is the **"ICL suffix adversarial attack"** but not the **"ICL suffix embedding attack"**. More specifically, during the test-time robustness analysis, adversarial perturbations are performed only in the original input space of LSA-E models but not in the embedding space.
> > >
> > > Secondly, to the best of our knowledge, the ICL suffix adversarial attack proposed by [r2] is the first and currently the only attack that can characterize real-world jailbreak attacks against LLMs under the ICL theoretical framework. Since our paper focuses on theoretically understanding the working mechanism of LLM CAT but not new families of jailbreak attacks, we therefore directly adopt the ICL suffix adversarial attack into our theoretical analysis to assess the robustness of LSA-E models.
> > >
> > > Finally, while the ICL suffix adversarial attack by [r2] was originally designed to approximate token-level suffix jailbreak attacks (i.e., GCG attack), we believe it can also be extended to analyze prompt-level attacks (e.g., PAIR attack). Concretely, a prompt-level attack would usually rewrite the whole input sequence adversarially. Such a rewriting strategy can be (approximately) seen as **adversarially perturbing every token in the input sequence**. Thus, to extend the ICL suffix adversarial attack, for the adversarial ICL input $Z^{\mathrm{adv}}\_{\tau,M}$ defined in Eq.(6), one can simply add adversarial perturbations to every ICL training sample (instead of a suffix) to approximate prompt-level attacks.
> > >
> > > **References**
> > >
> > > [r1] Ahn et al. Linear attention is (maybe) all you need (to understand transformer optimization). ICLR 2024.
> > >
> > > [r2] Fu et al. Short-length Adversarial Training Helps LLMs Defend Long-length Jailbreak Attacks: Theoretical and Empirical Evidence. NeurIPS 2025.
> > >
> > > [r3] Shi et al. Why Larger Language Models Do In-context Learning Differently? ICML 2024. https://arxiv.org/abs/2405.19592

---

### Meta-Review · Area_Chair_Zwyc · 2026-01-04

**Summary:**

All in all, I find the paper important from a theoretical point of view as it attempts (compellingly) to explain why CAT works in practice. This is a very important problem with both practical and theoretical implications for LLM adversarial training - a topic that has seen a lot of research interest recently and holds huge potential to improve current AI's security. The practical implications are experimentally demonstrated by the authors by showing that one can modify a standard CAT based on their theoretical analysis results to boost its performance. That said, I find that the paper can be further strengthened both practically and theoretically. From a theoretical point of view, there are parts of the ICL theoretical model that do not yet neatly translate to the practical CAT, GCG, and LLM training (see below). From a practical point of view, the results are not yet clear-cut, about the improved robustness-utility tradeoff (see below). The authors are strongly suggested to better address this in the writing, and emphasize the experiments in Figures 2 and 3 in the main paper. Further, the choice of loss, while loosely based on Theorem 2, still appears non-principally chosen. All in all, however, I think that the positives of the paper outweigh its negatives.

**Reviewer Concerns:**

**Outstanding Reviewer concerns**
- **ROC-style curves for CAT and ER-CAT (Reviewer 8RQd, Q2)**
I think that the provided $\beta$ experiment doesn't truly provide an ROC-style experiment, as by the authors' own admission, $\beta$ seems not to be currently well correlated with utility and robustness. Instead, it seems that the newest $\epsilon$ experiments in Figure 2/3 provide much better ROC-style curves. What I find important there is that ER-CAT seems more stable than CAT in terms of $\epsilon$ behaviour (which has been a long-standing q in the community), and that Figures 2 and 3 give a better understanding of Tables 1 and 2. I would therefore compel the authors to do Figures 2 and 3 for all models and put them in the main paper. Another alternative to vary the utility-robustness curve is proposed in [1], where the LoRA weight strength is varied, which the authors can also consider.
- **Correspondence between model risk on ICL suffix attacks and GCG (Reviewer 8RQd, Q5)**
Eq. 11 and the corresponding Eq. 12 talk about the worst-case bounded perturbation of the last M tokens (in ICL). While this attempts to model the generalization risk to GCG-style attacks, I want to point out to the authors that GCG would not perturb the last M-tokens - it will add completely new ones in an unbounded way. Further, a jailbreak-style attack (e.g. PAIR) will also unboundedly change all tokens (even a simple swap of the order of adjective tokens that completely keeps the meaning in a rewriting attack represents an unbounded change to two tokens). Thus, there is a discrepancy that the authors do not comment on between the robustness model they prove and the perturbation model of actual attacks. I think the authors should extend Section 4.2 to comment on this.

- **The results provided regarding the performance of ER-CAT are somewhat inconclusive (Reviewer 3RMS, Q2)**
I am still a bit uncertain about the experimental proof that the ER-CAT **always** provides a better utility-robustness trade-off  compared to regular CAT. There are experiments (especially the Llama 3.1 8B) where ER-CAT clearly achieves slightly worse robustness **and** slightly worse utility. I have authored papers in the area, and I understand the randomness of these measures and why not to over-examplify single experiments, but I also find it a bit problematic not to acknowledge this in the paper. Further, I actually think that the $\epsilon$ experiments in Figures 2 and 3 are more compelling proof of the claim than Tables 1 and 2, and will compel the authors to extend them to all models. I will also compel authors to provide the promised experiments to other questions for Reviewer 3RMS.

**Outstanding AC concerns**
- **How tight is the surrogate and bound? (very important)**
I would like the authors to comment in the main paper on the tightness of their surrogate problem (Eq. 13) and the tightness of the bound in Theorem 2. Ideally, this is done both empirically and theoretically, but at least one of those will substantially strengthen the paper. As it stands currently, I do not know if the bounds provided are not completely vacuous. I also think it prevents future work, as theoreticians do not know if they can qualitatively improve the bound (e.g., by a substantial big O factor) or not.
- **The variance loss in ER-CAT is a bit random and unprincipled (important)**
I do understand the point of the authors that not too low and not too high $\sigma_i$s are suggested by the bound in Theorem 2. However, I find the argument a bit too hand-wavy, especially for a theoretical paper. In particular, it is not clear to me that the bound in Theorem 2 is optimized when the $\sigma_i$s' variance is the smallest. I think the authors could probably actually exactly optimize the $\sigma_i$s w.r.t. the bound. Further, as justified right now, there are other simpler alternatives from an optimization point of view to the variance. One is given by the authors themselves ($\sigma_{max} - \sigma_{min}$), which seems to be much easier to optimize and might give better results. Therefore, the authors should justify better the variance loss in the paper and experiment with other alternative losses like $\sigma_{max} - \sigma_{min}$ and the bound in Theorem 2 itself ($\sum_i \sigma_i^4/(\sigma_{min}^4 + \epsilon^4)$).

**Outstanding Typos**
- Missing $w$ in Remark 1: Should be "$ w^V_{21} $ to ensure symmetric initialization".
- Text still says $\beta=0.2$ in the paragraph **Adversarial training**.

[1] https://arxiv.org/abs/2505.16947

**Reviewer Scores:**

- **Reviewer #8RQd:**
I think the reviewer would have maintained their rating of 6. In particular, the authors address well the correspondence between ICL and CAT training. However, I believe the Reviewer's ROC-style curve experiment suggestion was not truly implemented through the $\beta$ experiment (especially given the result of the experiment) and that the answer to the question regarding the correspondence of the token attack model in ICL and GCG is still not completely satisfactory.

- **Reviewer #hwXR:**
I think the reviewers' questions were largely addressed and that the reviewer would have raised their score to a 6.

- **Reviewer #iaCj:**
The reviewer has already raised their score to 8. While I think this is a bit inflated, it seems to reflect well the discussion that particular reviewer had with the authors

- **Reviewer #3RMS:**
I find the added $\epsilon$-experiments hugely important. I think they have improved the paper hugely. That said, I am still a bit uncertain about the experimental proof that the ER-CAT **always** provides a better utility-robustness trade-off  compared to regular CAT. There are experiments (especially the Llama 3.1 8B) where ER-CAT clearly achieves slightly worse robustness **and** slightly worse utility. I have authored papers in the area, and I understand the randomness of these measures and why not to over-examplify single experiments, but I also find it a bit problematic not to acknowledge this in the paper. I think, therefore, the reviewer would have ended up retaining a score of 6 (after the initial raise).

---

### Decision · Program_Chairs · 2026-01-26

Accept (Poster)